# SHARED DYNAMIC MODEL-ALIGNED HYPERNETWORKS FOR ZERO-SHOT GENERALIZATION IN CONTEXTUAL REINFORCEMENT LEARNING

## ABSTRACT

Zero-shot generalization in contextual reinforcement learning (RL) remains a core challenge, particularly when explicit context information is unavailable and must be inferred from data. We propose DMA*-SH, a framework based on dynamics model-aligned (DMA) context inference, where a shared hypernetwork jointly parameterizes the dynamics model, policy, and action-value function. This design enforces consistency between learned context representations and transition dynamics, while normalization and random masking in the context encoder improve stability and robustness. To evaluate our method, we introduce the Actuator Inversion Benchmark (AIB), distinguishing overlapping from non-overlapping contexts, the latter generated via a discontinuous action sign flip that is provably unsolvable under standard domain randomization. We formalize the strict expressiveness advantage of DMA*-SH over concatenation-based approaches in non-overlapping settings, and show that the shared hypernetwork acts as an implicit regularizer steering RL gradients towards dynamically coherent solutions. Across the AIB benchmark, DMA*-SH delivers strong zero-shot generalization and outperforms both context-aware and context-unaware baselines, with the largest gains in non-overlapping contexts. Our results show hypernetworks enable effective and scalable context inference.

## 1 INTRODUCTION

Reinforcement Learning (RL) has achieved remarkable success in solving complex tasks such as robotic manipulation (Nair et al., 2018) and locomotion (Duan et al., 2016a). Yet, a persistent challenge remains: RL agents often struggle to generalize when exposed to variations in task dynamics, such as changes in object mass or surface friction (Moos et al., 2022). These variations typically require extensive retraining, which undermines the robustness and adaptability of learned policies (Beck et al., 2023a). The problem is particularly acute in sim-to-real transfer, where discrepancies between simulated and real-world dynamics frequently lead to instability and degraded performance.

To address these challenges, we develop a method for zero-shot generalization (Kirk et al., 2023) and robust representation learning within the framework of Contextual Markov Decision Processes (CMDPs) (Hallak et al., 2015). In this setting, each *context* corresponds to a distinct variation in transition dynamics, such as changes in physical properties like object mass or surface friction.

Prior work in contextual RL generally assumes either (i) the agent has access to explicit context information (*context-aware*), or (ii) the agent lacks such access (*context-unaware*) and must infer context implicitly from observed transitions. We focus on the latter, more challenging setting, aiming to learn latent context representations directly from data to enable robust, generalizable behavior across diverse environments.

**Main contributions.** This paper makes the following contributions:

- We propose a framework for contextual RL that incorporates dynamics model-aligned (DMA) context inference through a hypernetwork (Ha et al., 2017), jointly trained with the dynamics model and shared with both the policy and Q-function. We denote this architecture DMA*-SH. We show that this shared design implicitly regularizes RL gradients toward dynamically consistent and generalizable policies. Additionally, carefully chosen normalizations and random masking in the context encoder yield more stable context representations, facilitating zero-shot generalization.

- We introduce the Actuator Inversion Benchmark **(AIB)**, a suite of contextualized environments that systematically studies the challenges of multiplicative context interactions. AIB distinguishes *overlapping* from *non-overlapping* contexts, with the latter generated via "actuator inversion", a discontinuous action sign flip that induces qualitative shifts in the transition dynamics. We prove these non-overlapping contexts are unsolvable with standard domain randomization (Lemma 9), establishing the need for dedicated context inference.

- We prove that hypernetwork-conditioned ReLU policies strictly subsume concatenation-based counterparts (Theorem 3). This shows why DMA*-SH has the right inductive bias to exactly model actuator inversion through multiplicative modulation of action effects.

- We demonstrate that DMA*-SH learns *directionally selective representations*, compressing nuisance continuous variations while preserving and separating task-critical directions such as actuator inversion, which in turn enables robust zero-shot generalization.

- DMA*-SH achieves strong zero-shot performance across diverse dynamics variations, often surpassing ground-truth context-aware agents. Notably, in six challenging non-overlapping environments, DMA*-SH outperforms context-aware concatenation-based approaches by an average of 10.5%, and the context-unaware DR baseline by a staggering 159.3%, demonstrating robust zero-shot adaptation to extreme context shifts induced via actuator inversion.

Our code is available at `https://github.com/dma-sh/dma-sh`.

## 2 BACKGROUND

**Contextual reinforcement learning.** We formalize the problem using the *Contextual Markov Decision Process* (CMDP) framework (Hallak et al., 2015; Benjamins et al., 2023). A CMDP is defined by the tuple $(\mathcal{C}, \mathcal{S}, \mathcal{A}, P, r, \gamma)$, where $\mathcal{C}$ denotes the context space, $\mathcal{S}$ and $\mathcal{A}$ are the state and action spaces, $P^c(s'|s, a)$ specifies the transition probability from state $s$ to $s'$ under action $a$ in context $c$, $r^c : \mathcal{S} \times \mathcal{A} \to \mathbb{R}$ is the context-dependent reward function, and $\gamma \in (0, 1)$ is the discount factor. Each context $c \in \mathcal{C}$ defines a distinct MDP with shared $\mathcal{S}$ and $\mathcal{A}$ but possibly differing dynamics $P^c$ and/or reward function $r^c$. The context is assumed to be fixed within an episode. Following prior work (Beukman et al., 2023; Benjamins et al., 2023; Prasanna et al., 2024; Röder et al., 2025), we focus on variations only in the transition dynamics, keeping the reward function fixed across contexts, $r^c = r, \forall c \in \mathcal{C}$.

**Zero-shot generalization.** To evaluate generalization, we define three disjoint context sets: $\mathcal{C}_{\text{train}}$ for training, and $\mathcal{C}_{\text{eval,in}}$ and $\mathcal{C}_{\text{eval,out}}$ for evaluation, with $\mathcal{C}_{\text{train}} \cap \mathcal{C}_{\text{eval,in}} \cap \mathcal{C}_{\text{eval,out}} = \emptyset$ (Kirk et al., 2023). Contexts in $\mathcal{C}_{\text{eval,in}}$ are sampled from the same distribution as training contexts, while contexts in $\mathcal{C}_{\text{eval,out}}$ are out-of-distribution. During zero-shot evaluation, the agent is not allowed to adapt via gradient updates to either evaluation set. The agent aims to learn a policy $\pi_\theta$ that maximizes expected return over the training contexts:

$$\frac{1}{|\mathcal{C}_{\text{train}}|} \sum_c \mathbb{E}_{\pi_\theta} \Big[ \sum_{t=0}^{\infty} \gamma^t r(s_t, a_t) \Big] \tag{1}$$

where $s_{t+1} \sim P^c(\cdot|s_t, a_t)$ and the expectation is over trajectories following $\pi_\theta$ with $c \in \mathcal{C}_{\text{train}}$.

## 3 RELATED WORK

**Zero-shot generalization in contextual RL.** Contextual RL has been studied from multiple perspectives, including contextual MDPs, domain randomization, and meta-RL (Hallak et al., 2015; Modi et al., 2018; Beck et al., 2023a). A recent survey (Kirk et al., 2023) highlights its importance for zero-shot generalization, noting that separating training and evaluation context sets enables systematic evaluation. Broadly, two directions are distinguished: (1) explicit context is observable as privileged information (Chen et al., 2018; Seyed Ghasemipour et al., 2019; Ball et al., 2021; Eghbal-zadeh et al., 2021; Sodhani et al., 2021; Mu et al., 2022; Benjamins et al., 2023; Prasanna et al., 2024), and (2) context must be inferred implicitly from past experience (Chen et al., 2018; Xu et al., 2019; Lee et al., 2020; Seo et al., 2020; Xian et al., 2021; Sodhani et al., 2022; Melo, 2022; Evans et al., 2022; Ndir et al., 2024; Röder et al., 2025).

Our work follows the second approach, focusing on self-supervised context inference via dynamics-model alignment. Recurrent agents may also learn internal context representations (Grigsby et al., 2024a;b; Luo et al., 2024; Hafner et al., 2019; 2025), though these are typically not aligned with the underlying dynamics. Closely related, Beukman et al. (2023) employ hypernetworks (Ha et al., 2017) to integrate context into RL models. Our approach differs fundamentally, as we do not assume access to explicit context. Moreover, Beukman et al. (2023) train separate hypernetworks for policy and Q-function, whereas we train a single hypernetwork jointly with the dynamics model, which is then shared across policy and value networks.

**Meta-RL.** Meta-RL aims to enable agents to rapidly adapt to unseen tasks with minimal data (Beck et al., 2023a), often by learning policies that infer task structure from prior interactions, sometimes using hypernetworks (Beck et al., 2022; 2023b). However, most meta-RL approaches require fine-tuning on new tasks across multiple episode rollouts (Duan et al., 2016b; Finn et al., 2017; Nagabandi et al., 2018; Rakelly et al., 2019; Zintgraf et al., 2019), which is incompatible with the zero-shot generalization setting considered here. VariBAD (Zintgraf et al., 2020) and TrMRL (Melo, 2022) are not subject to this limitation, as they have been shown to adapt to the task within the first rollout. Recent advances in context-based offline meta-RL (COMRL) are also promising, as these algorithms aim to infer latent task information, both reward- and dynamics-based, from static datasets (Li et al., 2021; Dorfman et al., 2021; Yuan & Lu, 2022; Li et al., 2024a;b). However, since these methods typically rely on assumptions inherent to the offline setting, it remains an open question whether they can be robustly applied to online RL evaluation.

**Context in cognition.** Beyond RL, cognitive modeling suggests that humans segment the environment into context-like events (Zacks & Tversky, 2001; Zacks et al., 2007; Butz, 2016). For instance, the recurrent REPRISE model learns latent context representations from scratch, distinguishing dynamic regimes (Butz et al., 2019). More recent work differentiates event segmentation from context inference, showing that contextual priors support learning of sensorimotor repertoires and memory structures (Heald et al., 2021; 2023).

Bayesian active inference models indicate that context can reduce computational effort while accurately modeling human behavior (Marković et al., 2021; Schwöbel et al., 2021; Butz, 2022; Cuevas Rivera & Kiebel, 2023; Parr et al., 2023; Mittenbühler et al., 2024). In cognitive modeling–inspired deep learning, contextualized hypernetworks have been introduced in various forms, showing superior generalization and emergent compositionality (Sugita et al., 2011), the emergence of affordance maps (Scholz et al., 2022), as well as the possibility to focus object-oriented encoding pipelines (Traub et al., 2024). At the intersection of neuroscience, developmental psychology, cognitive modeling, and machine learning, context inference and context-conditioned learning appear critical for enabling robust behavioral learning in complex environments (Butz et al., 2024).

## 4 Context encoding and utilization

In this section, we first focus on representation learning for a **d**ynamic **m**odel **a**ligned (DMA) context representation. We highlight our enhancements to improve this representation, which we refer to as DMA*. We then introduce our approach that incorporates latent context information using a shared hypernetwork. We refer to this method as DMA*-SH, as it extends DMA* with a **s**hared **h**ypernetwork that jointly informs the dynamics model, policy, and value function.

### 4.1 Context inference by dynamic model-aligned representation learning

We denote by $\tau_t^c$ a sliding window of the past $K$ transitions from the same context $c$, each given as a tuple $(s_t, a_t, \delta s_{t+1})$, where $\delta s_{t+1} = s_{t+1} - s_t$ is the state difference. The sequence $\tau_t^c$ is passed through a *context encoder* $g_\phi(\tau_t^c)$ to produce a context representation $z_t$. The context encoder is trained jointly with a forward dynamics model $f_\theta$ that predicts the next state difference $\delta \hat{s}_{t+1}$ given the current state $s_t$, action $a_t$, and inferred context $z_t$. The objective is a reconstruction loss between predicted and true next state differences:

$$L_{\phi,\theta} = \|\delta \hat{s}_{t+1} - \delta s_{t+1}\|_2. \tag{2}$$

Next, we describe two key modifications that improve the quality and robustness of the learned context representations: random input masking and specialized normalization. An overview of the full context encoder pipeline appears in Figure 7 (Appendix C).

**Input masking.** Random masking of input features has been shown to improve representation learning across vision, language, and decision-making domains (Devlin et al., 2019; Liu et al., 2022; He et al., 2022). In our case, since the context encoder already relies on a forward dynamics prediction (cf. equation 2), we do not adopt the common masked prediction objective. Instead, we apply random masking independently to states, actions, and next state differences within $\tau_t^c$.

**Input normalization.** After masking, the concatenations of $(s_t, a_t, \delta s_{t+1})$ from $\tau_t^c$ are processed by a linear layer and then normalized via AvgL1Norm (Fujimoto et al., 2023). It normalizes each input vector by its average absolute value across dimensions:

$$\text{AvgL1Norm}(x) = \frac{x}{\frac{1}{N}\sum_i |x_i|}. \tag{3}$$

Unlike BatchNorm (Ioffe & Szegedy, 2015), which relies on running statistics and performs poorly in small-batch online RL, AvgL1Norm is statistic-free and per-sample, making it suitable for sliding $K$-step windows. It prevents monotonic growth in representation space (Gelada et al., 2019) while preserving relative scales, enabling consistent embeddings from $\mathcal{C}_{\text{train}}$ to $\mathcal{C}_{\text{eval,out}}$.

**Output normalization.** The normalized and masked sequence is fed into an LSTM, and the final hidden state is projected to a compact context embedding $z_t \in \mathbb{R}^8$. Finally, we normalize $z_t$ using SimNorm (Lavoie et al., 2023; Hansen et al., 2024), which projects the embedding into a $V$-dimensional simplex via a softmax. It stabilizes online RL by bounding the representation scale and promoting sparsity through soft penalties, without relying on batch statistics.

### 4.2 Context utilization by a shared dynamic model-aligned hypernetwork

In the vanilla DMA setup, the policy and Q-function receive the concatenation of the state $s_t$ and the inferred context $z_t$ as input (cf. Figure 1a). In contrast, we incorporate $z_t$ using a hypernetwork (Ha et al., 2017), which is a meta- or second-order neural network (Pollack, 1990; Sugita et al., 2011) that generates (hyper-)weights for a target network in an end-to-end differentiable manner. In our approach, the hypernetwork generates weights for only a subset of the main network. We refer to these second order parametrized parts as *adapters*.

As described in Section 4.1, the context representation $z_t$ is first inferred from past transitions in $\tau_t^c$ via dynamic model-aligned representation learning. A hypernetwork $h_\eta$ is then conditioned on $z_t$ to produce weights $\omega$ for the adapters in the dynamic model $f_{\theta,\omega}$, whose parameters are therefore split into generated weights $\omega$ and remaining base weights $\theta$. The parameters $\phi$, $\theta$, and $\eta$ for the context encoder, hypernetwork, and dynamic model, respectively, are updated jointly using the reconstruction loss:

$$L_{\phi,\theta,\eta} = \|\delta\hat{s}_{t+1} - \delta s_{t+1}\|_2. \tag{4}$$

Finally, without further modification, the generated adapter weights $\omega$ are shared with the adapters in the policy $\pi_{\xi,\omega}$ and Q-function $Q_{\zeta,\omega}$. In this way, the hypernetwork $h_\eta$ is fully aligned with the dynamic model. An overview of the shared hypernetwork architecture of DMA*-SH is shown in Figure 1b, while Figure 2 illustrates the interaction between a hypernetwork and a model network with an adapter.

### 4.3 Expressive Advantage of DMA*-SH via Multiplicative Adapters

A key architectural advantage of DMA*-SH over vanilla DMA arises from the use of hypernetwork-conditioned *multiplicative adapters* rather than simple concatenation. In vanilla DMA, the policy and Q-function receive the state $s_t$ and inferred context $z_t$ via concatenation: $\pi(s_t, z_t) = \text{MLP}([s_t, z_t])$. Here, the interaction between $s_t$ and $z_t$ is primarily *additive*: the effect of $z_t$ is mediated only through linear combinations at each layer. As a consequence, representing sharp or discontinuous context-dependent transformations of $s_t$ requires indirect approximation, which can necessitate larger networks or substantially more data. In contrast, DMA*-SH uses a hypernetwork $h_\eta(z_t)$ to generate adapter parameters $\omega = h_\eta(z_t)$ that modulate small modules injected into the policy, action-value, and dynamics models:

$$\pi(s_t, z_t) = f_{\text{base}}(s_t) + g_{\text{adapter}}(s_t; \omega = h_\eta(z_t)). \tag{5}$$

Since $g_{\text{adapter}}$ is parameterized by $\omega$, the context $z_t$ affects the forward computation *multiplicatively*: the adapter output rescales or transforms the contribution of the input $s_t$ in a context-dependent

manner. This form of multiplicative modulation is strictly richer than additive conditioning via concatenation (Jayakumar et al., 2020).

Theorem 3 in Appendix A.1 shows that hypernetwork-conditioned ReLU policies with multiplicative adapters strictly subsume concatenation-based ReLU policies. In non-overlapping context settings, DMA*-SH leverages hypernetwork conditioning to precisely capture hard discontinuities, such as policy sign flips, outperforming concatenation-based methods that cannot exactly model such multiplicative action effects (see Remark 4).

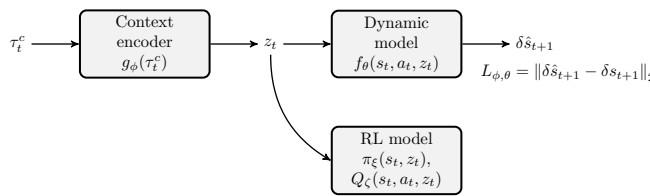

(a) Vanilla DMA: Context concatenated to dynamics and RL models.

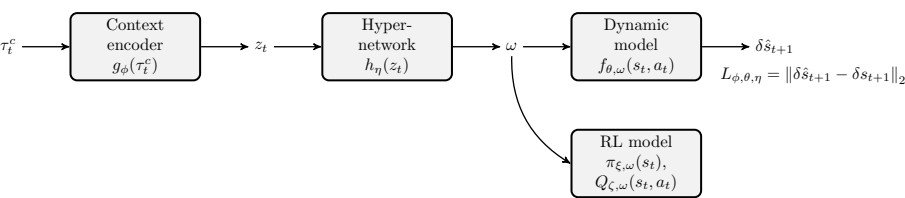

(b) DMA*-SH: Shared, dynamics-aligned hyperweights for dynamics and RL models.

Figure 1: Schematic overview of context utilization. (a) In vanilla DMA, the inferred context $z_t$ is provided as input to the RL models. (b) In our approach, DMA*-SH, a hypernetwork $h_\eta$ conditioned on $z_t$ generates adapter weights $\omega$, which are used by the dynamic model and RL networks. The hypernetwork and context encoder are trained jointly using the reconstruction loss $L_{\phi,\theta,\eta}$ (equation 4), while gradients through $h_\eta$ are stopped during RL updates.

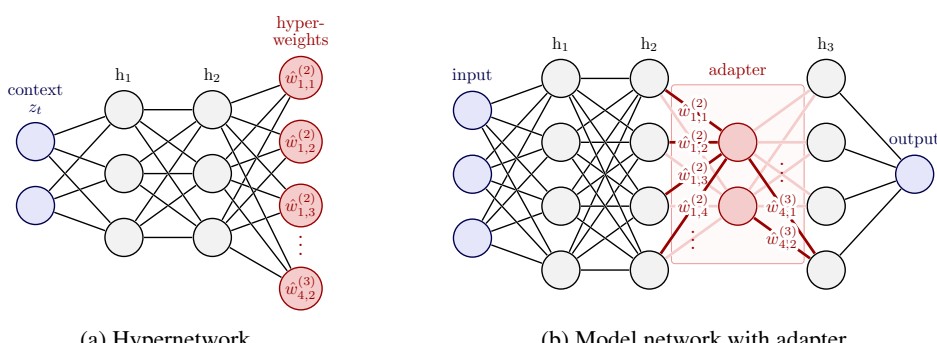

(a) Hypernetwork.      (b) Model network with adapter.

Figure 2: Network architecture. A hypernetwork (a) predicts parameters that are used within the dynamic model and RL networks (b).

### 4.4 SUMMARY OF DESIGN CHOICES

We briefly summarize the key design choices. Corresponding ablation studies are provided in Appendix E. Additional implementation details for the context encoder and hypernetwork architecture can be found in Appendix C. See Appendices A.3-A.4 for an extended discussion on how our design choices and shared hypernetwork architecture jointly shape the geometry of the learned context space.

**Input masking.** A masking ratio of 40% was found to be effective for DMA*-SH. In general, the method is robust to substantial masking ratios (cf. Figure 11 in the Appendix).

**AvgL1Norm input normalization.** Figure 12 in the Appendix compares various input normalization techniques within the context window, including LayerNorm (Ba et al., 2016), AvgL1Norm (Fujimoto et al., 2023), SimNorm (Lavoie et al., 2023; Hansen et al., 2024), and a custom WindowNorm that computes statistics across transitions in $\tau_t^c$. This comparison indicates that AvgL1Norm is most suitable for input normalization.

**SimNorm output normalization.** Output normalization was found to be crucial. Among LayerNorm, AvgL1Norm, and SimNorm, the best performance was achieved with SimNorm (cf. Figure 13 in the Appendix).

**(Hyper-)weight sharing.** Sharing the generated weights $\omega$ with the adapters in the policy $\pi_{\xi,\omega}$ and Q-function $Q_{\zeta,\omega}$ was more effective than training separate hypernetworks for the RL modules. An ablation using independent hypernetworks is presented in Figure 15 in the Appendix.

## 5 RESULTS

### 5.1 METRICS

We adopt a standard evaluation protocol for zero-shot generalization in contextual RL (Kirk et al., 2023; Beukman et al., 2023; Benjamins et al., 2023). Specifically, we sample $n_c = 20$ contexts from the environment-specific context ranges listed in Table 3 (Appendix F) to create the sets $\mathcal{C}_{\text{train}}$, $\mathcal{C}_{\text{eval,in}}$ and $\mathcal{C}_{\text{eval,out}}$, respectively. The agent is trained on $\mathcal{C}_{\text{train}}$. For evaluation, we measure the cumulative episodic return of the trained agent across $n_e = 10$ rollouts per context, then average within each context set. This yields three averaged episodic returns (AER) (Beukman et al., 2023), one per set. Following Agarwal et al. (2021), we report the interquartile mean (IQM) with empirical confidence intervals, after min–max scaling by environment-specific return bounds (Table 4, Appendix). Unless stated otherwise, results are averaged over $n_s = 10$ independent random seeds.

### 5.2 BASELINES

For our methods DMA* and DMA*-SH, as well as for all baselines except Amago, we use Soft Actor–Critic (SAC) (Haarnoja et al., 2018) as the underlying RL algorithm. To ensure comparability, we use SAC with its standard hyperparameters and avoid additional tuning. Full hyperparameter settings and implementation details are provided in Appendix C. All approaches are trained under the same procedure: the agent is trained in parallel across the $n_c = 20$ contexts in $\mathcal{C}_{\text{train}}$.

**Concat** (*Context-Aware*). This baseline assumes access to explicit context. Context is concatenated with the state and provided directly as input to the policy and Q-function. Despite its simplicity, this approach is widely used when context variables are available (Ball et al., 2021; Eghbal-zadeh et al., 2021).

**Decision Adapter (DA)** (*Context-Aware*). Beukman et al. (2023) propose a stronger context-aware baseline. Instead of concatenating context to the state, they use a hypernetwork architecture inside the policy (and optionally the Q-function), where parameters are adapted based on the context. This method achieves strong performance relative to other context-aware approaches such as FLAP (Peng et al., 2021) and cGate (Benjamins et al., 2023).

**Domain Randomization (DR)** (*Context-Unaware*). This baseline ignores explicit context, relying solely on domain randomization (Tobin et al., 2017) across multiple contexts.

**Amago** (*Context-Unaware*). In recurrent agents, latent information about the environment can accumulate over time, enabling in-context adaptation. Amago (Grigsby et al., 2024a) is a general-purpose in-context meta-RL algorithm, not specifically designed for contextual transition dynamics but nevertheless competitive. We use the improved Amago-2 variant (Grigsby et al., 2024b) and employ a GRU trajectory encoder.

**Dynamic Model Alignment (DMA)** (*Context-Inferred*). Prior methods such as DALI (Röder et al., 2025), IIDA (Evans et al., 2022), and CaDM (Lee et al., 2020) infer context from recent experience via dynamic model alignment. Typically, transition order is randomized to reduce temporal correlations, and dropout is applied to improve robustness. The resulting latent representation is passed to the policy and Q-function. As DMA* extends this paradigm, we include vanilla DMA as a baseline.

**DMA-Pearl** (*Context-Inferred*). Pearl (Rakelly et al., 2019) is a meta-RL algorithm that infers context with a probabilistic encoder trained via Q-function gradients. While Rakelly et al. (2019) evaluated Pearl only under reward variations, we adapt it to transition dynamics variations by training the context encoder jointly with a dynamic model. This yields a probabilistic extension of DMA, where the context representation is regularized by a KL penalty against a unit Gaussian prior $\mathcal{N}(0, I)$, weighted by $\beta = 0.2$ (see Figure 16 in the Appendix for ablations).

## 5.3 THE ACTUATOR INVERSION BENCHMARK (AIB)

We introduce the **Actuator Inversion Benchmark (AIB)**, a suite of contextualized environments designed to isolate and analyze challenges from multiplicative context interactions. AIB environments feature two context dimensions and are classified as either *(i) overlapping*, where context-unaware policies can achieve reasonable performance, or *(ii) non-overlapping*, where such policies provably fail (see Appendix A.2, Lemma 9 and Definition 6). Table 3 summarizes the contextualization schemes, including the ranges for $\mathcal{C}_{\text{train}}$, $\mathcal{C}_{\text{eval,in}}$, and $\mathcal{C}_{\text{eval,out}}$. Also see Appendix G.2. Unlike existing contextual RL benchmarks like CARL (Benjamins et al., 2023) that focus on continuous parameter variations, AIB systematically studies *discontinuous context shifts* through actuator inversion, creating qualitatively distinct policy requirements (e.g., policy sign flips).

To obtain true non-overlapping behavior between different context instances, the effect of a context variation has to be drastic with respect to the transition dynamics in the environment. For this reason, in some of the environments listed below we invert the action effect by multiplying the agent's intended action by $-1$, producing a discontinuous change in the transition dynamics. We refer to this action sign flip as actuator inversion. For a formal definition, refer to Appendix A.2.

Example 1 (Trackpad inversion). Consider the scrolling direction of a computer trackpad: some users prefer congruent scrolling (screen moves with the fingers), while others prefer inverted scrolling. When confronted with the non-preferred setting, it is impossible to operate effectively without adaptation.

Remark 2 (Actuator inversion for true non-overlapping contexts). In AIB, we employ actuator inversion as the canonical mechanism for creating non-overlapping contexts. It is canonical because it is both minimal and drastic: continuous parameters (e.g., mass, gravity) allow overlap (gradual policy shifts), whereas inversion forces binary incompatibility in the transition dynamics, so that policies that succeed in one context fail in the other (see Lemma 9). This provides a clean test of zero-shot generalization to extreme, discontinuous binary shifts. Actuator inversion represents the *hardest* scenario for context-inferred agents, as we show in Appendix A.2.3.

**DI.** We create a custom two-dimensional double-integrator environment without friction. The agent is a point mass initialized randomly in a corner and tasked with reaching the origin $[0, 0]$. Actions apply forces in the $x, y$ directions, and the state consists of positions and velocities. Rewards are sparse: $+1$ on reaching the goal, $0$ otherwise. Context is defined by the agent's mass and an actuator factor $(\pm 1)$, the latter creating *non-overlapping* contexts.

**DI-Friction.** Identical to DI but with friction. Context variables are mass and friction coefficient. Since contexts are continuous physical parameters, they are considered *overlapping*.

**ODE.** The environment from Beukman et al. (2023), governed by an ordinary differential equation parameterized by two context variables $c_0$ and $c_1$: $x_{t+1} = x_t + \dot{x}_t \, dt$, $\quad \dot{x} = c_0 a + c_1 a^2$. The goal of the agent is to control the action $a$ to keep the state close to $x = 0$. Context-unaware agents perform poorly, indicating weakly *non-overlapping* contexts (Beukman et al., 2023).

**Cartpole.** From the DM Control Suite (cartpole-balance-v0) (Tassa et al., 2018), the task is to balance a pole by applying horizontal forces to its base (Barto et al., 1983). This environment is contextualized by the pole length and similar to DI by an actuator factor which can either be $-1$ or $1$. Contexts are *non-overlapping*.

**BallInCup.** From the DM Control Suite (ball_in_cup-catch-v0) (Tassa et al., 2018). An actuated receptacle can move in the vertical plane in order to swing and catch a ball attached to its bottom. The reward signal is sparse, i.e., $+1$ if the ball is in the cup, $0$ otherwise. The environment is contextualized such that the tendon length and the gravity can be varied, similar to Röder et al. (2025). Since contexts are continuous physical parameters, they are considered *overlapping*.

**Walker.** From the DM Control Suite (walker-walk-v0) (Tassa et al., 2018). A planar walker is rewarded for moving forward (Lillicrap et al., 2015). Context variables are actuator strength (referred to here as an actuator factor) and gravity, following Prasanna et al. (2024). Since contexts are continuous physical parameters, they are considered *overlapping*.

## 5.4 ZERO-SHOT GENERALIZATION

When evaluating our proposed approaches, we place the main emphasis on zero-shot generalization. As outlined in Sections 2 and 5.1, we distinguish three evaluation regimes corresponding to the context sets $\mathcal{C}_{\text{train}}$ for training, and $\mathcal{C}_{\text{eval,in}}$ and $\mathcal{C}_{\text{eval,out}}$ for within- and out-of-distribution evaluation, respectively. IQM scores aggregated across all contextualized environments (cf. Figure 3) show that DMA* and DMA*-SH achieve strong generalization, particularly in the challenging out-of-distribution setting.

The strongest competitors are the context-aware Concat and DA baselines, which are consistently outperformed by DMA*-SH across all three regimes. Across the diverse environments and contextualization types, DMA*-SH also achieves consistently strong AER scores (cf. Table 1). We complement the aggregated metrics with a detailed analysis at the level of individual context instances. This highlights how performance varies as evaluation contexts diverge from the training distribution and exposes failure modes that aggregated statistics can obscure. Full per-context heatmaps, bar plots, learning curves, and results for additional environments are provided in Appendix G.

Interestingly, simple domain randomization suffices in the Walker environment, suggesting that in some cases explicit or inferred context information can even hinder performance. Despite not being specifically designed for variations in transition dynamics, the context-unaware Amago algorithm performs competitively in most environments, including those with non-overlapping contexts such as DI, which cannot be solved by simple domain randomization, as opposed to DI-friction with overlapping contexts.

DMA-Pearl achieves strong results in overlapping contextualizations. However, its smooth prior in the KL-term makes it uncompetitive in non-overlapping settings (see Remark 11 and Appendix E).

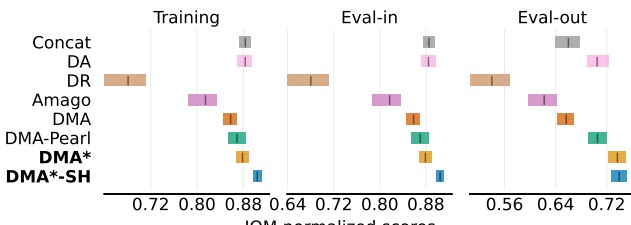

Figure 3: Interquartile mean (IQM) based on AER scores (cf. Section 5.1) aggregated across environments (cf. Section 5.3). Results are reported for the three context sets, and we compare our approaches DMA* and DMA*-SH against all baselines (cf. Section 5.2).

| | Context-Aware | | Context-Unaware | | Context-Inferred | | | |
| Name | Concat | DA | DR | Amago | DMA | DMA-Pearl | **DMA*** | **DMA*-SH** |
|---|---|---|---|---|---|---|---|---|
| DI | 72±5 | 75±1 | 16±12 | 61±15 | 63±3 | 68±3 | 75±1 | **76±1** |
| DI-friction | 65±23 | 76±2 | 69±23 | **79±1** | 56±23 | 74±2 | 68±23 | 77±1 |
| ODE | 162±10 | **179±9** | 63±15 | 168±2 | 166±6 | 171±10 | 175±8 | **179±5** |
| Cartpole | 863±35 | 892±59 | 644±78 | 639±119 | 900±38 | 884±67 | 927±38 | **967±20** |
| BallInCup | **918±16** | 872±29 | 845±51 | 634±197 | 906±19 | 901±15 | 893±24 | 884±24 |
| Walker | 770±22 | 775±33 | **789±22** | 745±26 | 754±63 | **792±19** | 769±25 | **790±31** |
| Norm. Mean | 0.79 | 0.82 | 0.57 | 0.71 | 0.76 | 0.81 | 0.82 | **0.84** |

Table 1: AER scores and standard deviations (cf. Section 5.1) for the contextualized environments in Section 5.3. Results are aggregated across context sets $\mathcal{C}_{\text{train}}$, $\mathcal{C}_{\text{eval,in}}$ and $\mathcal{C}_{\text{eval,out}}$. We compare our approaches DMA* and DMA*-SH against all baselines (cf. Section 5.2). Best AER scores are bold; if multiple methods are highlighted for an environment, their scores lie within 99% of the maximum. Environment-specific normalization factors are applied for *Norm. Mean* (cf. Section 5.3).

## 5.5 Variability of Context Representations and Informativeness

DMA* consistently outperforms vanilla DMA, even though the only difference lies in the context encoder and, consequently, the context representation $z_t$. To analyze how RL task performance relates to $z_t$, we introduce two evaluation criteria: *Informativeness* and *Variability*.

We construct three datasets of $n_d$ trajectories $\tau_t^c$, separately for $c \in \mathcal{C}_{\text{train}}$, $c \in \mathcal{C}_{\text{eval,in}}$ and $c \in \mathcal{C}_{\text{eval,out}}$. For each dataset and each trajectory we infer the corresponding context representation $z_t$ using the context encoder $g_\phi(\tau_t^c)$.

**Informativeness.** We measure how much information the learned embedding $z_t$ conveys about the true context $c$ using the mutual information $I(z_t; c)$, estimated via the $k$-nearest neighbors entropy estimator (Kraskov et al., 2004). This quantity reflects the encoder's ability to reliably distinguish different contexts: Higher $I(z_t; c)$ indicates that $z_t$ is more informative about the underlying context. Mutual information thus provides a natural metric for assessing the quality of learned context embeddings (Garcin et al., 2025).

**Variability.** We measure the spread of context representations $z \in \mathbb{R}^d$ within each dataset as $\frac{1}{d}\sum_{i=1}^{d}\text{Var}[z_i]$. Low Variability provides stable context signals for robust policy training.

Figure 4 shows that DMA*-SH enhancements, including input and output normalization, random input masking, and hypernetwork-based context utilization, consistently reduce Variability in $z_t$. We additionally observe that higher Informativeness does not necessarily translate to improved RL performance. Instead, lower Variability appears more important, likely because highly variable context representations can impede stable RL training. This pattern is consistent across all considered environments (Appendix D). t-SNE visualizations (Figure 8) further reveal why DMA*-SH achieves low Variability: it *compresses* overlapping dimensions (e.g. mass) while *separating* actuator-inversion modes. These findings are reinforced by a Representation-Overlap (RO) analysis. RO quantifies the average cosine similarity between all pairs of context-mean representations, reflecting their global directional concentration. DMA*-SH achieves the highest RO as it collapses irrelevant mass variation more aggressively, concentrating representations along a strongly shared axis. DMA*-SH thus learns *directionally selective* representations, compressing irrelevant variations while separating discontinuous ones, creating effective representations for robust zero-shot adaptation. See Appendices A.3–A.4 for a discussion of how our shared design amplifies directional concentration.

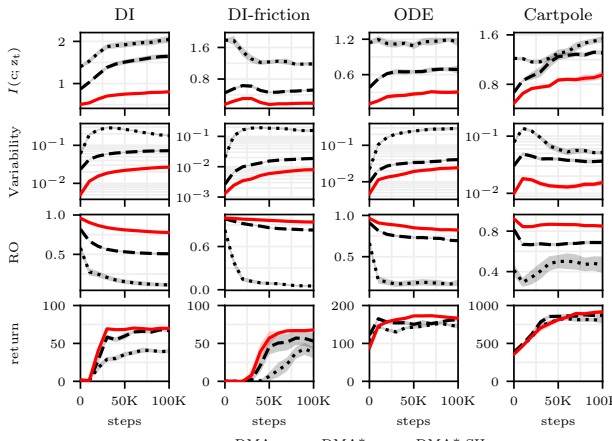

Figure 4: Mutual information $I(z_t; c)$, Variability, Representation-Overlap (RO), and episodic returns over training steps for the context set $\mathcal{C}_{\text{eval,out}}$. DMA*-SH shows consistently lower Variability than DMA* and DMA, and higher RO despite lower $I(z_t; c)$. The close correlation between low Variability, high RO and improved RL performance underscores the importance of stable context representations, while mutual information alone proves insufficient for predicting strong performance.

## 5.6 Implicit Gradient Regularization via Shared Hypernetworks

A single dynamics-trained hypernetwork in DMA*-SH acts as an *implicit gradient regularizer* for RL. By sharing hypernetwork parameters $\eta$ across dynamics and policy modules, RL gradients are constrained to operate under physically consistent features, reducing variance and improving generalization. To illustrate this effect, we focus on two key metrics: the mean gradient norm of the policy hypernetwork $\eta^\pi$ and the cosine similarity $\text{Cos}(\nabla_\eta L_d, \nabla_\eta L_\pi)$ between dynamics and policy gradients. Figure 5 shows that in the shared case (DMA*-SH; where $\nabla_\eta L_\pi$ is obtained through a "shadow" computation that hypothetically enables gradient flow from $L_\pi$ to $\eta$ during evaluation),

the *persistently high policy gradient norms* indicate that RL gradients continuously interact with dynamics-aligned parameters, while the cosine similarity highlights *strong coupling* between RL and dynamics objectives. In contrast, the separate case (DMA*-H) collapse policy hypernetwork gradients and show near-zero cosine similarity, reflecting uncoordinated adaptation. These results demonstrate that sharing $\eta$ implicitly regularizes the policy, directing RL gradients toward behaviors that are consistent with the underlying dynamics. For an analysis, see Appendix A.5 and Figure 6.

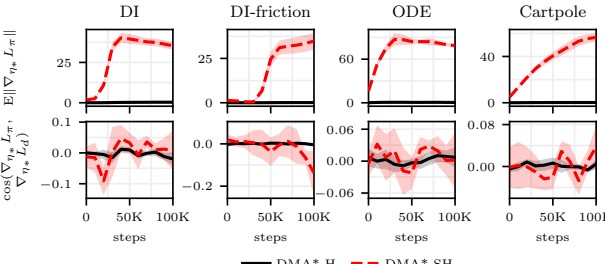

Figure 5: Implicit regularization of RL via a dynamics-trained shared hypernetwork. Top: Mean gradient norm of policy hypernetwork $\eta^\pi$ (shadow gradients for shared). Bottom: Cosine similarity $\mathrm{Cos}(\nabla_\eta L_d, \nabla_\eta L_\pi)$ between dynamics and policy gradients. Sharing $\eta$ amplifies alignment, enforcing dynamics-consistent RL updates.

## 6 CONCLUSIONS AND LIMITATIONS

We introduced DMA*-SH, a framework for contextual RL that leverages a shared hypernetwork to align latent context inference with the underlying dynamics model. By combining dynamics-model alignment with carefully chosen normalization and random input masking, DMA*-SH learns stable latent context representations that are shared with the policy and action-value networks through hypernetwork-generated adapter weights. Our normalization strategy combined with hypernetwork conditioning creates an approximately scale-controlled representation space in which functional modulation depends primarily on the *direction* of the context embedding. See Appendices A.3-A.4 for a detailed discussion. This results in hypernetwork-generated adapter weights that vary predominantly through directional differences in representation space, a phenomenon supported empirically by our t-SNE structure and Representation-Overlap analyses (Appendix D).

To stress-test zero-shot generalization, we introduced the *Actuator Inversion Benchmark* that highlights the challenges posed by multiplicative context interactions. We showed that hypernetwork conditioning confers a strict expressiveness advantage over concatenation-based architectures, enabling exact representation of hard discontinuities such as policy sign flips under actuator inversion. The dynamics-trained shared hypernetwork supplies an implicit form of gradient regularization that steers policy updates toward dynamics-consistent solutions. DMA*-SH achieves strong generalization by learning representations with beneficial geometric structure that compress overlapping contexts and separate non-overlapping ones, providing a principled approach to contextual policy learning. Together, these design principles account for DMA*-SH's superior zero-shot generalization across diverse settings, especially in adversarial non-overlapping contexts where standard domain randomization fails.

**Limitations and Future Work.** DMA*-SH inherits several structural constraints. The shared hypernetwork tightly couples dynamics learning with context inference, so model errors propagate directly into the latent context and can impair adaptation when dynamics are misspecified or rapidly shifting. The multiplicative modulation mechanism is well suited for actuator inversion and continuous parameter variations but may be less effective when context modifies reward structure or induces non-factorizable changes in optimal policy. Finally, since all modules depend on hypernetwork-generated weights, the capacity and conditioning of the hypernetworks become critical bottlenecks: too little capacity limits expressiveness, while too much capacity risks overfitting and reduces the stability benefits from shared dynamics alignment. Several extensions offer promising directions. Making DMA*-SH more robust to model uncertainty, for instance using ensemble dynamics or Bayesian hypernetworks (Krueger et al., 2017), may improve stabilty and resilence. Extending the framework to contexts that modify rewards may require dual hypernetworks or multi-view encoders. The geometric structure observed in learned representations suggests incorporating explicit information-theoretic or contrastive objectives (Li et al., 2024a;b) to strengthen compression of overlapping contexts and separation of discontinuous ones. Finally, real-robot deployments involving actuator degradation or intermittent faults provide natural testbeds for the multiplicative modulation mechanism and its ability to enable DMA*-SH to handle truly discontinuous dynamics shifts.

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

APPENDIX

## A  THEORETICAL RESULTS AND SUPPLEMENTARY ANALYSES

### A.1  MULTIPLICATIVE INTERACTIONS IN CONTEXTUAL POLICIES

The standard approach to conditioning a policy on a context embedding $z_t \in \mathbb{R}^d$ is through concatenation with the state $s_t$, yielding an input $[s_t; z_t]$ to a ReLU MLP policy (as in vanilla DMA and Concat baselines, Section 5.2). This imposes an *additive interaction* in the initial linear layer: $f(s_t, z_t) = W[s_t; z_t] + b$, with subsequent nonlinearities enabling approximation of interactions.

In contrast, *multiplicative interactions* enable richer fusions via bilinear forms: $f(s_t, z_t) = z_t^T W s_t + z_t^T U + V s_t + b$, where $W$ is a tensor capturing cross-terms between $s_t$ and $z_t$ (Jayakumar et al., 2020; Galanti & Wolf, 2020). Hypernetworks can be viewed as a structured instance of such multiplicative interactions (Jayakumar et al., 2020). Specifically, when a hypernetwork $h_\eta(z_t)$ generates affine weights $\omega = z_t^T W + V$ and bias $b' = z_t^T U + b$ for a linear policy layer, it exactly recovers the bilinear form, providing a dynamic, context-dependent modulation.

Inspired by the general strategy of Jayakumar et al. (2020), we show that hypernetworks strictly subsume concatenation for contextual ReLU policies:

**Theorem 3 (Hypernetworks are strictly more expressive than Concatenation for contextual ReLU policies).** Let $\mathcal{H}_{\text{concat}}$ be the hypothesis class of contextual policies implemented by ReLU MLPs that take the concatenated input $[s_t; z_t]$ and produce an action $a_t$. Let $\mathcal{H}_{\text{hyper}}$ be the hypothesis class of contextual policies implemented by a hypernetwork $h_\eta(z_t)$ (a ReLU MLP with a linear output layer) which outputs parameters $\omega$ for a policy network $\pi_{\xi,\omega}(s_t)$ (a ReLU MLP with a linear output layer). Assume $s_t$ and $z_t$ range over compact sets with non-empty interior. Then

$$\mathcal{H}_{\text{concat}} \subsetneq \mathcal{H}_{\text{hyper}}.$$

*Proof.* **Inclusion** ($\mathcal{H}_{\text{concat}} \subseteq \mathcal{H}_{\text{hyper}}$): Let $f \in \mathcal{H}_{\text{concat}}$ be implemented by a ReLU MLP with finite width and depth and parameter vector $\Theta$. Define a policy network $\pi_{\xi,\omega}$ with the same architecture as $f$ but whose weights are supplied by $\omega$. Choose the hypernetwork $h_\eta$ to be the constant function $h_\eta(z) = \Theta$ for all $z$. This constant map can be implemented by a ReLU MLP whose input-to-output weights are all zero and whose final-layer bias equals $\Theta$, where we crucially assume the hypernetwork's final layer is linear (so the bias may take any real value). Thus $\pi_{\xi, h_\eta(z)}(s) = f(s, z)$ for every $(s, z)$, so every function in $\mathcal{H}_{\text{concat}}$ is also in $\mathcal{H}_{\text{hyper}}$.

**Strictness** ($\subsetneq$): We produce a function $f^*$ in $\mathcal{H}_{\text{hyper}}$ that cannot be represented exactly by any element of $\mathcal{H}_{\text{concat}}$. Consider the scalar case $s, z \in \mathbb{R}$ and define $f^*(s, z) = s \cdot z$.

1. $f^* \in \mathcal{H}_{\text{hyper}}$: Take the policy to be a single linear unit $\pi_{\xi,\omega}(s) = \omega s$ and take the hypernetwork to be the identity map $h_\eta(z) = z$. Both of these maps can be realized exactly by (degenerate) ReLU MLPs with linear outputs (e.g. no hidden layer and output layer weights chosen appropriately). Then $\pi_{\xi, h_\eta(z)}(s) = zs = f^*(s, z)$.

2. $f^* \notin \mathcal{H}_{\text{concat}}$: Any function realized by a finite ReLU MLP is Continuous Piecewise Linear (CPWL) on its input domain (Montúfar et al., 2014). CPWL functions are affine on finitely many regions whose interiors cover the domain; therefore all second partial derivatives of a CPWL function vanish almost everywhere in the domain interior (the Hessian is zero almost everywhere). In contrast, the bilinear function $f^*(s, z) = sz$ has mixed second derivative $\partial^2 f^* / \partial s \partial z = 1$ everywhere, hence its Hessian is nonzero on any open subset of the domain. Consequently, $f^*$ cannot equal any CPWL function on a compact domain with non-empty interior, so $f^*$ is not representable exactly by any ReLU MLP on $[s; z]$. This contradiction proves $f^* \notin \mathcal{H}_{\text{concat}}$.

Therefore $\mathcal{H}_{\text{concat}} \subsetneq \mathcal{H}_{\text{hyper}}$. $\square$

The theoretical separation demonstrated above has direct implications for generalization. The hypernetwork's ability to exactly represent multiplicative interactions allows it to model specific context-dependent dynamics that a concatenated MLP can only approximate, often inefficiently. This approximation error can compound over a trajectory, leading to significant performance degradation, particularly in non-overlapping contexts where policies must be distinctly different.

**Remark 4 (The hypernetwork advantage of DMA*-SH for actuator inversion).** Actuator inversion provides a concrete illustration of the expressive gap (see Definition 6). Suppose the environment

contains a latent context variable $c \in \{-1, +1\}$, and the optimal policy is

$$\pi^\star(s_t, c) = c \cdot \pi_{\text{base}}(s_t),$$

corresponding to a sign flip in the action space. Since the agent does not observe $c$ directly, DMA*-SH infers it as $z_t \approx c$. A hypernetwork can map this inferred context to a scalar multiplicative factor:

$$h_\eta(z_t) = \omega_t \in \{-1, +1\},$$

and an adapter of the form $g_{\text{adapter}}(s_t; \omega_t) = (\omega_t - 1) \cdot f_{\text{base}}(s_t)$ yields (see equation 5)

$$\pi(s_t, z_t) = f_{\text{base}}(s_t) + g_{\text{adapter}}(s_t; \omega = h_\eta(z_t)) = f_{\text{base}}(s_t) + (\omega_t - 1) f_{\text{base}}(s_t) = \omega_t \cdot f_{\text{base}}(s_t).$$

So for $\omega_t \in \{-1, +1\}$, this realizes the sign flip $\pi(s_t, z_t) = \pm f_{\text{base}}(s_t)$ matching $\pi^\star(s_t, c) = c \cdot \pi_{\text{base}}(s_t)$ and realizing actuator inversion *exactly*.

More generally, a hypernetwork-conditioned ReLU module can implement transformations such as

$$\pi(s_t, z_t) = \text{ReLU}((I \cdot h_\eta(z_t)) s_t + b),$$

since $h_\eta(z_t)$ directly parameterizes the weight scaling. This direct multiplicative control is impossible to achieve through concatenation alone without approximation. To see this, recall that a concatenation policy (e.g., in Concat/DMA baselines) has the form $\pi_{\text{concat}} = \text{MLP}([s; z])$. To realize $\pi^\star(s, c) = c \cdot \pi_{\text{base}}(s)$ with an MLP that only sees $z$ concatenated, the network needs to produce the mapping that for $z$ in region corresponding to $c = +1$ outputs one linear map $A_{+1}s$ and for $z$ in region corresponding to $c = -1$ outputs $A_{-1}s$ with $A_{-1} = -A_{+1}$. While a sufficiently wide/deep ReLU MLP can approximate piecewise linear functions, constructing an exact global negation for all $s$ requires the network to implement a switch that selects two opposite linear operators. Practically, the network must learn precise decision boundaries in $z$-space that separate contexts, and then implement the two opposite linear maps. This is possible in principle but typically requires larger capacity and more data, and the resulting decision boundary can be brittle to encoder noise. Thus concatenation provides no simple, compact architectural path to exact multiplicative sign flips.

In environments with non-overlapping contexts created via actuator inversion (cf. Section 5.3), the optimal policy can exhibit discrete, high-magnitude shifts across contexts. The hypernetwork's multiplicative structure is an ideal inductive bias for this, efficiently modeling the context as a "switch" or "modulator" of the base policy. Table 7 shows a pronounced gap in Eval-out AER between the context-aware Concat and our DMA*-SH across the DI, Cartpole, and ODE environments.

**Remark 5 (Parameter complexity of DMA*-SH).** While hypernetworks introduce additional parameters and coupling, they allow for *exact* modeling of actuator inversion. A concatenation-based ReLU MLP must approximate such transformations through standard feedforward nonlinearities, requiring a complex combination of ReLU breakpoints to emulate a global sign flip across the entirety of the state space. Such mappings are nonlinear and discontinuous in the context variable, making them difficult to represent exactly with fixed shared weights. Consequently, concatenation models often need larger width, depth, or more training data to approximate the same transformation that multiplicative adapters implement directly. Importantly, the hypernetwork generates only a subset of adapter weights, and the increased parameter count is offset by lower sample complexity and improved zero-shot generalization, especially in challenging non-overlapping contexts.

Training curves in Figure 18 demonstrate that DMA*-SH converges in comparable or fewer steps than the concatenation baseline DMA while achieving superior zero-shot performance on non-overlapping contexts (Table 7). This indicates that the expressive advantage outweighs the additional parameter cost.

## A.2 DIFFICULTY OF OVERLAPPING VS. NON-OVERLAPPING CONTEXTS

We examine how overlapping and non-overlapping context structures influence task difficulty and the stability of learned policies.

**Definition 6 (Overlapping and Non-Overlapping Contexts).** Let $\mathcal{C}$ be a set of contexts, and for each $c \in \mathcal{C}$ let $P^c$ denote the corresponding transition dynamics and $\pi_c^*$ an optimal policy achieving return $J_c(\pi) = \mathbb{E}_{\pi, P^c} \left[ \sum_{t=0}^{\infty} \gamma^t r(s_t, a_t) \right]$. We say that $\mathcal{C}$ is *non-overlapping* if there exists $\epsilon > 0$ (significant relative to the task scale) such that for every context-unaware policy $\pi$,

$$\max_{c \in \mathcal{C}} \big( J_c(\pi_c^*) - J_c(\pi) \big) \geq \epsilon.$$

That is, no single policy without access to context achieves near-optimal performance across all contexts. If this condition is violated, we call $\mathcal{C}$ *overlapping*.

**Definition 7 (Policy-Overlap).** Policy-Overlap (PO) quantifies whether a single context-unaware policy can achieve near-optimal performance across all contexts. Define the normalized worst-case relative performance of the best context-unaware policy as

$$\text{PO}(\mathcal{C}) := \max_{\pi \in \Pi_{\text{unaware}}} \min_{c \in \mathcal{C}} \frac{J_c(\pi) - J_c^{\min}}{J_c(\pi_c^*) - J_c^{\min}} \in [0, 1], \tag{6}$$

where $\Pi_{\text{unaware}}$ is the set of all policies $\pi(s)$ that do not condition on any context information (either explicit or inferred), and $J_c^{\min}$ is the minimum achievable per-context return.

$\text{PO} \approx 1$ indicates high policy overlap: a single context-free policy attains near-optimal performance across contexts. $\text{PO} \approx 0$ indicates low policy overlap: no single context-free policy performs well uniformly.

- High PO (*overlapping* contexts): Similar optimal actions across contexts (e.g., mass variations in DI). A single robust policy can handle all contexts effectively.
- Low PO (*non-overlapping* contexts): Drastically different optimal policies required (e.g., actuator inversion). Domain randomization (DR) fails as shown in Lemma 9, achieving $\text{PO} \approx 0$.

The PO measure directly captures the fundamental challenge that contextual policies must address: environments with low-PO require explicit context conditioning, while high-PO environments can be solved with robust control.

We explicitly use actuator inversion as the canonical way to create true non-overlapping contexts ($\text{PO} \approx 0$), because it induces a hard qualitative discontinuity (see Remark 2). Formally:

**Definition 8 (Actuator Inversion).** A context $c \in \{+1, -1\}$ defines actuator-inverted dynamics

$$P^c(s_{t+1} \mid s_t, a_t) = P(s_{t+1} \mid s_t, c \cdot a_t),$$

where $P$ is the nominal physical dynamics. The reward $r(s_t, a_t)$ is assumed strictly increasing in correctly directed actions.

Actuator inversion forces binary incompatibility in the transition dynamics, such that policies that succeed in one context fail in the other. Per Theorem 3, in inverted contexts the optimal policies differ multiplicatively, so no single concatenation-based policy can approximate both.

### A.2.1 CONTEXT-UNAWARE AGENT (E.G., DOMAIN RANDOMIZATION (DR))

- *Overlapping* contexts (**Solvable**)
  The policy class $\Pi_{\text{unaware}}$ is the set of all *context-unaware* policies that don't receive any context information as input, either explicit or inferred. If the contexts are overlapping, the optimal policies $\pi_c^*$ for different $c$ are similar. Therefore, a single policy $\pi \in \Pi_{\text{unaware}}$ can exist that is near-optimal for all $c \in \mathcal{C}_{\text{train}}$. The agent is effectively solving a single, slightly broader MDP.

- *Non-overlapping* contexts (**Theoretically unsolvable**)
  By Definition 6, for non-overlapping contexts, a single context-unaware policy cannot be optimal for all contexts. Formally, $\forall \pi \in \Pi_{\text{unaware}}, \exists c \in \mathcal{C}$ such that the performance $J^c(\pi)$ is arbitrarily poor. The agent is faced with a set of fundamentally different MDPs and is forced to learn a single, compromised policy that is mediocre everywhere.

**Lemma 9 (Failure of DR under actuator inversion).** Let contexts be $c \in \{+1, -1\}$ with dynamics $P^c(s' \mid s, a) = P(s' \mid s, c \cdot a)$. Assume the task satisfies the following: For every policy $\pi$ and its negation $-\pi$ there exists a constant $\Delta \geq 0$ such that $J_{+1}(\pi) + J_{+1}(-\pi) \leq 2\Delta$. Then the domain-randomized (DR) policy that maximizes expected return under $c \sim \text{Unif}\{+1, -1\}$ has average return at most $\Delta$. In particular, if $\Delta$ is small (e.g., negligible compared to per-context optima), DR fails to achieve non-trivial average return.

The assumption (the inequality with $\Delta$) formalizes the intuition that no single policy and its negation both achieve high reward in the nominal task. This is implied for many reach/goal tasks where action negation reverses progress.

*Proof.* For any fixed policy $\pi \in \Pi_{\text{unaware}}$,

$$J_{\text{DR}}(\pi) = \tfrac{1}{2}\big(J_{+1}(\pi) + J_{-1}(\pi)\big) = \tfrac{1}{2}\big(J_{+1}(\pi) + J_{+1}(-\pi)\big) \leq \tfrac{1}{2} \cdot 2\Delta = \Delta,$$

where we used $J_{-1}(\pi) = J_{+1}(-\pi)$ from the actuator-inversion symmetry and the assumption in the lemma. Maximizing over $\pi$ yields the stated bound. $\square$

**Remark 10 (Context-unaware policies are epistemic POMDP solvers).** When the policy is context-unaware (as in DR), the problem becomes an *epistemic POMDP* (Ghosh et al., 2021): the true state is $(s_t, c)$, but the agent only observes $s_t$ and must implicitly maintain a belief over the hidden context $c$. Thus, unknown contexts induce partial observability even when the raw state is fully observed. For overlapping contexts (e.g., small changes in mass or friction), the dynamics $P^c$ vary smoothly. Small belief errors lead to small prediction errors, so the induced belief-MDP remains easy to optimize. For non-overlapping contexts (e.g., actuator inversion $c = \pm 1$), the dynamics for different contexts are mutually incompatible. Even slight uncertainty over $c$ yields drastically different predictions for $s_{t+1}$. The effective mixture dynamics

$$\bar{P}(s_{t+1} \mid s_t, a_t) = \mathbb{E}_{c \sim \mathcal{C}_{\text{train}}}[P^c(s_{t+1} \mid s_t, a_t)],$$

become sharply multimodal. A context-unaware policy is therefore forced to average over contradictory behaviors, producing near-zero return. Maintaining a high-confidence belief under such conditions requires a sharp separation in the agent's internal representation, a representation that is both difficult to learn and highly sensitive to noise, leading to higher optimization variance and poorer generalization.

Providing the agent with an accurately inferred context signal $z_t = g_\phi(\tau_t^c)$ (as in DMA*-SH) sidesteps the epistemic POMDP problem: the policy can condition directly on the correct mode instead of hedging across incompatible ones. This explains the dramatic performance gap of DMA*-SH on non-overlapping benchmarks (Tables 1 and 8), while the advantage often disappears on overlapping benchmarks where even context-unaware baselines can succeed.

### A.2.2 CONTEXT-AWARE AGENT (E.G., CONCAT/DA BASELINES)

- *Overlapping* contexts (**Easy**)

  The *context-aware* policy class $\Pi_{\text{aware}} = \pi(a \mid s, c)$ depends on the ground-truth context $c$. Since the functions $\pi^*(s, c)$ are similar for different $c$, the agent can smoothly vary its behavior based on $c$. The complexity is effectively that of $|\mathcal{C}_{\text{train}}|$ separate policies, but shared structure across contexts can facilitate learning and enable generalization to $\mathcal{C}_{\text{eval,out}}$ via continuity.

- *Non-overlapping* contexts (**More difficult, but solvable**)

  The key challenge here is extrapolation and discontinuous function approximation. The optimal policy $\pi^*(s, c)$ may be a discontinuous function of $c$. For example, for actuator inversion, $\pi^*(s, -1) \approx -\pi^*(s, +1)$. A continuous function approximator (like an MLP) learning from $\mathcal{C}_{\text{train}}$ will have to learn this sharp transition. If $\mathcal{C}_{\text{train}}$ does not contain contexts on both "sides" of the discontinuity, generalization to $\mathcal{C}_{\text{eval,out}}$ will fail. Per Theorem 3, DA's hypernetworks provide a stronger inductive bias for modeling these discontinuities (multiplicative interactions) compared to Concat, giving it an advantage. A Concat agent learns $\pi(s, c)$ but may produce jerky actions near $c = 0$ boundaries. DA adapts parameters multiplicatively ($\omega = c \cdot \omega_{\text{base}}$), exactly capturing the flip for stable zero-shot performance (see Table 7).

### A.2.3 CONTEXT-INFERRED AGENT (OUR METHOD, DMA*-SH)

- *Overlapping* contexts (**Moderately difficult**)

  The policy class of context-inferred agents, $\Pi_{\text{inferred}} = \pi(a \mid s, z)$, is explicitly conditioned on the inferred context $z$. The agent must solve two coupled problems: (1) *Context inference*: infer $z$ from a window of past $K$ transitions $\tau = \{(s_k, a_k, \delta s_{k+1})\}$ via the encoder $g_\phi$, and (2) *Control*: learn the policy $\pi(s, z)$. Inference difficulty scales inversely with context distinguishability. Since the dynamics differ only mildly across contexts, the inferred representation $z$ may be noisy or weakly informative. However, the control problem is comparatively easier: small errors in $z$ induce only small policy deviations, so errors degrade performance smoothly.

- *Non-overlapping* contexts (**Very difficult**)

  This is the hardest setting. Non-overlapping contexts provide strong statistical signals for inference (high *Informativeness*, e.g., large $I(\tau; c)$), so in principle the encoder can recover $c$ from few transitions. In practice, however, even tiny inference errors are catastrophic: misclassifying $c = +1$ as $c = -1$ induces the *opposite* control law, and the agent immediately fails. The policy therefore cannot learn unless the encoder $g_\phi(\tau)$ is near-perfect. This creates a difficult credit-assignment loop during joint training.

  Encoder imprecision may arise from finite window size $K$ (partial observability), stochasticity in $P^c$ (e.g., sensor noise), or approximation limits of $g_\phi$. In non-overlapping regimes, such small errors are amplified severely in RL performance (e.g., through error propagation in value targets or large policy regret), since the failure modes are binary with no "graceful degradation." The brittleness is worse for concatenation-based baselines, which must learn hard boundaries in their inputs, whereas hypernetwork-conditioning (as in DMA*-SH) naturally captures the multiplicative structure of actuator inversion (Theorem 3).

**Remark 11 (The smoothness inductive bias of Latent Dynamic Models).** VariBAD (Zintgraf et al., 2020) is a meta-learning method for POMDPs that formulates context inference as a variational latent-variable model in which the agent maintains a belief distribution over latent environment parameters. Concretely, it optimizes an ELBO of the form

$$\mathcal{L}_{\text{ELBO}} = \mathbb{E}_{q_\phi(z|\tau)}[\log p_\theta(\tau \mid z)] - \text{KL}(q_\phi(z \mid \tau) \parallel p(z)),$$

where both the posterior $q_\phi(z \mid \tau)$ and the prior $p(z)$ are *unimodal* Gaussians. The KL term enforces proximity of $q_\phi(z \mid \tau)$ to $p(z)$, and therefore penalizes any multi-modal, discontinuous, or sign-flipped posterior geometry. Tasks with actuator inversion require a representation satisfying $z(c = +1) \approx -z(c = -1)$, which is *discontinuous* under any smooth prior. Any smooth prior necessarily *interpolates* between these modes. This forces the posterior $q_\phi(z \mid \tau)$ to place probability mass on latent values $z$ that correspond to no valid dynamics model at all, producing "averaged" latents lying between the $+1$ and $-1$ actuator modes. This mismatch yields catastrophic gradients: slight errors in $z_t$ produce policies that command the wrong action sign. This explains the empirical failures of VariBAD on ODE and DI-inversion tasks (see Figure 17).

In contrast, DMA*-SH completely avoids smooth latent priors by representing context through *multiplicative hypernetwork modulation*. Given a context embedding $z_t$, the hypernetwork generates adapter weights $\omega = h_\eta(z_t)$, allowing the policy and critic to implement discontinuous transformations such as

$$\pi_{\xi,\omega}(s) \approx -\pi_{\xi,\omega'}(s) \quad \text{when} \quad z_t \text{ crosses the inversion boundary.}$$

These multiplicative interactions supply the correct inductive bias for actuator inversion and other non-overlapping context regimes, enabling stable learning where ELBO-based methods like VariBAD fundamentally fail.

## A.3 DIRECTIONAL EFFECTS OF NORMALIZATION AND SHARED HYPERNETWORKS

This section analyzes how DMA*-SH's normalization strategy and shared hypernetwork architecture jointly shape the geometry of the learned context space. Although our empirical experiments (t-SNE projections in Appendix D and Representation-Overlap cosine analysis in Appendix A.4) directly measure only the qualitative expansion and compression patterns in the context encoder, we provide two hypotheses that explain these observations in terms of (i) approximate normalization-induced scale control and (ii) directional concentration induced by sharing a single hypernetwork across dynamics, policy, and value functions. These hypotheses are consistent with our architectural design and supported indirectly by the empirical evidence.

**Hypothesis 1: Directional encoding under approximate normalization-induced scale control.**

In DMA*-SH, approximate end-to-end scale invariance arises from the interaction between input normalization, SimNorm-based context normalization, and hypernetwork-driven weight generation. AvgL1Norm ensures that the raw transition inputs $(s_t, a_t, \delta s_{t+1})$ occupy a fixed-scale space, removing environment-specific magnitude variations. SimNorm projects $z_t$ onto the probability simplex, eliminating absolute scale and emphasizing relative components; while

softmax is not strictly scale-invariant, this projection induces an approximate focus on the directional structure of $z_t$. Because the adapter networks use ReLU activations between layers, the mapping $z_t \mapsto \omega_t = h_\eta(z_t)$ is positively homogeneous of degree 1 almost everywhere, i.e., $h_\eta(\alpha z_t) \approx \alpha\, h_\eta(z_t)$ up to normalization effects. After SimNorm removes the radial component, the hypernetwork output becomes sensitive primarily to the directional component $z_t/\|z_t\|$, yielding an *approximately direction-selective weight generator*. Consequently, the variation of adapter weights with respect to context is concentrated along the Jacobian directions

$$J_{h_\eta}(z_t) \;=\; \frac{\partial\, h_\eta(z_t)}{\partial z_t},$$

which describe how the shared hypernetwork modulates the main networks through a restricted, low-dimensional set of directions induced by $z_t$'s geometry on the simplex. This yields a principled form of *directional encoding*: overlapping contexts with similar directional embeddings generate adapter weights within the same low-dimensional modulation cone, while discontinuous contexts (e.g., actuator inversion) map to sharply separated directions, enabling the model to implement multiplicative sign flips or non-smooth transitions in downstream policies.

**Hypothesis 2: Shared hypernetworks amplify directional concentration.**

In DMA*-SH, the same hypernetwork $h_\eta$ is trained solely through the dynamics loss $L_{\phi,\theta,\eta}$, yet its outputs $\omega_t = h_\eta(z_t)$ parameterize not only the dynamics model but also the policy and Q-function. Because a single set of hypernetwork weights must simultaneously support accurate dynamics prediction and effective downstream control, the optimizer is pressured to encode context-discriminative information in stable, low-dimensional, functionally relevant *directions* of $\omega_t$ rather than in arbitrary magnitude variations. This induces a preference for directional semantics in the normalized hypernetwork outputs $\hat{\omega}_t = \omega_t/\|\omega_t\|$, and correspondingly encourages the encoder to suppress context dimensions that do not impact shared functional structure. This architectural coupling provides a natural explanation for our empirical findings: the encoder expands along the actuator-inversion axis, which is critical for control, while compressing along the redundant mass dimension. These patterns are consistent with a system where shared hypernetwork modulation reinforces directional concentration in representation space.

Our normalization strategy combined with hypernetwork conditioning creates an approximately scale-controlled representation space where functional modulation depends primarily on the *direction* of the context embedding. This results in hypernetwork-generated adapter weights that vary predominantly through directional differences in representation space, a phenomenon supported empirically by our t-SNE structure (Appendix D) and Representation-Overlap analyses (Appendix A.4).

## A.4 Variability, Representation-Overlap (RO), and Directional Geometry

We analyze the geometric structure of learned context representations through two complementary measures: Variability (representation spread), and Representation-Overlap (pairwise context similarity).

**Variability.** We relate the empirical "Variability" of inferred embeddings $z_t = g_\phi(\tau)$ (Section 5.5) to the local geometry induced by the context encoder.

Given contextual environments parameterized by ground-truth context $c = (c_1, c_2) \in \mathcal{C}$, the context encoder $g_\phi$ maps a trajectory window $\tau_t^c$ of length $K$ to a latent representation $z_t$. Variability is the average per-coordinate variance of these representations:

$$\text{Variability} = \frac{1}{d} \sum_{i=1}^{d} \text{Var}\big[z_i\big], \quad z = g_\phi(\tau_t^c) \in \mathbb{R}^d. \tag{7}$$

Variability measures how widely the encoder spreads context representations *across different ground-truth contexts* in a given dataset. High Variability indicates that $g_\phi$ maps trajectories from different contexts to widely separated points in representation space, reflecting *geometric expansion* of the context manifold. Low Variability indicates that the encoder maps many contexts to nearby

representations, reflecting *geometric compression*. This expresses how the learned representation space contracts or expands the underlying contextual variation present in the environment.

Conceptually, each ground-truth context $c$ corresponds to a distribution over trajectory windows $\tau_t^c$, and the encoder induces a map from context space to representation space,

$$c \mapsto z(c) := \mathbb{E}_{\tau \sim p(\tau|c)}[g_\phi(\tau)].$$

Consider two neighboring contexts $c$ and $c + \delta c$. A first-order approximation yields

$$z(c + \delta c) \approx z(c) + J_g(c)\, \delta c, \qquad \text{where } J_g(c) := \frac{\partial z}{\partial c}$$

is the Jacobian of the *context-to-representation* map, measuring local expansion or compression of the context manifold $\mathcal{C} \in \mathbb{R}^2$ under the encoder. However, since the encoder processes trajectories rather than contexts directly, we obtain a composite mapping $c \mapsto \tau^c \mapsto z$. The corresponding Jacobian decomposes as:

$$\frac{\partial z}{\partial c} = \underbrace{\frac{\partial z}{\partial \tau^c}}_{\text{encoder sensitivity}} \cdot \underbrace{\frac{\partial \tau^c}{\partial c}}_{\text{environment sensitivity}} \cdot$$

The first term measures sensitivity to trajectory-level perturbations for a fixed $c$. This is the mapping implemented by the encoder, and Variability is an empirical proxy for this sensitivity.

The operator norm $\|J_g(c)\|$ determines the local geometric behavior. $\|J_g(c)\| \gg 1$ implies local *expansion*: small changes in context produce large displacements in representation space; $\|J_g(c)\| \ll 1$ implies local *compression*: the encoder collapses variation in $c$ into a smaller region of representation space.

Our empirical Variability measure is computed over the set of contexts present in each dataset (train, eval-in, eval-out). Hence the empirical variance $\mathrm{Var}(Z)$ estimates the degree of global expansion or compression induced by the encoder when mapping *different contexts* to latent space. Although Variability is computed over representations $z = g_\phi(\tau)$ (with $\tau$ sampled from each context) rather than directly from contexts $c$, high Variability indicates that the encoder produces context-dependent variation of large magnitude (on average), consistent with a larger $\|J_g(c)\|$ in the regions of context space represented by the dataset. Conversely, low Variability corresponds to geometric compression of the context manifold, consistent with smaller Jacobian norms.

This geometric interpretation explains phenomena observed in contextual environments with mixed structure, such as actuator inversion plus mass variation (DI): the encoder learns to *expand* along the actuator-inversion axis, which is critical for control, while *compressing* along the mass dimension, which is redundant for control. Importantly, the encoder compresses only those context dimensions for which the optimal policies exhibit high overlap, thus producing a representation that appears "matched" to the control geometry of the environment. This pattern is reflected in both the t-SNE (Figure 8) and and cosine-similarity plots (Figure 9).

**Representation Overlap** (RO)**.** We introduce a formal notion of Representation-Overlap (RO). Intuitively, RO measures whether the context encoder $g_\phi$ maps different ground-truth contexts $c$ to nearby latent embeddings.

The combination of input normalization, SimNorm, and shared hypernetwork conditioning induces an approximate scale-controlled representation space (see Appendix A.3). Since SimNorm enforces exact scale invariance on $z_t$ and the hypernetwork $h_\eta$ receives only normalized embeddings, the adapter weights $\omega = h_\eta(z_t)$ depend primarily on the direction of $z_t$. Thus, embeddings that differ only by a positive scalar factor, $[z] = \{\alpha z : \alpha > 0\}$, belong to the same functional equivalence class: they generate approximately identical adapter parameters and therefore induce similar functional modulation of the dynamics, Q-function, and policy networks. Under this equivalence, the similarity geometry that matters for functional behavior lies on the unit sphere $S^{d-1} \subset \mathbb{R}^d$. Cosine similarity is therefore the correct metric for assessing representation geometry. It is invariant to all positive radial scalings,

$$\cos(\alpha u, \beta v) = \cos(u, v) \qquad \forall \alpha, \beta > 0,$$

and thus respects the equivalence classes induced by normalization and the hypernetwork architecture. Moreover, when the hypernetwork is direction-sensitive, embeddings with high cosine similarity generate adapter weights $\omega_i = h_\eta(z_i)$ and $\omega_j = h_\eta(z_j)$ that are close in function space. Cosine

similarity therefore provides a principled proxy for functional similarity between hypernetwork-generated adapters.

In DMA*-SH, the functional effect of a mean direction is what matters for adapters because, to first order, the shared hypernetwork maps directional differences in $z$ to directional differences in adapter weights (Appendix A.3). Therefore cosine similarity between context-means is a geometry-preserving proxy for functional similarity of the generated adapters. We use the cosine similarity between per-context mean embeddings to measure how the encoder arranges contexts in latent space.

**Definition 12 (Representation-Overlap (RO)).** Let $d$ be the embedding dimension and let $\{c^{(1)}, c^{(2)}, \ldots, c^{(n)}\}$ be $n$ distinct context values sampled from the 2D context space. For each context $c^{(i)}$, given a batch of $B$ representations $z_{c^{(i)}}^{(b)} \in \mathbb{R}^d$, define its mean embedding $\mu_{c^{(i)}} = \frac{1}{B} \sum_{b=1}^{B} z_{c^{(i)}}^{(b)}$. The pairwise cosine similarity between contexts $c^{(i)}$ and $c^{(j)}$ is

$$\cos(\mu_{c^{(i)}}, \mu_{c^{(j)}}) = \frac{\mu_{c^{(i)}}^{\top} \mu_{c^{(j)}}}{\|\mu_{c^{(i)}}\| \, \|\mu_{c^{(j)}}\|}. \tag{8}$$

The global Representation-Overlap (RO) score is the average cosine similarity over all $n^2$ context pairs (including self-similarities):

$$\mathrm{RO} = \frac{1}{n^2} \sum_{i=1}^{n} \sum_{j=1}^{n} \cos(\mu_{c^{(i)}}, \mu_{c^{(j)}}). \tag{9}$$

RO is invariant to radial rescaling of the means and therefore measures global directional alignment in representation space. Larger values of RO indicate greater alignment of per-context mean embeddings and therefore greater similarity in the hypernetwork-generated adapter functions. Figure 4 shows that DMA*-SH achieves higher RO (compared to the baselines DMA* and DMA) by aggressively compressing irrelevant mass variations, causing context representations to concentrate along shared directions. This observation perfectly matches the t-SNEs in Figure 8. The pairwise cosine similarity matrix (Figure 9) further reveals which context dimensions are compressed versus separated, consistent with the directional geometry induced by DMA*-SH (see Appendix D). These empirical signatures indicate that the encoder of DMA*-SH preferentially compresses those context dimensions that leave the optimal policy approximately invariant (e.g., mass in DI), while preserving those dimensions that induce distinct control laws (actuator inversion). In this sense, the learned representation appears "matched" to the control geometry of the environment, concentrating variation along behaviorally relevant axes while compressing irrelevant dimensions.

### A.5 IMPLICIT REGULARIZATION VIA SHARED HYPERNETWORK GRADIENTS

We explore the advantages of the shared hypernetwork design in DMA*-SH. By sharing a single hypernetwork $\eta$ across the dynamics model, policy, and Q-function, DMA*-SH ensures that RL gradients are *implicitly regularized* by physically meaningful dynamics gradients. This effect arises from the interleaved training loop in Algorithm 2, which alternates between the following two steps:

- RL updates:

$$\xi \leftarrow \xi - \alpha_1 \nabla_\xi \sum_c L_\xi^c, \qquad \zeta \leftarrow \zeta - \alpha_2 \nabla_\zeta \sum_c L_\zeta^c,$$

where $L_\xi^c$ and $L_\zeta^c$ are the actor and critic losses depending on $\pi_{\xi,\omega}$ and $Q_{\zeta,\omega}$, respectively, with $\omega = h_\eta(z_t)$.

- Dynamics updates:

$$\phi \leftarrow \phi - \alpha_3 \nabla_\phi \sum_c L_{\phi,\theta,\eta}^c, \qquad \theta \leftarrow \theta - \alpha_3 \nabla_\theta \sum_c L_{\phi,\theta,\eta}^c, \qquad \eta \leftarrow \eta - \alpha_3 \nabla_\eta \sum_c L_{\phi,\theta,\eta}^c,$$

where $L_{\phi,\theta,\eta}^c = \|\delta \hat{s}_{t+1} - \delta s_{t+1}\|_2$ depends on $f_{\theta,\omega}(s_t, a_t)$ with $\omega = h_\eta(z_t)$.

Since $\omega$ is detached in the RL losses, the gradients $\nabla_\xi L_\xi^c$ and $\nabla_\zeta L_\zeta^c$ do not propagate to $\eta$ (updates are only w.r.t. $\xi$ and $\zeta$). Nonetheless, the shared $\omega$ implicitly couples the objectives: during the

RL phase, $\xi$ and $\zeta$ are optimized under dynamics-aligned adapters, while the dynamics phase updates $\eta$ to better support the current RL components. This coupling reduces conflicting updates and stabilizes learning, providing a natural regularization that improves zero-shot generalization.

Intuitively, in a separate hypernetwork design (denoted DMA*-H), each module, dynamics $f_{\theta,\omega^f}$, policy $\pi_{\xi,\omega^\pi}$, and Q-function $Q_{\zeta,\omega^Q}$ has its own hypernetwork $\eta^f, \eta^\pi, \eta^Q$. Gradients from the forward dynamics (FD) loss $L_{\text{FD}} = \|\delta\hat{s}_{t+1} - \delta s_{t+1}\|_2$ update only $\eta^f$, while actor and critic losses ($L_\xi$ and $L_\zeta$, comprising $L_{\text{RL}}$) update $\eta^\pi$ and $\eta^Q$ independently. This decoupling can lead to conflicts: RL hypernetworks may prioritize short-term reward maximization, potentially learning adapters that exploit reward artifacts or ignore physical consistency, resulting in brittle policies that fail in non-overlapping contexts (e.g., actuator inversion in DI or Cartpole). In contrast, DMA*-SH's shared $\eta$ is optimized *exclusively* via $L_{\text{FD}}$, embedding dynamics-aligned features into $\omega$. The RL losses then optimize base parameters $\xi$ and $\zeta$ under this fixed (per step) but evolving $\omega$, implicitly regularizing RL toward behaviors that respect the inferred dynamics. This embeds a "physics prior" into RL optimization reducing gradient variance and fostering robust zero-shot generalization, as evidenced by our ablations (see Figure 15).

Mathematically, the coupling arises because $\nabla_\xi L_{\text{RL}}$ and $\nabla_\zeta L_{\text{RL}}$ depend directly on the shared $\omega$, which is tuned via $L_{\text{FD}}$. Thus, RL gradients are computed in a landscape constrained by dynamics accuracy, smoothing the objective and reducing variance compared to separate cases (where adapters evolve freely). To quantify this implicit regularization, consider how $L_{\text{RL}}$ depends on $\eta$ through $\omega$. By the chain rule,

$$\frac{\partial L_{\text{RL}}}{\partial \eta} = \frac{\partial L_{\text{RL}}}{\partial \omega} \cdot \frac{\partial \omega}{\partial \eta},$$

where $\frac{\partial \omega}{\partial \eta} = \frac{\partial h_\eta(z_t)}{\partial \eta}$. The term $\frac{\partial L_{\text{RL}}}{\partial \omega}$ chains through both the policy $\pi_{\xi,\omega}$ and the critic $Q_{\zeta,\omega}$. For instance, for the actor, the relevant component of this gradient is $-\frac{\partial}{\partial \omega}\mathbb{E}_{a_t \sim \pi_{\xi,\omega}}[Q_{\zeta,\omega} - \alpha \log \pi_{\xi,\omega}]$. Although $\frac{\partial L_{\text{RL}}}{\partial \omega}$ is not used to update $\eta$ (to avoid direct conflict with $L_{\text{FD}}$), it still exposes a latent "tug-of-war": RL would prefer to adjust $\eta$ for reward, but must instead adapt $\xi$ and $\zeta$ under a dynamics-aligned $\omega$, effectively regularizing against unphysical solutions.

To empirically validate this coupling, we analyze gradients during training across four contextualized environments: DI, DI-Friction, ODE, and Cartpole, using alignment metrics in hypernetwork space. For separate hypernetworks, we compute the cosine similarity $\text{Cos}(\nabla_{\eta^f} L_{\text{FD}}, \nabla_{\eta^\pi} L_\xi)$. For the shared case, we use a "shadow" computation that hypothetically enables gradient flow from $L_{\text{RL}}$ to $\eta$ during evaluation, yielding analogous terms $\text{Cos}(\nabla_\eta L_{\text{FD}}, \nabla_\eta L_\xi)$. Higher alignments and norms in these shadow gradients indicate that sharing amplifies meaningful interactions, enforcing a compromise that grounds RL in dynamics.

Figure 6 compares DMA*-SH (shared, red) against DMA*-H (separate, black), where dashed lines indicate shadow gradients. The trends are consistent across the four environments:

- (Panel 1) Gradient norm variance for context encoder ($g_\phi$): Separate hypernetworks induce uncoordinated pulls on $z_t$ from RL and dynamics objectives, leading to noisier $z_t$ updates and thus higher variance in $\nabla_\phi$. In contrast, the shared hypernetwork stabilizes $z_t$ via dynamics alignment.

- (Panel 2) Gradient norm variance for policy base parameters ($\xi$): Shared adapters constrain the policy optimization landscape to physically plausible regions, lowering variance in $\nabla_\xi$. Separate hypernetworks allow RL-specific adapters to overfit reward quirks, sustaining higher variance even late in training.

- (Panel 3) Mean gradient norm on policy hypernetwork $\eta^\pi$ (shadow for shared): In separate, direct optimization of $\eta^\pi$ by actor loss drives gradients towards zero, indicating rapid convergence to potentially suboptimal, unphysical minima. In shared, *persistent high shadow norms* reveal continual RL-dynamics tension: The actor "wishes" to tweak $\eta$ to make $\omega$ better for rewards, but it can't since $\eta$ is locked to physics. So RL settles by tweaking its own base parameters ($\xi, \zeta$) to match the given $\omega$. Since sharing enforces a compromise, this mismatch creates persistently larger norms, reflecting RL's push against physics-driven constraints. These norms increase as policy complexity rises, highlighting the implicit regularization at play.

- (Panel 4, 5) $\text{Cos}(\nabla_\eta L_d, \nabla_\eta L_\pi)$ (shadow $\nabla_\eta L_\pi$ for shared): Shared case shows markedly higher alignment, with raw cosines exhibiting larger signed fluctuations. This reflects *strong coupling* in

the unified $\eta$-space: hypothetical actor gradients meaningfully interact (sometimes oppose, sometimes align) with actual dynamics gradients, enforcing physically consistent adaptation. In separate, independent parameter spaces produce near-zero alignment, with only minor noise-driven fluctuations.

- (Panel 6, 7) $\text{Cos}(\nabla_\eta L_\pi, \nabla_\eta L_Q)$ (shadow for both in shared): Actor and critic objectives are naturally aligned as both are reward-driven. Sharing amplifies this synergy in a common $\eta$-space, whereas separate hypernetworks permit divergence and decoherence, highlighting the consistency benefit of parameter tying. Separate allows divergence, keeping cosines very low.

- (Panel 8) Returns: Shared hypernetwork yields faster rise and higher returns. Dynamics-aligned adapters regularize RL toward generalizable, physically plausible behaviors; separate hypernetworks suffer gradient conflicts, delaying convergence and reducing asymptotic performance.

In summary, sharing a single dynamics-trained hypernetwork implicitly regularizes RL gradients with physical consistency. This reduces variance, preserves informative signals, and aligns objectives, yielding more stable training and superior zero-shot generalization compared to separate, uncoordinated hypernetworks. See Figure 15.

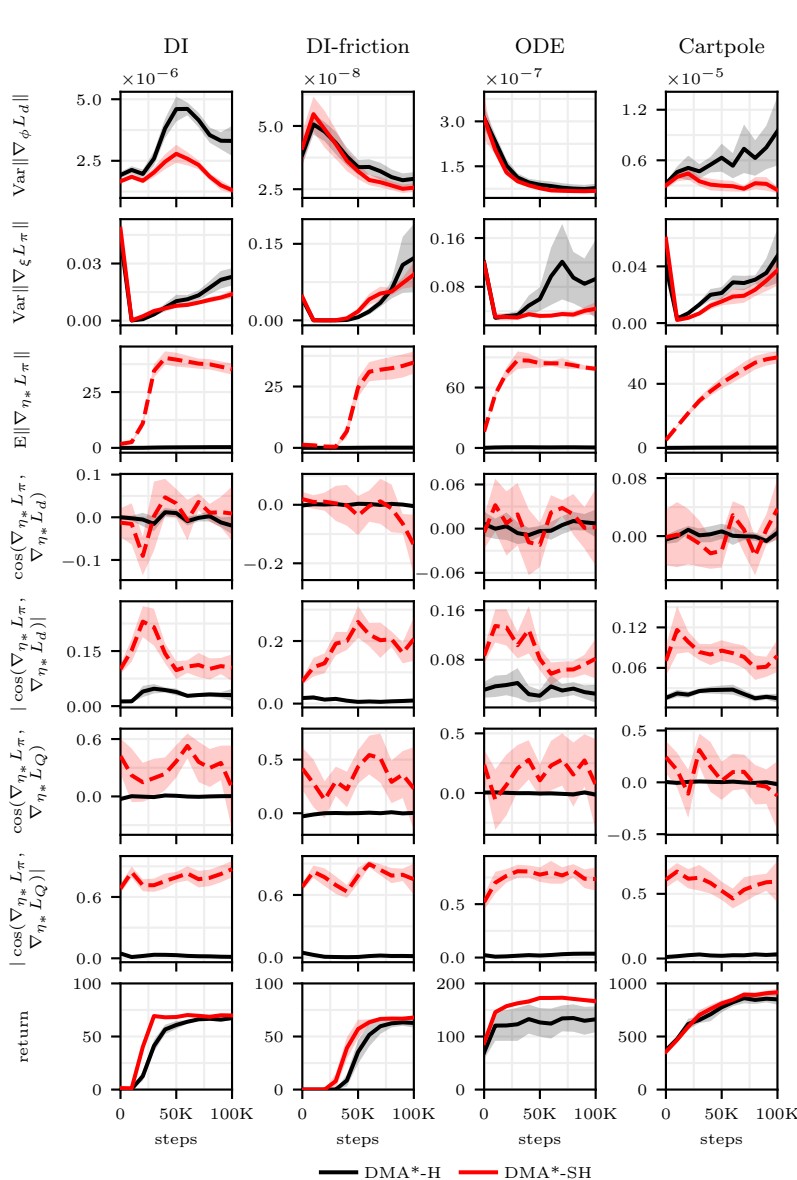

Figure 6: Gradient analysis comparing shared hypernetworks (DMA*-SH, red) vs. separate hypernetworks (DMA*-H, black) across DI, DI-Friction, ODE, and Cartpole environments. Dashed lines indicate gradients computed via a shadow graph (e.g., $\nabla_\eta L_\pi$ in shared, where $\eta$ is not updated by the policy loss $L_\pi$ during training). This enables hypothetical gradient evaluation without altering the training loop. Here, $\eta_*$ denotes the relevant hypernetwork parameters: in shared, the single $\eta$ (optimized solely via dynamics loss $L_d$); in separate, the module-specific hypernetworks (e.g., $\eta^\pi$ for policy gradients). $L_d$, $L_\pi$ and $L_Q$ correspond resp. to $L_{\text{FD}}$, $L_\xi$ (actor) and $L_\zeta$ (Q-function).

# B  ALGORITHMS

**Algorithm 1** Training loop DMA/DMA*

**Require:** Context set $\mathcal{C}_{\text{train}} = \{c_i\}_{i=1\ldots n_c}$ sampled from context range for training, learning rates $\alpha_1, \alpha_2, \alpha_3$

1: Init. replay buffers $\mathcal{B}^c$ for each context
2: Init. context windows (deque) $\tau^c_{t=0} = \{(s_{t-k}, a_{t-k}, \delta s_{t+1-k})\}_{k:1\ldots K} \sim \pi_{random}$ for each context
3: **for** step in training steps **do**
4:    *// Collect data in environment*
5:    **for** $c$ in $\mathcal{C}_{\text{train}}$ **do**
6:       Encode past transitions $z_t = g_\phi(\tau^c_t)$
7:
8:       Gather data from environment interaction with $a_t \sim \pi_\xi(\cdot|s_t, z_t)$
9:       Add data to $\mathcal{B}^c$ and update $\tau^c_t$
10:   **end for**
11:   *// Training*
12:   **for** $c$ in $\mathcal{C}_{\text{train}}$ **do**
13:      Sample RL batch $b^c \sim \mathcal{B}^c$ with corresponding context windows $\tau^c_t$
14:      Encode past transitions $z_t = g_\phi(\tau^c_t)$
15:
16:      Predict $\delta\hat{s}_{t+1} = f_\theta(s_t, a_t, z_t)$
17:      $L^c_\xi = L_\xi(\pi_\xi, b^c, z_t)$
18:      $L^c_\zeta = L_\zeta(Q_\zeta, b^c, z_t)$
19:      $L^c_{\phi,\theta} = \|\delta\hat{s}_{t+1} - \delta s_{t+1}\|_2$
20:   **end for**
21:   $\xi \leftarrow \xi - \alpha_1 \nabla_\xi \sum_c L^c_\xi$
22:   $\zeta \leftarrow \zeta - \alpha_2 \nabla_\zeta \sum_c L^c_\zeta$
23:   $\phi \leftarrow \phi - \alpha_3 \nabla_\phi \sum_c L^c_{\phi,\theta}$
24:   $\theta \leftarrow \theta - \alpha_3 \nabla_\theta \sum_c L^c_{\phi,\theta}$
25:
26: **end for**

**Algorithm 2** Training loop DMA*-SH

**Require:** Context set $\mathcal{C}_{\text{train}} = \{c_i\}_{i=1\ldots n_c}$ sampled from context range for training, learning rates $\alpha_1, \alpha_2, \alpha_3$

1: Init. replay buffers $\mathcal{B}^c$ for each context
2: Init. context windows (deque) $\tau^c_{t=0} = \{(s_{t-k}, a_{t-k}, \delta s_{t+1-k})\}_{k:1\ldots K} \sim \pi_{random}$ for each context
3: **for** step in training steps **do**
4:    *// Collect data in environment*
5:    **for** $c$ in $\mathcal{C}_{\text{train}}$ **do**
6:       Encode past transitions $z_t = g_\phi(\tau^c_t)$
7:       Compute hyperweights $\omega = h_\eta(z_t)$
8:       Gather data from environment interaction with $a_t \sim \pi_{\xi,\omega}(\cdot|s_t)$
9:       Add data to $\mathcal{B}^c$ and update $\tau^c_t$
10:   **end for**
11:   *// Training*
12:   **for** $c$ in $\mathcal{C}_{\text{train}}$ **do**
13:      Sample RL batch $b^c \sim \mathcal{B}^c$ with corresponding context windows $\tau^c_t$
14:      Encode past transitions $z_t = g_\phi(\tau^c_t)$
15:      Compute hyperweights $\omega = h_\eta(z_t)$
16:      Predict $\delta\hat{s}_{t+1} = f_{\theta,\omega}(s_t, a_t)$
17:      $L^c_\xi = L_\xi(\pi_{\xi,\omega}, b^c)$
18:      $L^c_\zeta = L_\zeta(Q_{\zeta,\omega}, b^c)$
19:      $L^c_{\phi,\theta,\eta} = \|\delta\hat{s}_{t+1} - \delta s_{t+1}\|_2$
20:   **end for**
21:   $\xi \leftarrow \xi - \alpha_1 \nabla_\xi \sum_c L^c_\xi$
22:   $\zeta \leftarrow \zeta - \alpha_2 \nabla_\zeta \sum_c L^c_\zeta$
23:   $\phi \leftarrow \phi - \alpha_3 \nabla_\phi \sum_c L^c_{\phi,\theta,\eta}$
24:   $\theta \leftarrow \theta - \alpha_3 \nabla_\theta \sum_c L^c_{\phi,\theta,\eta}$
25:   $\eta \leftarrow \eta - \alpha_3 \nabla_\eta \sum_c L^c_{\phi,\theta,\eta}$
26: **end for**

## C HYPERPARAMETERS AND IMPLEMENTATION DETAILS

Table 2 provides an overview of the used hyperparameters of the SAC agent, the context encoder, the dynamic model and the hypernetwork. We did not perform any tuning for SAC and kept hyperparameters standard as provided in CleanRL (Huang et al., 2022).

At the core of the context encoder, we use an LSTM layer whose final hidden state serves as the context representation $z_t$. This follows prior work employing MLPs, RNNs, or Transformer encoder layers, with minor architectural modifications, as context encoders (Rakelly et al., 2019; Evans et al., 2022). Prior work (Rakelly et al., 2019; Evans et al., 2022) also highlighted the benefit of processing the transitions in $\tau_t^c$ in random order, so that the latent state of the context encoder does not encode the temporal structure of $\tau_t^c$; we adopt this important idea. Moreover, we found that using only a fraction of the transitions within the context window is beneficial. The context window size $K$ depends on the environment: tasks derived from the DM Control Suite require a larger $K$ than others. The context encoder then samples a random fraction of the $K$ transitions as input; we use a relatively small fraction of $20\%$. For example, in the DM Control Suite, the context encoder observes only $128 \times 0.2 \approx 25$ transitions as input $\tau_t^c$.

For our hypernetworks, we use the framework of von Oswald et al. (2020). The adapter introduces a bottleneck, and importantly, we do not apply an activation function before it. Our design also allows the adapter to be bypassed via a skip connection. The design choices regarding the hypernetworks and adapters match those in DA (Beukman et al., 2023), where the placement of activation functions is likewise crucial. We reimplemented DA and verified that its performance is comparable to the original implementation.

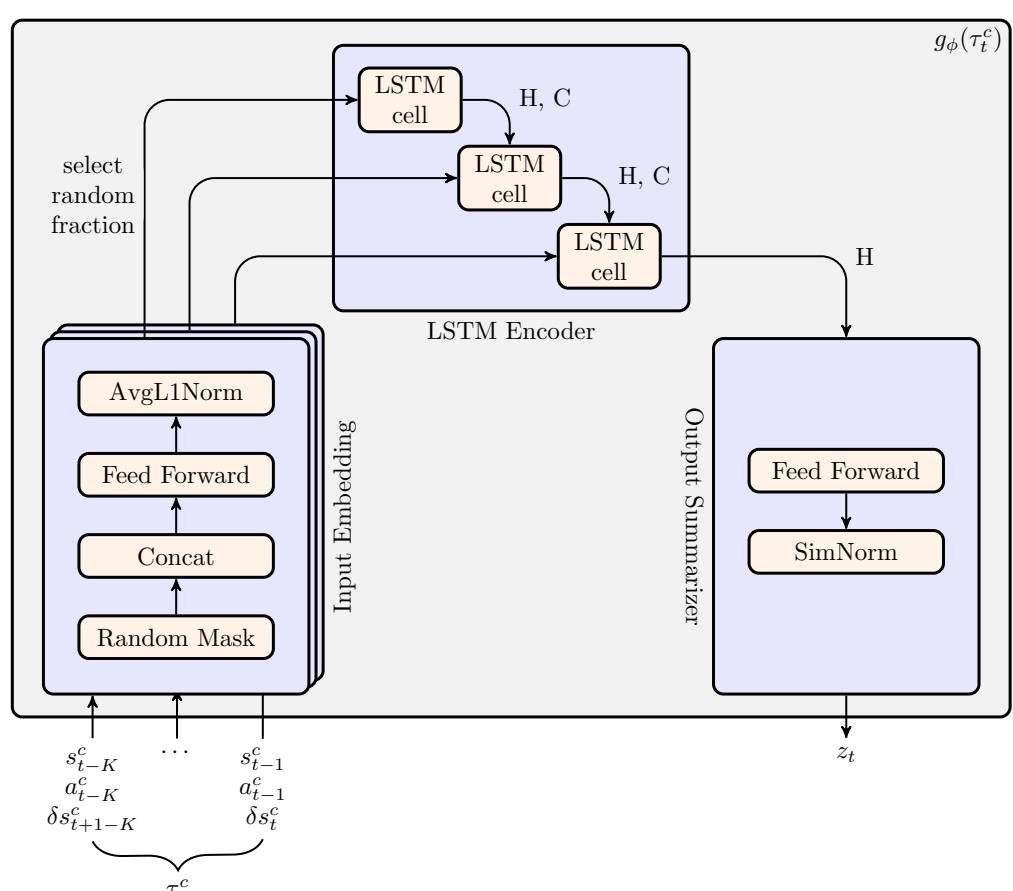

Figure 7: Architecture of the context encoder.

| Module | Name | Value |
|---|---|---|
| SAC | Buffer capacity | 1 000 000 |
| | Batch size | 256 |
| | Discount $\gamma$ | 0.99 |
| | Optimizer | Adam |
| | Critic LR | 0.0003 |
| | Actor LR | 0.0003 |
| | Temperature LR | 0.0003 |
| | Critic soft target update $\tau$ | 0.005 |
| | Init temperature (SAC) | 1.0 |
| | Init temperature (DrQ) | 0.1 |
| | Hidden dims | (256, 256) |
| | Activation function | ReLU |
| Context encoder | LR | 0.0003 |
| | Model dim | 32 |
| | Dropout | 0.1 |
| | Context dim | 8 |
| | Context window size $K$ (general) | 24 |
| | Context window size $K$ (DMC environments) | 128 |
| | Context window fraction | 0.2 |
| | Context encoder type | LSTM |
| | Activation function | ReLU |
| Dynamic model | LR | 0.0003 |
| | Hidden dims | (256, 256) |
| | Activation function | ReLU |
| Hypernetwork | LR | 0.0003 |
| | Hidden dims | (64, 64) |
| | Activation function | ReLU |
| Adapter | Bottleneck | 32 |
| | Skip connection | True |
| | Pre adapter activation function | None |
| | Post adapter activation function | ReLU |

Table 2: Hyperparameters.

# D  REPRESENTATION-OVERLAP: t-SNE VISUALIZATION AND COSINE SIMILARITY ANALYSIS

To complement our quantitative analysis of Variability and Informativeness, we visualize the geometry of inferred context embeddings $z_t$ using t-SNE (Van der Maaten & Hinton, 2008) and Representation Overlap (RO) via cosine similarity (see Definition 12). These methods provide complementary geometric perspectives: t-SNE reveals *local* cluster structure and neighborhood relationships within the context manifold, while cosine similarity quantifies *global* angular separation between context classes. Together, they characterize both the fine-grained organization and overall geometric alignment of learned representations.

Figure 8 presents t-SNE plots for the DI environment (non-overlapping) comparing DMA*-SH with the baselines DMA* and DMA. Each dot represents $z_t = g_\phi(\tau_t^c)$ and is colored according to the context $(c_1, c_2)$, where $c_1$ denotes mass and $c_2$ the action_factor ($\pm 1$).

Two distinct clusters emerge along the action_factor dimension across all methods: one for $c_2 = +1$ (left) and one for $c_2 = -1$ (right). Within each cluster, variations in mass ($c_1$) overlap more for DMA*-SH than for DMA or DMA*, which display more clearly separated blobs for different masses. Despite this reduced separability along the mass axis, DMA*-SH attains the highest RL performance. This is consistent with the fact that small mass differences yield largely overlapping optimal policies in DI, making fine-grained separation along this dimension less important. By contrast, accurate separation along the action_factor dimension is crucial due to the non-overlapping nature of the policies, and DMA*-SH preserves this separation effectively.

The t-SNE plot sheds light on why DMA*-SH achieves the lowest Variability: it compresses the overlapping context dimension (mass) while sharply separating the actuator dimension (action_factor), producing exactly the representation structure required for reliable zero-shot adaptation. Reduced Variability ensures that the policy receives stable, consistent context signals, explaining DMA*-SH's superior performance even when $I(z_t; c)$ is lower. Highly informative embeddings can fail if they fluctuate across trajectories: a misaligned $z_t$ may induce the opposite control law. These visualizations reinforce the quantitative results in Figure 4 and highlight that low Variability is more critical than maximal Informativeness.

We complement the t-SNE analysis with pairwise cosine similarities between context representations, averaged per context (see Definition 12), using the same contextualized DI environment.

From Figure 9, we observe:

**Impact of Input/Output Normalization.** Comparing DMA* (with normalization) to DMA (without) reveals that normalization:

- *Prevents artificial antagonisms*: DMA produces extreme negative cosines ($-0.4$ to $-0.97$) between different actuator modes, indicating pathological over-separation, while DMA* maintains near-orthogonal ($0.01$–$0.03$) relationships.

- *Enables stable gradient dynamics*: Extreme negative cosines in DMA create conflicting gradient directions during training, whereas DMA*'s near-zero cosines provide stable, consistent learning signals.

- *Controls representation scale*: Without normalization, DMA's encoder learns arbitrarily scaled representations that exaggerate geometric distortions. Normalization bounds the representation space, preventing these instabilities.

Normalizations thus shapes the emergent geometric structure by preventing pathological scale-driven distortions.

**Impact of Shared Hypernetworks.** Comparing DMA*-SH (with shared hypernetwork) to DMA* (without) demonstrates that hypernetworks:

- *Enhance within-mode compression*: DMA*-SH achieves perfect within-mode alignment ($\cos = 1.0$) versus DMA*'s imperfect compression ($\cos = 0.53$–$1.0$).

- *Optimizes mode discrimination*: DMA*-SH tunes inter-cluster distance ($0.14$–$0.23$) to enable reliable context identification while avoiding the gradient conflicts that arise from extreme separation.

- *Enforce directional encoding*: Sharing the hypernetwork forces the same $h_\eta$ to generate adapters that must work for dynamics, policy, and action-value networks simultaneously. This creates pressure for *directional concentration*: contextual differences are encoded in directions that are both functionally meaningful and stable. Empirically this boosts intra-cluster alignment means and reduces within-cluster variance. The modest positive inter-cluster cosines indicate the encoder keeps the actuator-separation functionally useful while avoiding pure sign-opposition that would amplify sensitivity.

The shared hypernetwork in DMA*-SH leverages the normalized scale space created by DMA* to concentrate semantic information in directional components, achieving the observed ideal cosine similarity structure. This shared design thus acts as a *geometric regularizer* that aligns the representation space with the true functional requirements of the task.

In summary, the progression DMA $\to$ DMA* $\to$ DMA*-SH observed in cosine matrices and t-SNE visualizations demonstrates that: (i) input/output normalization eliminates harmful scale effects and pathological geometry, while (ii) shared hypernetwork conditioning concentrates contextual distinctions into functionally meaningful directional axes. Together, these components yield the representation geometry beneficial for zero-shot robustness.

For an extended discussion, see Appendix A.4.

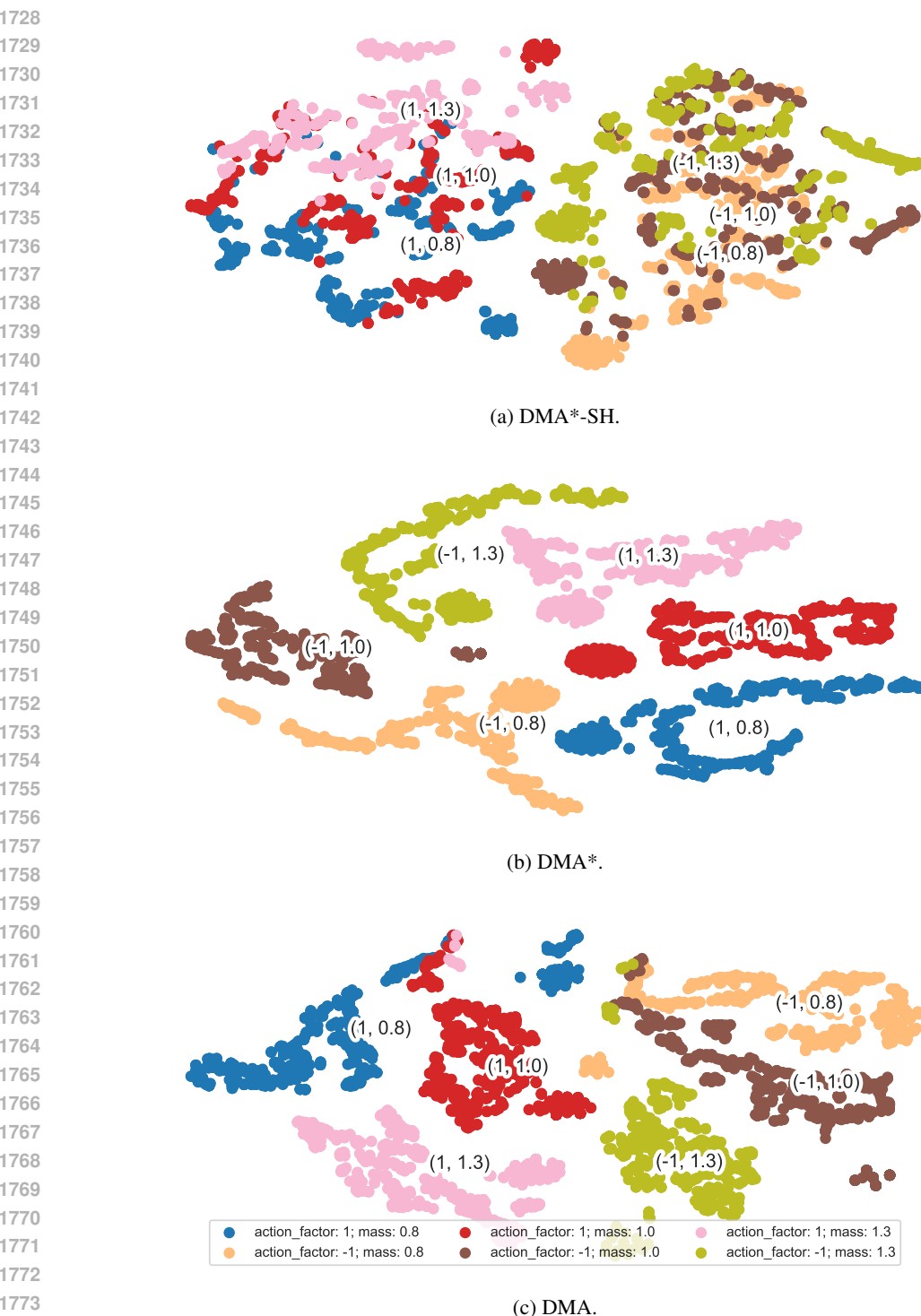

(a) DMA*-SH.

(b) DMA*.

(c) DMA.

Figure 8: t-SNE visualization of inferred context embeddings $z_t$ in the DI environment (non-overlapping) for DMA*-SH, DMA*, and DMA. Contextualization is handcrafted with lesser context instances for better clarity. DI is contextualized with mass and an action_factor of either $-1$ or $1$. Each dot corresponds to $z_t$ for a trajectory $\tau_t^c$ and is colored by $(c_1, c_2)$ with $c_1 =$ mass and $c_2 =$ action_factor ($\pm 1$). DMA*-SH achieves reduced Variability while preserving critical separability along the action_factor dimension. Mass clusters overlap more for DMA*-SH than for the baselines, yet episodic returns are higher, consistent with the mass dimension having largely overlapping policy effects.

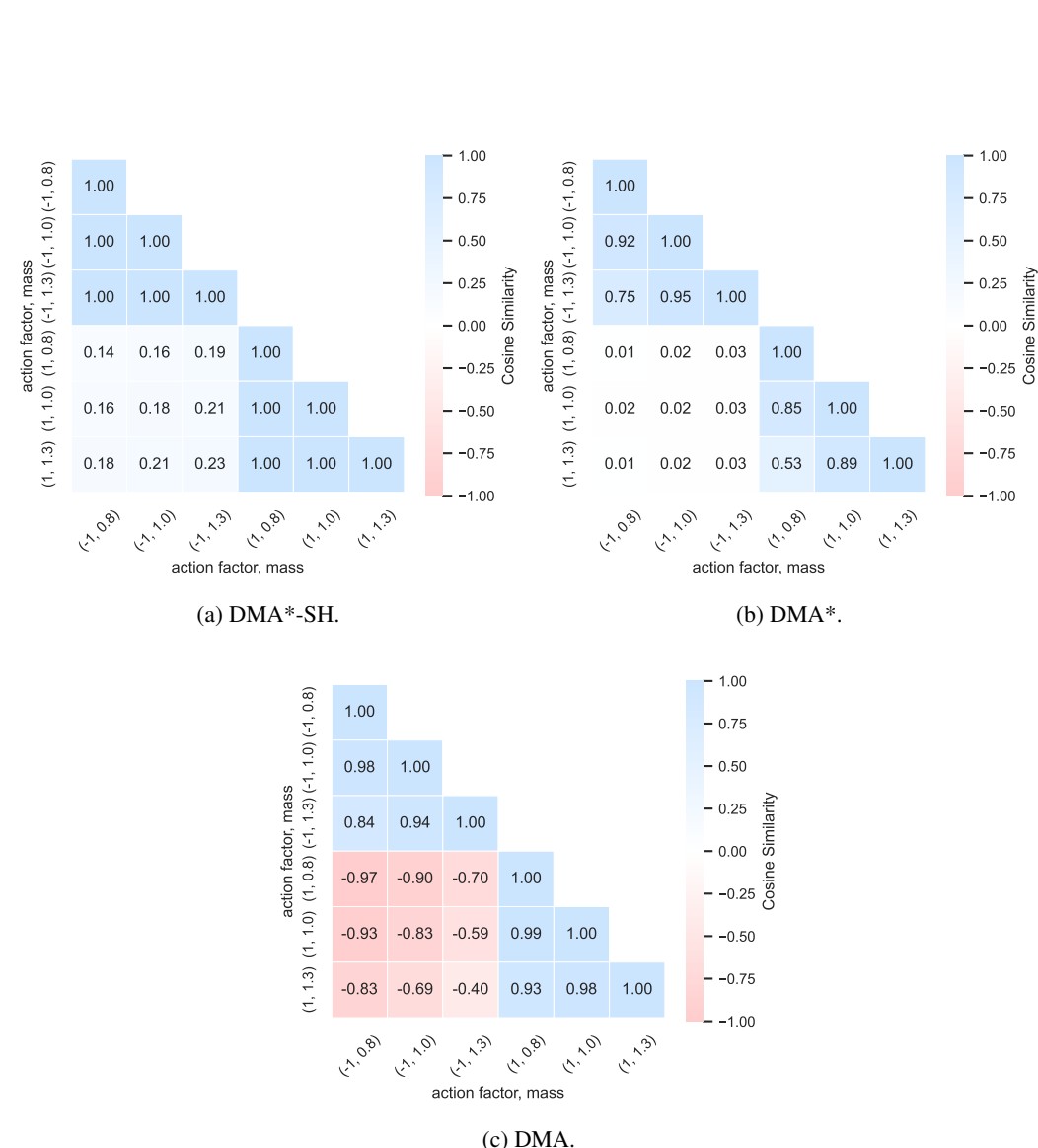

(a) DMA*-SH.

(b) DMA*.

(c) DMA.

Figure 9: Pairwise cosine similarity heatmap of inferred context embeddings $z_t$ averaged per context in the DI environment (non-overlapping) for DMA*-SH, DMA* and DMA. Contextualization is handcrafted with lesser context instances for better clarity. DI is contextualized with mass and an action_factor of either $-1$ or $1$. For a context pair $c$ and $s$, each value in the heatmap corresponds to the pairwise cosine similarity of averaged context representations, computed as in equation 8. DMA*-SH shows high similarity for overlapping contexts while preserving dissimilarity along the action_factor dimension. Episodic returns are higher compared to DMA* and DMA, consistent with the mass dimension having largely overlapping policy effects.

## E    Ablations for the Design Choices

We perform a range of ablations on which we base the design choices in Section 4.1. Figure 10 for DMA* and DMA*-SH show probability of improvements as suggested by Agarwal et al. (2021). They only show if there is a likely improvement using our choices compared to the alternatives. They do not necessarily tell us something about the magnitude. In Figure 3 we compare the vanilla DMA to DMA* and DMA*-SH, indicating that our design choices cumulatively have significant impact.

In Figure 11 we compare IQM scores (Agarwal et al., 2021) for different ratios of the random input masking of actions, states, and next state differences in $\tau_t^c$, resulting in a ratio of $20\%$ to be beneficial for DMA* and a ratio of $40\%$ to be beneficial for DMA*-SH. Especially for the latter, significant performance drops only occur at quite high masking ratios indicating robustness to varying input trajectories $\tau_t^c$.

In Figures 12 and 13 we compare different normalization attempts for the input and the output of the context encoder. The intuition about AvgL1Norm and SimNorm provided in Section 4.1 and in the literature (Fujimoto et al., 2023; Lavoie et al., 2023; Hansen et al., 2024) is also reflected in the performance. Our dynamics-alignment loss equation 2 encourages the encoder to organize $z_t$ according to *relative* differences between contexts, not absolute magnitudes. This is motivated from a scale-invariance perspective. Figure 14 justifies the choice of window size $K = 24$ for DI, DI-friction, and ODE, and $K = 128$ for the DMC-based environments. In terms of performance, the impact of $K$ appears to be minimal, provided that a sufficiently large minimum window size is used.

The DMA loss equation 2 operates on state differences. This encourages the encoder to capture relationships between contexts rather than absolute state values. The combination of DMA loss on state differences, input/output normalization, and hypernetwork properties collectively encourage the encoder to organize representations around relative context differences in a scale-invariant manner. See Appendix A.3 for a discussion.

The ablations indicate, that proper normalization is vital for dynamic model-aligned context encoders for zero-shot generalization in contextual RL. Input masking can improve performance even further. This can also be observed in Figure 15 for DMA*-SH. Further, it clearly indicates that the use of a shared hypernetwork outperforms an architecture that uses separate hypernetworks for dynamic model, policy and Q-value function.

Pearl (Rakelly et al., 2019) is used as a baseline to compare our proposed DMA* and DMA*-SH. To bring Pearl into the perspective of contextual RL, originally, (Rakelly et al., 2019) used Pearl for reward variations in an environment. In this case, it seems indicated to update the parameters of the context encoder using gradients from the Bellman updates for the Q-function. To make a fair comparison to our work, we align Pearl with a dynamics model, which we denote as DMA-Pearl. A comparison of Pearl and DMA-Pearl is provided in Figure 16a. Furthermore, Figure 16b shows the performances for different $\beta$ to weight the KL regularization in Pearl. DMA-Pearl demonstrates improved performance over vanilla DMA (see Table 1), highlighting the benefits of the probabilistic context encoder and KL regularization. However, integrating these design elements into DMA* and DMA*-SH does not yield further gains (see Figure 16a).

We also conducted experiments with VariBAD (Zintgraf et al., 2020), testing two different KL-weights $\beta$. While it achieves comparable performance in the overlapping DI–friction setting, it struggles considerably with the non-overlapping contextualizations in DI and ODE. For this reason, and given that we already include DMA-Pearl in our comparisons, we exclude VariBAD as a baseline in the remainder of the paper (cf. Section 5.2).

Methods built on smooth latent dynamics priors such as VariBAD and DMA-Pearl struggle as their objectives explicitly encourage latent embeddings to vary continuously with respect to context trajectories. This inductive bias is incompatible with tasks whose true context-to-dynamics map exhibits genuine discontinuities (as in DI with actuator inversion), where the correct representation requires a sign flip rather than a smooth interpolation (see Remark 11). In contrast, DMA*-SH succeeds by *structurally embedding* this discontinuity into the hypernetwork modulation pathway: multiplicative weight generation allows sharp directional changes in the induced policy/critic without incurring any ELBO or latent-prior penalty. Consequently, DMA*-SH avoids the continuity bias

inherent to latent-prior methods and implements the correct functional geometry for discontinuous contextual RL.

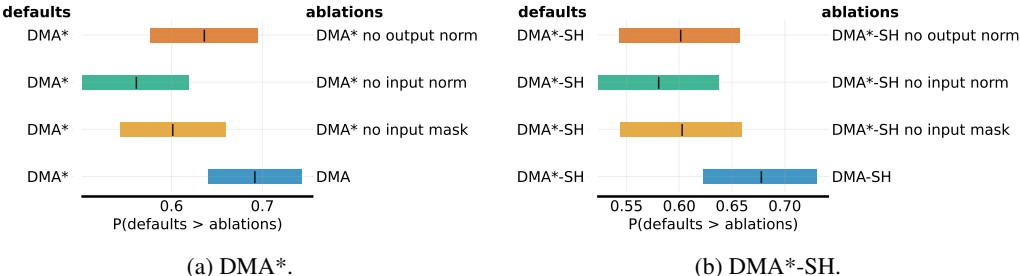

(a) DMA*.       (b) DMA*-SH.

Figure 10: Probability of improvement (POI) (Agarwal et al., 2021) based on AER scores (cf. Section 5.1) aggregated over the contextualized environments (cf. Section 5.3) and over contexts in the three context sets $\mathcal{C}_{\text{train}}$, $\mathcal{C}_{\text{eval,in}}$ and $\mathcal{C}_{\text{eval,out}}$. For the proposed DMA* and DMA*-SH, we ablate separately the random masking, input and output normalization, or everything at once (DMA, DMA-SH).

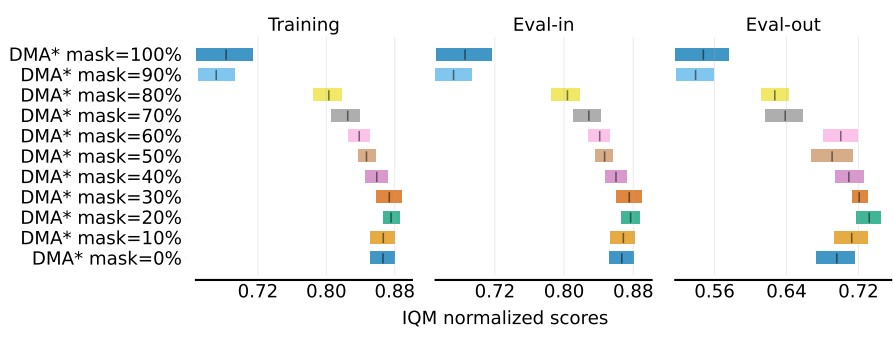

(a) DMA*.

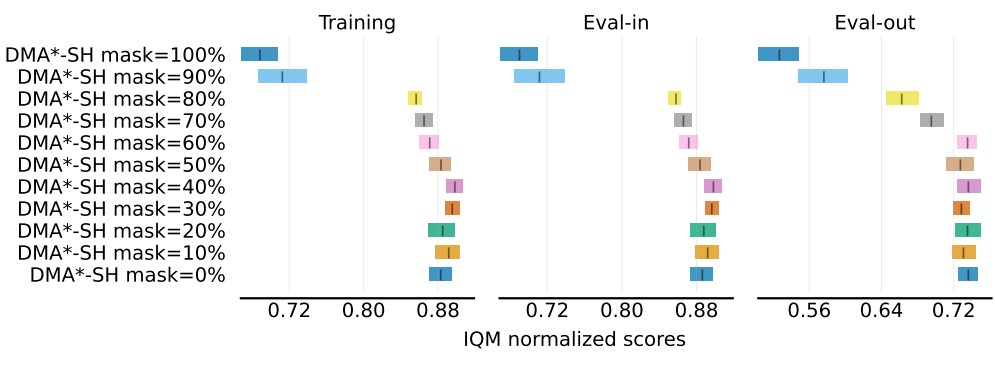

(b) DMA*-SH.

Figure 11: Interquartile mean (IQM) (Agarwal et al., 2021) based on AER scores (cf. Section 5.1) aggregated over the contextualized environments (cf. Section 5.3). We distinguish results for contexts in the three context sets $\mathcal{C}_{\text{train}}$, $\mathcal{C}_{\text{eval,in}}$ and $\mathcal{C}_{\text{eval,out}}$. We compare different ratios for the random input masking. When averaging over the three context sets, best performance is achieved using a ratio of $20\%$ for DMA* and $40\%$ for DMA*-SH.

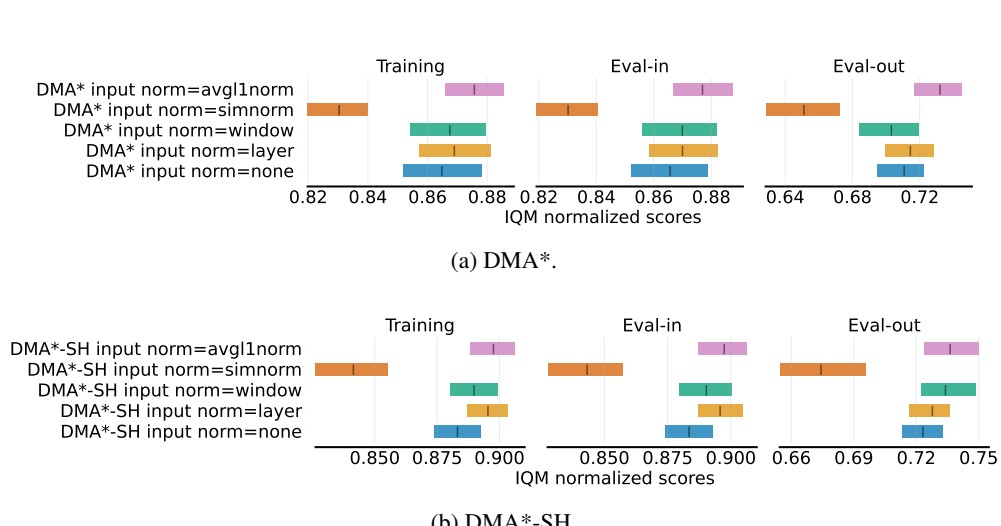

Figure 12: Interquartile mean (IQM) (Agarwal et al., 2021) based on AER scores (cf. Section 5.1) aggregated over the contextualized environments (cf. Section 5.3). We distinguish results for contexts in the three context sets $\mathcal{C}_{\text{train}}$, $\mathcal{C}_{\text{eval,in}}$ and $\mathcal{C}_{\text{eval,out}}$. We compare different types of input normalization. When averaging over the three context sets, best performance is achieved using AvgL1Norm in both DMA* and DMA*-SH.

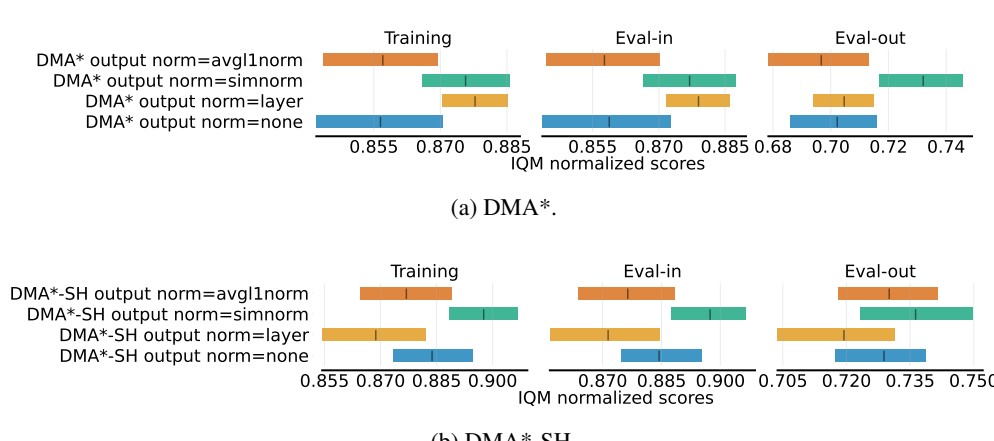

Figure 13: Interquartile mean (IQM) (Agarwal et al., 2021) based on AER scores (cf. Section 5.1) aggregated over the contextualized environments (cf. Section 5.3). We distinguish results for contexts in the three context sets $\mathcal{C}_{\text{train}}$, $\mathcal{C}_{\text{eval,in}}$ and $\mathcal{C}_{\text{eval,out}}$. We compare different types of output normalization. When averaging over the three context sets, best performance is achieved using SimNorm in both DMA* and DMA*-SH.

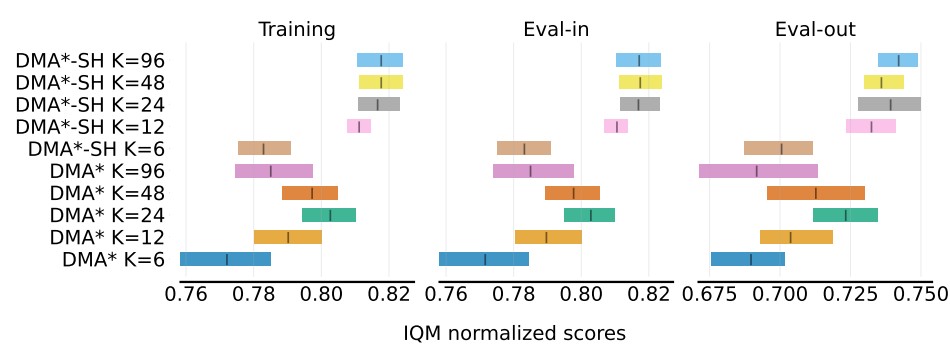

(a) For DI, DI-friction and ODE with shorter context window.

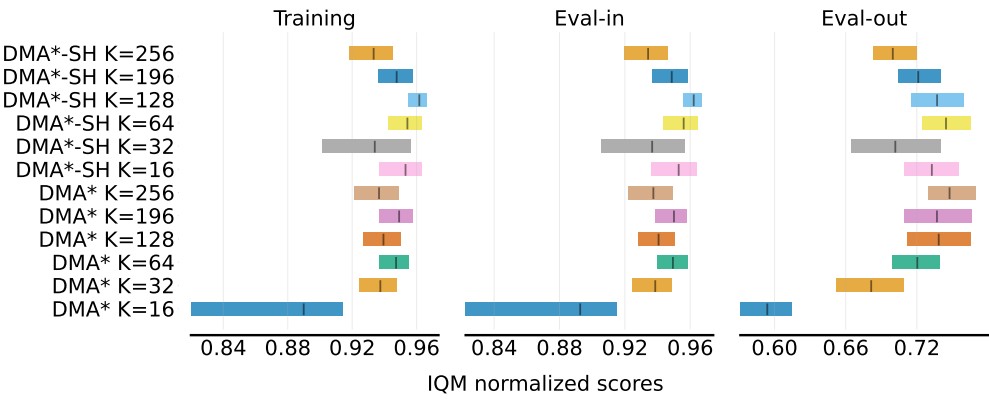

(b) For Cartpole, BallInCup and Walker with longer context window.

Figure 14: Interquartile mean (IQM) comparing different context window sizes justifying the choice of 24 for DI and ODE environments and 128 for DMC-based environments.

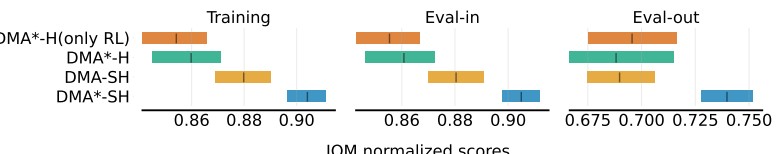

Figure 15: Interquartile mean (IQM) (Agarwal et al., 2021) based on AER scores (cf. Section 5.1) aggregated across the contextualized environments (cf. Section 5.3). We distinguish results for contexts in the three context sets $\mathcal{C}_{train}$, $\mathcal{C}_{eval,in}$, and $\mathcal{C}_{eval,out}$. We compare DMA*-SH to a variant without normalization and masking (DMA-SH) and to an architecture that does not share the hypernetwork (DMA*-H). Instead, DMA*-H uses separate hypernetworks for the dynamics model, policy, and Q-value function. Apart from a KL-loss term and a contrastive-loss term, DMA*-H (RL only) closely resembles R2PGO (Li et al., 2024b) in an online RL setting. It does not employ a hypernetwork for the dynamics model, so the hyperweights for the RL modules are not aligned with the dynamics model. Our results indicate that normalization, masking, hypernetwork sharing, and dynamics-model alignment are all beneficial. For a detailed discussion, see Appendix A.3.

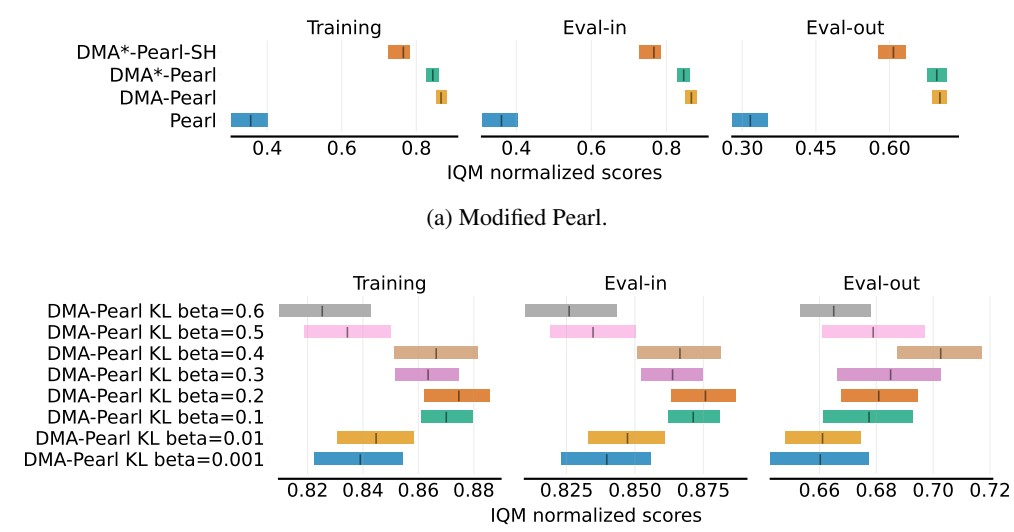

(a) Modified Pearl.

(b) $\beta$ KL weighting.

Figure 16: Interquartile mean (IQM) (Agarwal et al., 2021) based on AER scores (cf. Section 5.1) aggregated over the contextualized environments (cf. Section 5.3). We distinguish results for contexts in the three context sets $\mathcal{C}_{\text{train}}$, $\mathcal{C}_{\text{eval,in}}$ and $\mathcal{C}_{\text{eval,out}}$. In a) we compare the original Pearl approach aligned with the Q-function to the dynamic model-aligned variant that we are using as a baseline, DMA-Pearl. Additionally, we incorporate our additions to DMA and the shared hypernetwork context utilization to Pearl. In b) we test different $\beta$ weighting parameters for the KL term in Pearl and decided for $\beta = 0.2$ when using DMA-Pearl as a baseline.

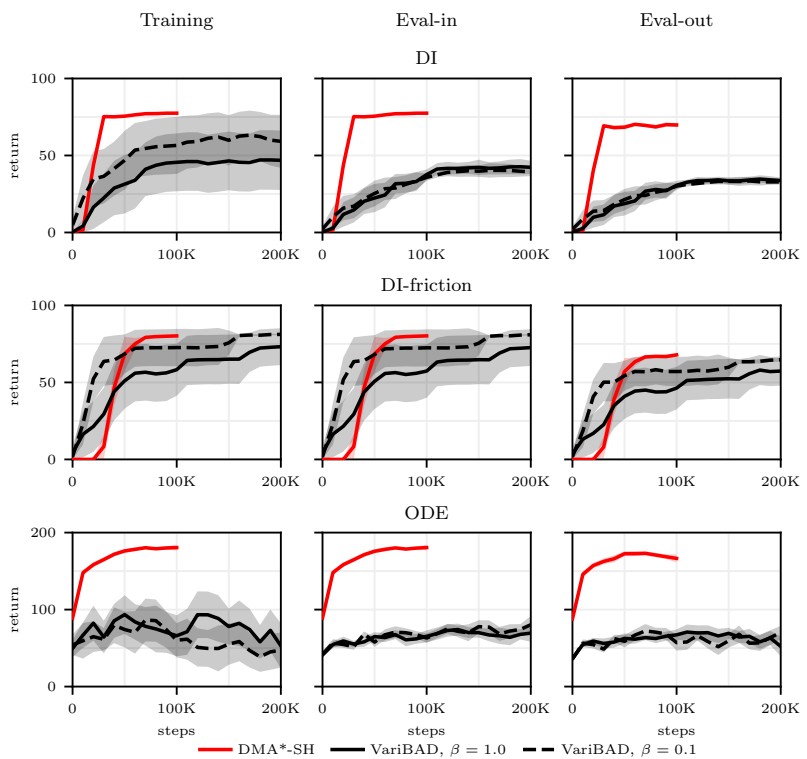

Figure 17: Returns over training steps, averaged over the contexts used for training. Comparison to the meta RL approach VariBAD (Zintgraf et al., 2020). See Remark 11. VariBAD is based on the on-policy PPO, hence we are allowing more environment steps. We do not see any improvement after 200K steps. Two KL-weights $\beta$. are tested.

# F  ENVIRONMENT CONTEXTUALIZATION

A summary of the contextualization for the environments introduced in Section 5.3 is provided in Table 3. Additionally, we include more environments that were not part of the main aggregation in Figure 3 or the ablation studies; results and brief descriptions are provided in Section G.2. Ranges correspond to the context sets $\mathcal{C}_{\text{train}}$, $\mathcal{C}_{\text{eval,in}}$ and $\mathcal{C}_{\text{eval,out}}$. All environments are contextualized in two context dimensions. Compared to a one-dimensional contextualization, this impedes training significantly, as also observed in Beukman et al. (2023).

| | | Context ranges | | |
|---|---|---|---|---|
| Name | Context | Training | Eval-in | Eval-out |
| DI | mass | $[0.5, 1.5]$ | $(0.5, 1.5)$ | $[0.1, 0.5) \cup (1.5, 2.0]$ |
| | actuator factor | $\{-1, 1\}$ | $\{-1, 1\}$ | $\{-1, 1\}$ |
| DI-friction | mass | $[0.5, 1.5]$ | $(0.5, 1.5)$ | $[0.1, 0.5) \cup (1.5, 2.0]$ |
| | friction | $[0.5, 1.5]$ | $(0.5, 1.5)$ | $[0.1, 0.5) \cup (1.5, 2.0]$ |
| ODE | $c_0$ | $[-5, 5]$ | $(-5, 5)$ | $[-10, -5) \cup (5, 10]$ |
| | $c_1$ | $[-5, 5]$ | $(-5, 5)$ | $[-10, -5) \cup (5, 10]$ |
| Cartpole | length | $[0.3, 0.85]$ | $(0.3, 0.85)$ | $[0.1, 0.3) \cup (0.85, 2.0]$ |
| | actuator factor | $\{-1, 1\}$ | $\{-1, 1\}$ | $\{-1, 1\}$ |
| BallInCup | gravity | $[8.0, 12.0]$ | $(8.0, 12.0)$ | $[1.0, 8.0) \cup (12.0, 20.0]$ |
| | tendon length | $[0.24, 0.36]$ | $(0.24, 0.36)$ | $[0.1, 0.24) \cup (0.36, 0.5]$ |
| Walker | gravity | $[4.9, 14.7]$ | $[4.9, 14.7]$ | $[1.0, 4.9) \cup (14.7, 19.6]$ |
| | actuator factor | $[0.5, 1.5]$ | $(0.5, 1.5)$ | $[0.1, 0.5) \cup (1.5, 2.0]$ |
| ReacherEasy | arm length factor | $[0.8, 1.2]$ | $(0.8, 1.2)$ | $[0.4, 0.8) \cup (1.2, 1.6]$ |
| | actuator factor | $\{-1, 1\}$ | $\{-1, 1\}$ | $\{-1, 1\}$ |
| ReacherHard | arm length factor | $[0.8, 1.2]$ | $(0.8, 1.2)$ | $[0.4, 0.8) \cup (1.2, 1.6]$ |
| | actuator factor | $\{-1, 1\}$ | $\{-1, 1\}$ | $\{-1, 1\}$ |
| Cheetah | leg length factor | $[0.8, 1.2]$ | $(0.8, 1.2)$ | $[0.4, 0.8) \cup (1.2, 1.6]$ |
| | actuator factor | $\{-1, 1\}$ | $\{-1, 1\}$ | $\{-1, 1\}$ |
| WalkerGym | gravity | $[4.9, 14.7]$ | $[4.9, 14.7]$ | $[1.0, 4.9) \cup (14.7, 19.6]$ |
| | actuator factor | $[0.5, 1.5]$ | $(0.5, 1.5)$ | $[0.1, 0.5) \cup (1.5, 2.0]$ |
| HopperGym | gravity | $[4.9, 14.7]$ | $[4.9, 14.7]$ | $[1.0, 4.9) \cup (14.7, 19.6]$ |
| | actuator factor | $[0.5, 1.5]$ | $(0.5, 1.5)$ | $[0.1, 0.5) \cup (1.5, 2.0]$ |

Table 3: Environment contextualization.

| Name | Bounds |
|---|---|
| DI | $[0, 100]$ |
| DI-friction | $[0, 100]$ |
| ODE | $[0, 200]$ |
| Cartpole | $[0, 1000]$ |
| BallInCup | $[0, 1000]$ |
| Walker | $[0, 1000]$ |
| ReacherEasy | $[0, 1000]$ |
| ReacherHard | $[0, 1000]$ |
| Cheetah | $[0, 1000]$ |
| WalkerGym | $[0, 5000]$ |
| HopperGym | $[0, 3800]$ |

Table 4: Environment specific bounds for episodic returns. Used to compute interquartile mean (IQM) scores that are comparable across environments.

# G DETAILED RESULTS

In Table 1 AER scores are aggregated over the three context sets. These are considered separately in the following Tables 5-7 for a more detailed view. DMA*-SH performs favorable throughout the context sets $\mathcal{C}_{\text{train}}$, $\mathcal{C}_{\text{eval,in}}$ and $\mathcal{C}_{\text{eval,out}}$. Learning curves are presented in Figure 18, separately for each environment and context set. Depending on the environment, for training we allow $100\,000 - 200\,000$ environment steps per context instance and the same amount of total gradient update steps. Note, that we use $n_c = 20$ contexts for training, hence, $2\,000\,000 - 4\,000\,000$ environment steps in total. DMA-SH* shows consistently desirable performance.

| Name | Context-Aware | | Context-Unaware | | Context-Inferred | | | |
|---|---|---|---|---|---|---|---|---|
| | Concat | DA | DR | Amago | DMA | DMA-Pearl | **DMA*** | **DMA*-SH** |
| DI | 75±4 | **78±1** | 17±14 | 66±16 | 74±2 | 73±3 | 77±1 | **78±1** |
| DI-friction | 71±24 | 80±1 | 72±24 | **82±1** | 62±26 | 79±1 | 71±24 | 81±1 |
| ODE | 180±8 | **183±7** | 63±15 | 178±3 | 173±5 | 174±11 | 179±7 | **183±6** |
| Cartpole | 929±28 | 934±45 | 658±78 | 667±134 | 919±42 | 904±71 | 941±34 | **972±18** |
| BallInCup | **974±4** | **971±5** | 960±31 | 718±246 | **975±2** | **976±2** | **974±4** | **972±7** |
| Walker | **895±12** | 875±33 | **896±18** | 838±24 | 860±75 | **900±17** | 876±23 | **900±32** |
| Norm. Mean | 0.86 | 0.88 | 0.62 | 0.76 | 0.83 | 0.86 | 0.86 | **0.89** |

Table 5: **Train AER scores** and standard deviations (cf. Section 5.1) for each contextualized environment (cf. Section 5.3). Results for context set $\mathcal{C}_{\text{train}}$. We compare our approaches DMA* and DMA*-SH to the baselines (cf. Section 5.2). Best AER scores are highlighted bold. In case multiple approaches are highlighted for an environment, they are within 99% of the maximal achieved AER score. Environment-specific normalization factors are used for the row *Norm. Mean* (cf. Section 5.3).

| Name | Context-Aware | | Context-Unaware | | Context-Inferred | | | |
|---|---|---|---|---|---|---|---|---|
| | Concat | DA | DR | Amago | DMA | DMA-Pearl | **DMA*** | **DMA*-SH** |
| DI | 75±4 | **78±1** | 17±14 | 66±16 | 74±2 | 73±2 | 77±1 | **78±1** |
| DI-friction | 71±24 | 80±1 | 72±24 | **82±1** | 62±26 | 79±1 | 71±24 | 81±1 |
| ODE | 181±8 | **183±7** | 63±15 | 178±3 | 173±5 | 173±12 | 178±7 | **182±5** |
| Cartpole | 930±27 | 935±45 | 659±79 | 668±134 | 919±42 | 905±71 | 941±34 | **972±17** |
| BallInCup | **974±4** | **972±4** | 955±48 | 721±241 | **975±3** | **976±4** | **975±4** | **974±4** |
| Walker | **896±13** | 878±34 | **903±15** | 841±27 | 862±76 | **900±19** | 881±28 | **898±36** |
| Norm. Mean | 0.86 | 0.88 | 0.62 | 0.77 | 0.83 | 0.86 | 0.86 | **0.89** |

Table 6: **Eval-in AER scores** and standard deviations (cf. Section 5.1) for each contextualized environment (cf. Section 5.3). Results for context set $\mathcal{C}_{\text{eval,in}}$. We compare our approaches DMA* and DMA*-SH to the baselines (cf. Section 5.2). Best AER scores are highlighted bold. In case multiple approaches are highlighted for an environment, they are within 99% of the maximal achieved AER score. Environment-specific normalization factors are used for the row *Norm. Mean* (cf. Section 5.3).

| Name | Context-Aware | | Context-Unaware | | Context-Inferred | | | |
|---|---|---|---|---|---|---|---|---|
| | Concat | DA | DR | Amago | DMA | DMA-Pearl | **DMA\*** | **DMA\*-SH** |
| DI | 65±5 | 70±2 | 16±8 | 52±12 | 42±6 | 58±6 | 70±2 | **71±3** |
| DI-friction | 54±21 | 68±3 | 61±20 | **73±1** | 45±16 | 65±5 | 62±21 | 69±2 |
| ODE | 126±14 | 172±11 | 63±14 | 148±1 | 152±8 | 165±7 | 168±11 | **173±5** |
| Cartpole | 731±49 | 808±87 | 613±78 | 581±89 | 862±30 | 842±58 | 901±47 | **958±25** |
| BallInCup | **806±41** | 674±78 | 618±73 | 462±104 | 769±53 | 751±39 | 729±64 | 708±61 |
| Walker | 519±42 | **573±31** | 568±34 | 556±27 | 540±37 | **576±20** | 550±25 | **571±25** |
| Norm. Mean | 0.65 | 0.72 | 0.48 | 0.6 | 0.63 | 0.7 | 0.72 | **0.75** |

Table 7: **Eval-out AER scores** and standard deviations (cf. Section 5.1) for each contextualized environment (cf. Section 5.3). Results for context set $\mathcal{C}_{\text{eval,out}}$. We compare our approaches DMA\* and DMA\*-SH to the baselines (cf. Section 5.2). Best AER scores are highlighted bold. In case multiple approaches are highlighted for an environment, they are within 99% of the maximal achieved AER score. Environment-specific normalization factors are used for the row *Norm. Mean* (cf. Section 5.3).

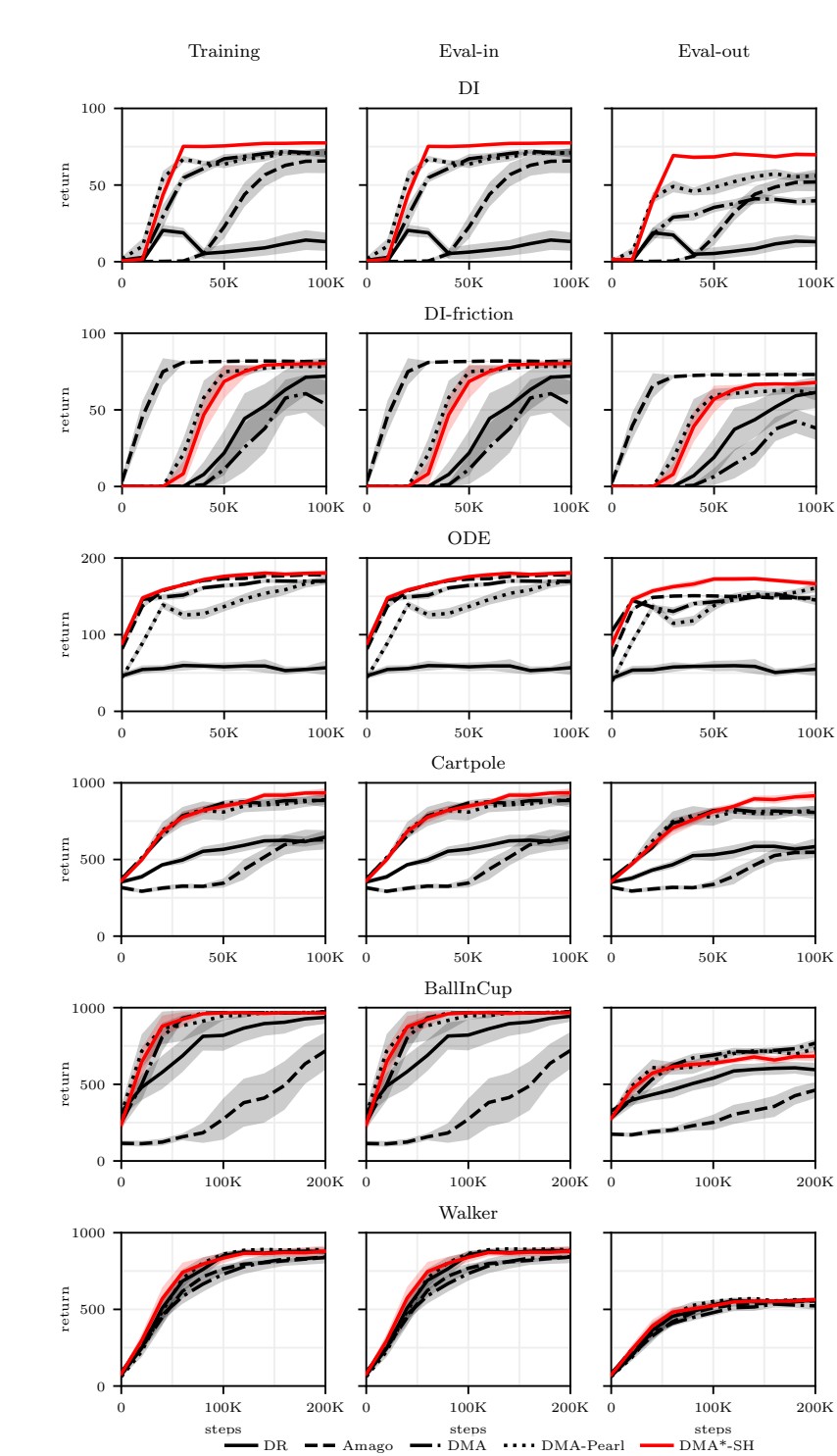

Figure 18: Returns over training steps, averaged over 20 contexts used for training. Comparison to baselines where context information is not explicitly available (cf. Section 5.2).

## G.1 CONTEXT-INSTANCE GENERALIZATION ANALYSIS

While the aggregated IQM provide a convenient high-level summary of performance across environments and context sets, contextual RL introduces an additional axis of variation that requires finer granularity (Benjamins et al., 2023; Ndir et al., 2024; Prasanna et al., 2024). To make generalization behavior explicit, we complement the aggregated metrics with detailed visualizations at the level of individual context instances.

For each environment, we evaluate our proposed DMA*-SH against baselines across the full grid of training and evaluation contexts and visualize the results using context-wise bar plots (Figures 19–20) and heatmaps (Figures 21–26). These plots reveal how performance changes as evaluation contexts drift from the training distribution and highlight failure cases, such as the inability of the context-unaware DR baseline to handle non-overlapping dynamics (e.g., DI), or the challenges faced by the context-aware Concat method when dealing with extreme values of the context *parameter_0* in the out-of-distribution regime of the ODE environment (Figure 23 and Table 7), indicating difficulties with both positive and negative values of *parameter_0*. In Cartpole, DMA*-SH achieves impressive consistency across all context instances (Figure 20). This instance-level view exposes trends that aggregate statistics may obscure and provides a clearer understanding of where and how generalization breaks down.

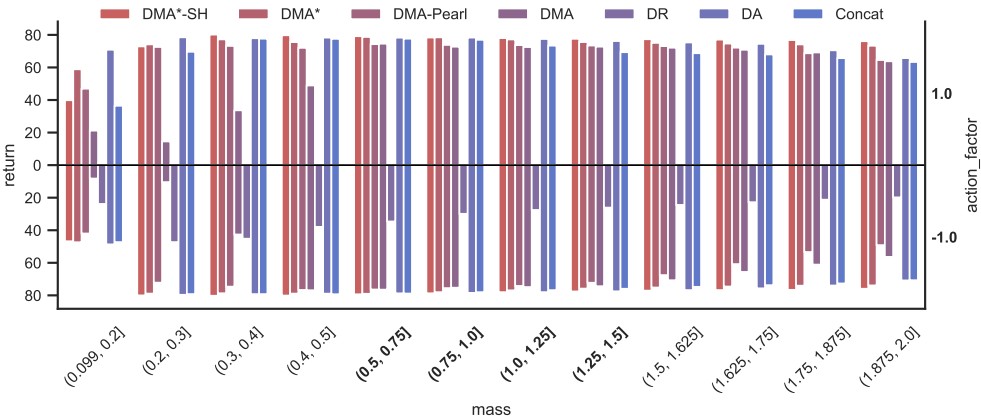

Figure 19: Bar plot for DI to visualize AER for individual context instances and different methods. Bold labels refer to contexts used during training.

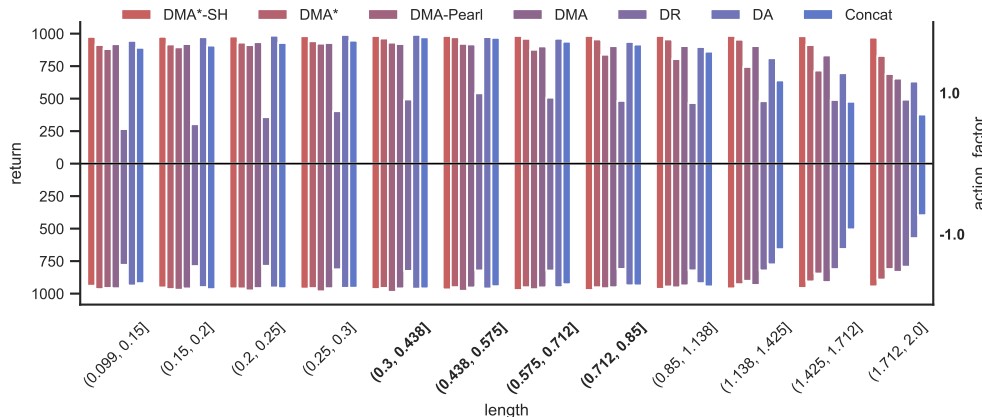

Figure 20: Bar plot for Cartpole to visualize AER for individual context instances and different methods. Bold labels refer to contexts used during training.

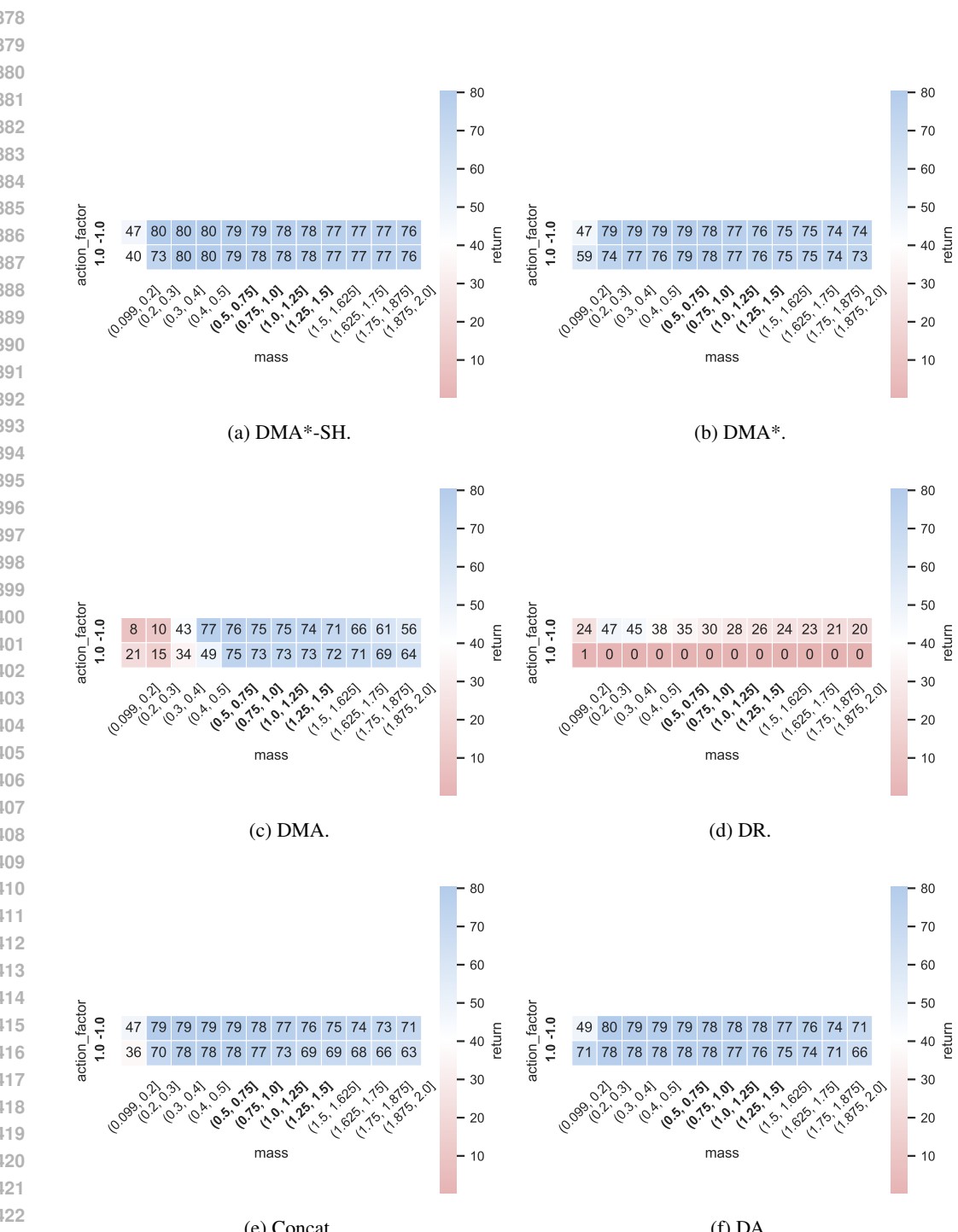

Figure 21: Heatmaps for DI to visualize AER for individual context instances. Bold labels refer to contexts used during training.

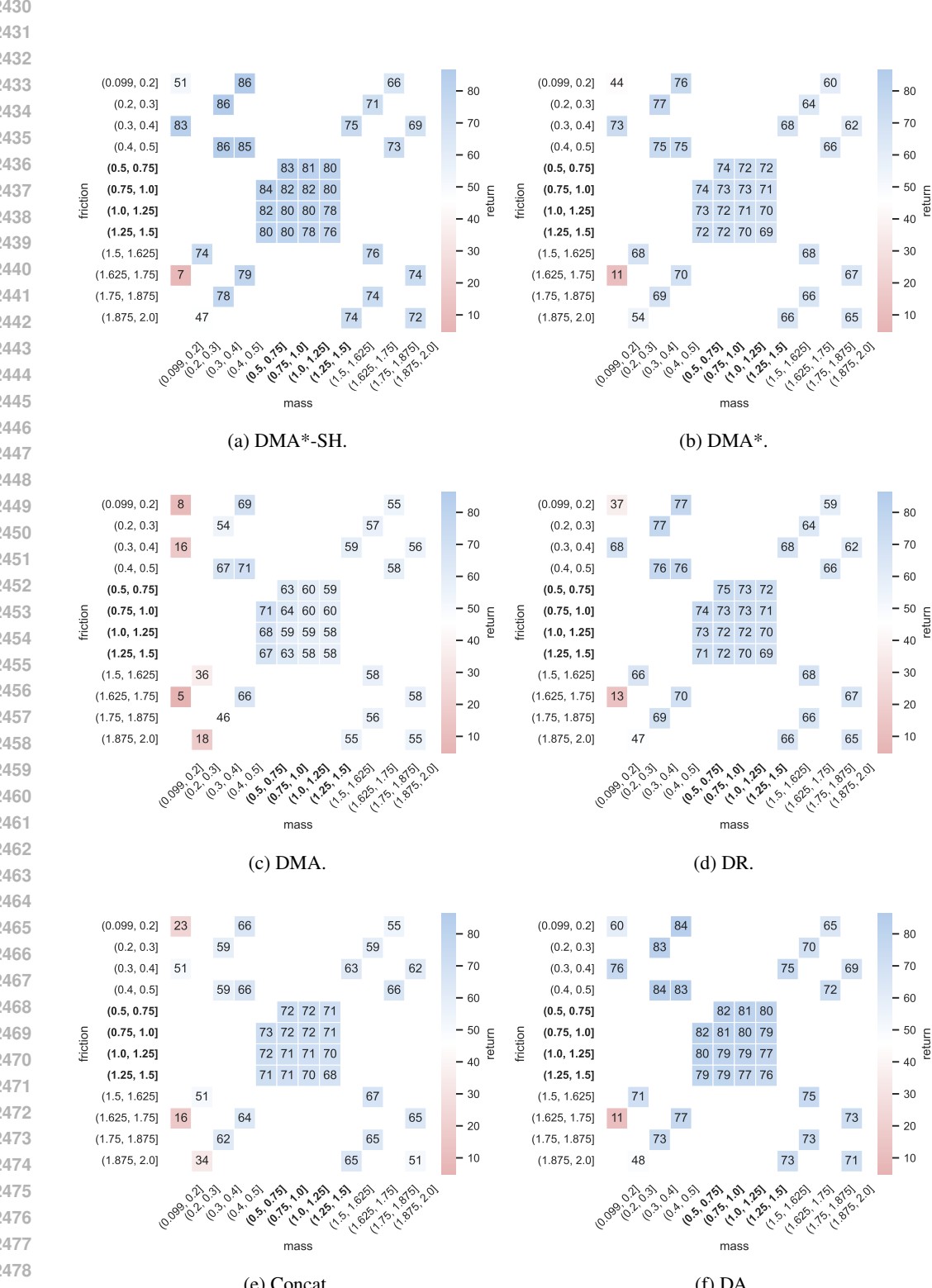

Figure 22: Heatmaps for DI-friction to visualize AER for individual context instances. Bold labels refer to contexts used during training.

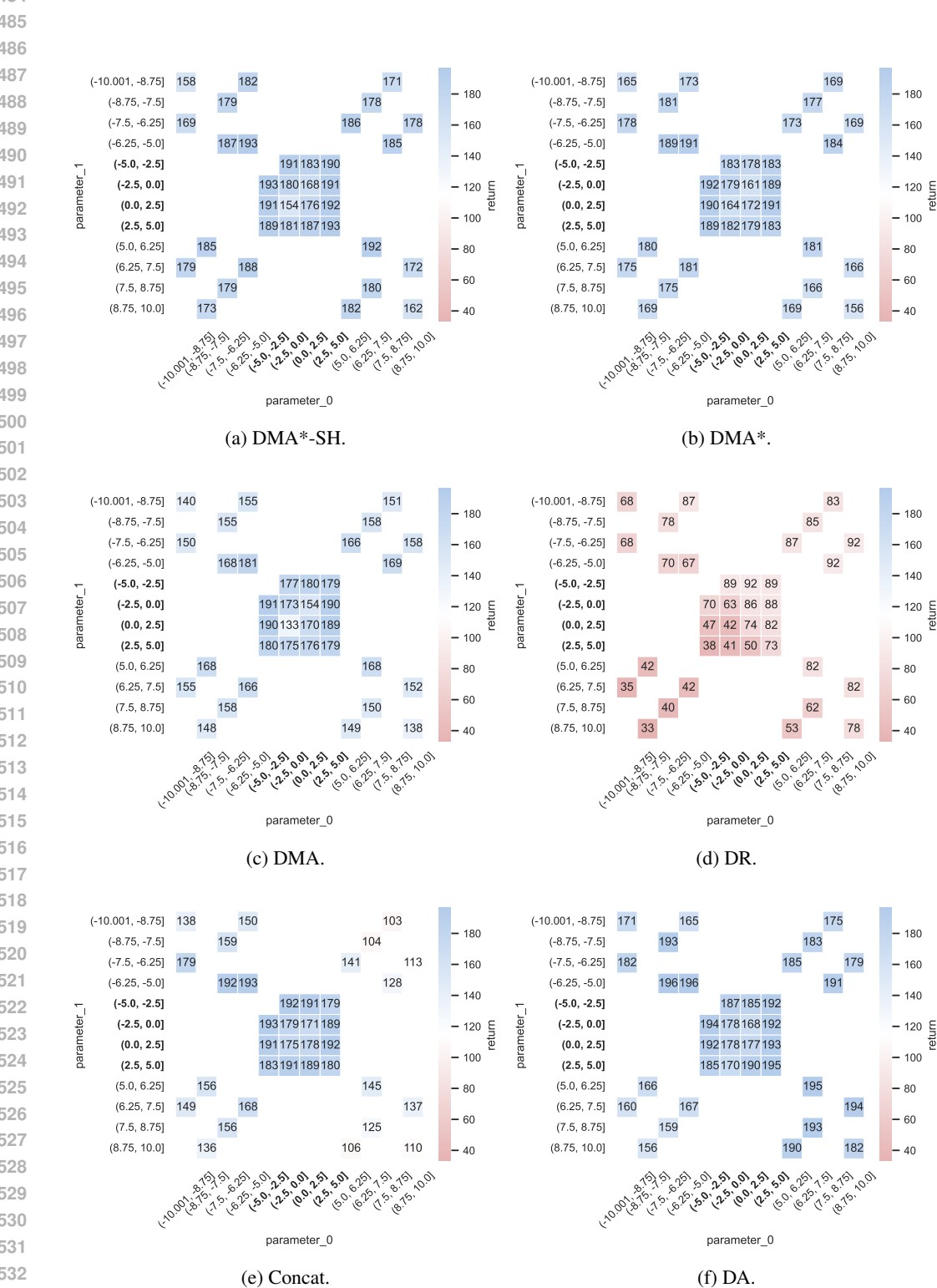

Figure 23: Heatmaps for ODE to visualize AER for individual context instances. Bold labels refer to contexts used during training.

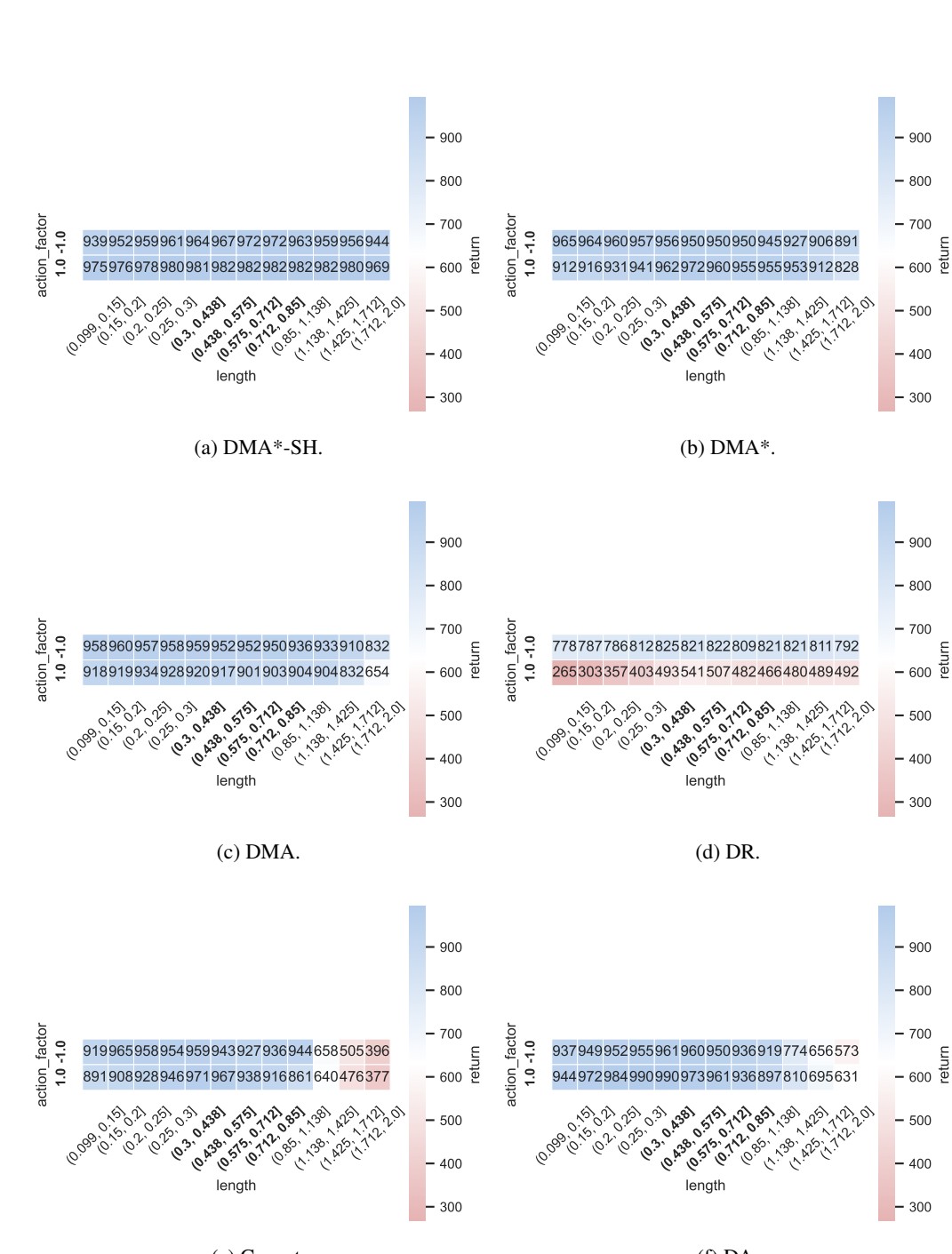

Figure 24: Heatmaps for Cartpole to visualize AER for individual context instances. Bold labels refer to contexts used during training.

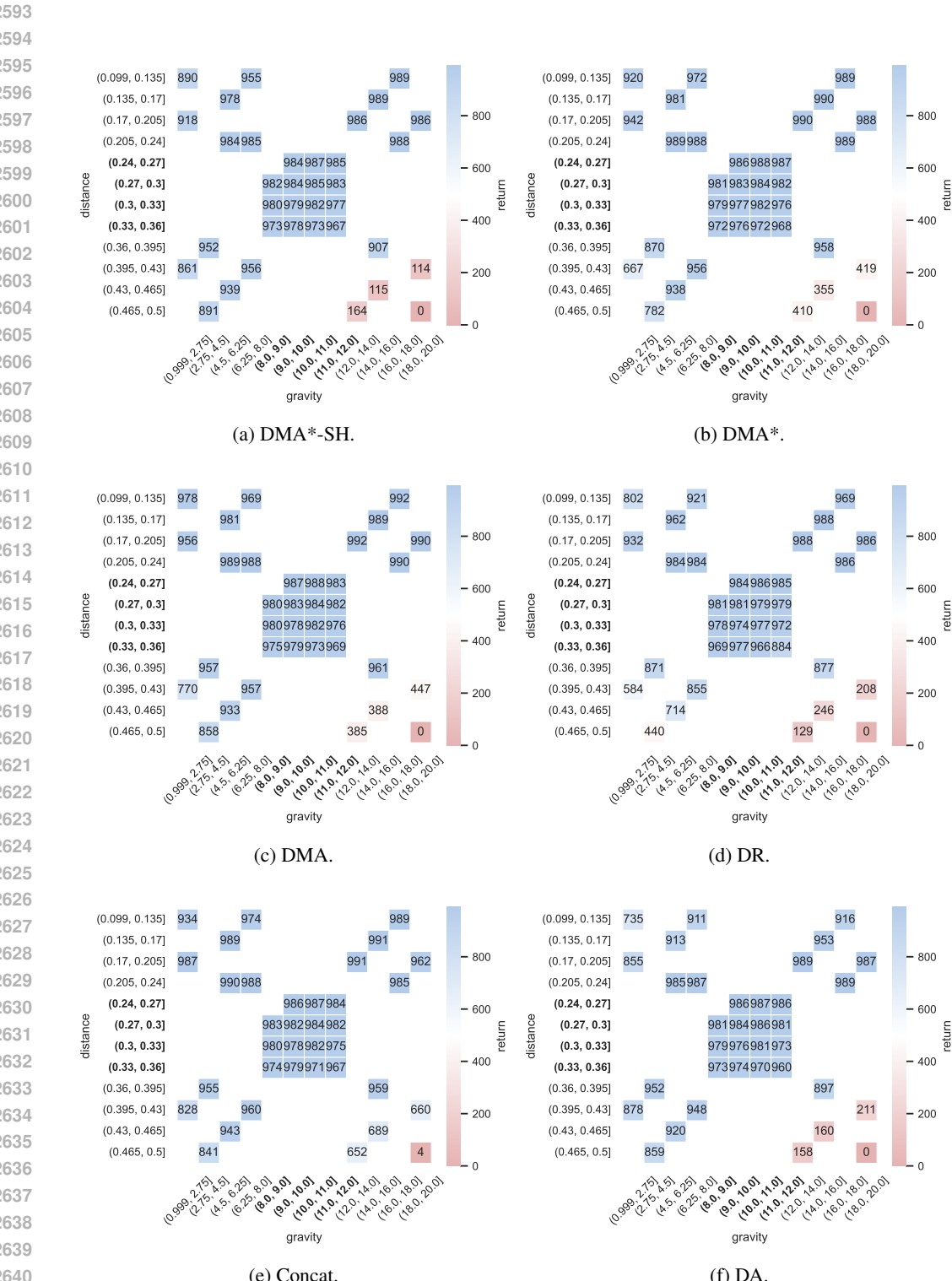

Figure 25: Heatmaps for BallInCup to visualize AER for individual context instances. Bold labels refer to contexts used during training.

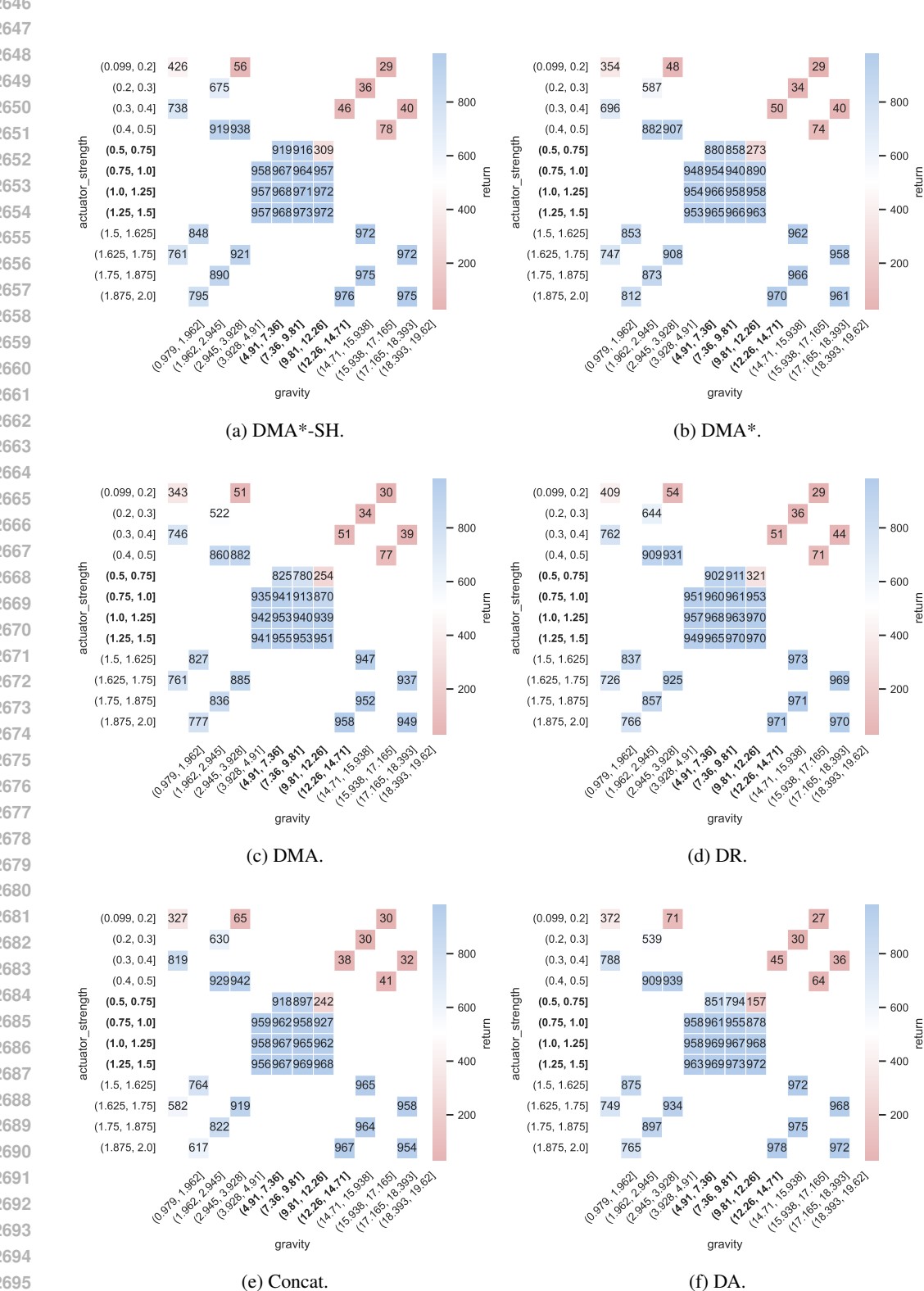

Figure 26: Heatmaps for Walker to visualize AER for individual context instances. Bold labels refer to contexts used during training.

## G.2 ADDITIONAL ENVIRONMENTS AND CONTEXTUALIZATIONS

In addition, more environments are included that were not used in the previous results section and during the ablation studies (Section E).

**ODE-X.** The environment from Beukman et al. (2023) (c.f. Section 5.3) can be easily extended with more than two context parameters. The ordinary differential equation, here, is parameterized by X context variables $c_0$, $c_1$, $c_2$, ...: $x_{t+1} = x_t + \dot{x}_t\, dt$, $\quad \dot{x} = c_0 a + c_1 a^2 + c_2 a^3 + \ldots$. The goal of the agent is to control the action $a$ to keep the state close to $x = 0$. Unaware agents perform poorly, indicating *non-overlapping* contexts (Beukman et al., 2023). More parameters/contexts pose a particular challenge in terms of scalability w.r.t. the context dimension.

**ReacherEasy/Hard.** From the DM Control Suite (reacher-easy/hard-v0) (Tassa et al., 2018). A two-link planar arm is rewarded for reaching a target sphere. In the easy task the target sphere is bigger than in the hard task. Context variables are arm length and an actuator factor ($\pm 1$). The former is highly involved in the movement dynamics and the latter creates *non-overlapping* contexts.

**Cheetah.** From the DM Control Suite (cheetah-run-v0) (Tassa et al., 2018). A planar biped is rewarded for moving forward fast (Wawrzyński, 2009). Context variables are leg lengths and an actuator factor ($\pm 1$). The former is highly involved in the movement dynamics and the latter creates *non-overlapping* contexts.

**WalkerGym.** From Gymnasium MuJoCo (Walker2d-v5) (Towers et al., 2024). A planar walker is rewarded for moving forward (Lillicrap et al., 2015). Context variables are actuator strength (referred to here as an actuator factor) and gravity. Since contexts are continuous physical parameters, they are considered *overlapping*.

**HopperGym.** From Gymnasium MuJoCo (Hopper-v5) (Towers et al., 2024). A planar hopper is rewarded for moving forward without falling over. Context variables are actuator strength (referred to here as an actuator factor) and gravity. Since contexts are continuous physical parameters, they are considered *overlapping*.

In Table 8 AER scores are aggregated over the three context sets. The three contexts sets are considered separately in Tables 9–11 for a more detailed view. For the DMC-based environments we allow $200\,000$, for the Gymnasium-based environments we allow $500\,000$ environment steps per context instance and the same amount of total gradient update steps. DMA*-SH performs particularly well in settings where arm and leg lengths are contextualized. These variations strongly influence the motion dynamics and are notably more difficult to infer. In particular, ReacherHard demands highly precise movements, and therefore a highly precise policy, to consistently reach the target position.

Figure 27 shows performance aggregated across six ODE-X variants, ODE-1, ODE-2, ..., ODE-6. It implies favorable scalability to higher context dimensions for DMA*-SH. Especially the context-aware Concat struggles with the Eval-in and Eval-out regimes.

| Name | Context-Aware | | C.-Unaware | Context-Inferred | | | |
|---|---|---|---|---|---|---|---|
| | Concat | DA | DR | DMA | DMA-Pearl | **DMA*** | **DMA*-SH** |
| ReacherEasy | 855±82 | 827±71 | 528±81 | 846±57 | 860±60 | **882±46** | **874±42** |
| ReacherHard | 605±150 | 676±62 | 231±113 | 610±144 | 638±150 | 601±179 | **780±103** |
| Cheetah | 379±31 | 374±19 | 274±35 | 384±38 | 345±43 | 380±31 | **393±36** |
| HopperGym | 2440±156 | 2453±116 | 2322±115 | **2588±143** | 2553±138 | 2547±100 | 2515±56 |
| WalkerGym | 2647±453 | **3194±287** | 2764±304 | 2945±390 | 3101±414 | 2875±538 | 3152±232 |
| Norm. Mean | 0.6 | 0.63 | 0.44 | 0.62 | 0.63 | 0.62 | **0.67** |

Table 8: AER scores and standard deviations (cf. Section 5.1) for the additional contextualized environments. Results are averaged across all contexts in the three context sets $\mathcal{C}_{\text{train}}$, $\mathcal{C}_{\text{eval,in}}$ and $\mathcal{C}_{\text{eval,out}}$. We compare our approaches DMA* and DMA*-SH to the baselines (cf. Section 5.2). Best AER scores are highlighted bold. In case multiple approaches are highlighted for an environment, they are within $99\%$ of the maximal achieved AER score. Environment-specific normalization factors are used for the row *Norm. Mean*.

| Name | Context-Aware | | C.-Unaware | Context-Inferred | | | |
| | Concat | DA | DR | DMA | DMA-Pearl | DMA* | DMA*-SH |
| --- | --- | --- | --- | --- | --- | --- | --- |
| ReacherEasy | 884±81 | 869±79 | 546±87 | 879±58 | 891±63 | **911±51** | **913±37** |
| ReacherHard | 650±164 | 725±64 | 254±117 | 659±153 | 691±161 | 656±188 | **829±117** |
| Cheetah | 432±33 | 444±15 | 311±40 | 458±45 | 418±50 | 460±37 | **465±36** |
| HopperGym | 2787±171 | 2763±129 | 2633±108 | **2918±155** | 2867±130 | 2881±99 | 2824±38 |
| WalkerGym | 3030±510 | **3691±325** | 3139±364 | 3360±437 | 3512±486 | 3267±636 | 3627±199 |
| Norm. Mean | 0.66 | 0.7 | 0.49 | 0.69 | 0.69 | 0.69 | **0.74** |

Table 9: AER scores and standard deviations (cf. Section 5.1) for the additional contextualized environments. Results for the **Training** context set $\mathcal{C}_{\text{train}}$. We compare our approaches DMA* and DMA*-SH to the baselines (cf. Section 5.2). Best AER scores are highlighted bold. In case multiple approaches are highlighted for an environment, they are within 99% of the maximal achieved AER score. Environment-specific normalization factors are used for the row *Norm. Mean*.

| Name | Context-Aware | | C.-Unaware | Context-Inferred | | | |
| | Concat | DA | DR | DMA | DMA-Pearl | DMA* | DMA*-SH |
| --- | --- | --- | --- | --- | --- | --- | --- |
| ReacherEasy | 884±81 | 870±80 | 544±87 | 880±58 | 892±64 | **912±52** | **912±38** |
| ReacherHard | 650±164 | 725±64 | 256±118 | 661±154 | 691±160 | 656±188 | **828±117** |
| Cheetah | 432±29 | 444±12 | 313±44 | 458±41 | 419±51 | 459±35 | **467±38** |
| HopperGym | 2803±164 | 2765±122 | 2645±101 | **2925±161** | 2881±126 | **2904±103** | 2828±31 |
| WalkerGym | 3045±552 | **3712±318** | 3159±376 | 3388±441 | 3538±478 | 3304±658 | 3638±210 |
| Norm. Mean | 0.66 | 0.7 | 0.49 | 0.69 | 0.69 | 0.69 | **0.74** |

Table 10: AER scores and standard deviations (cf. Section 5.1) for the additional contextualized environments. Results for the **Eval-in** context set $\mathcal{C}_{\text{eval,in}}$. We compare our approaches DMA* and DMA*-SH to the baselines (cf. Section 5.2). Best AER scores are highlighted bold. In case multiple approaches are highlighted for an environment, they are within 99% of the maximal achieved AER score. Environment-specific normalization factors are used for the row *Norm. Mean*.

| Name | Context-Aware | | C.-Unaware | Context-Inferred | | | |
| | Concat | DA | DR | DMA | DMA-Pearl | DMA* | DMA*-SH |
| --- | --- | --- | --- | --- | --- | --- | --- |
| ReacherEasy | 796±83 | 744±53 | 493±69 | 780±56 | 798±55 | **822±35** | 799±50 |
| ReacherHard | 515±120 | 578±59 | 181±103 | 512±125 | 534±128 | 492±161 | **682±76** |
| Cheetah | **272±32** | 234±30 | 198±23 | 237±29 | 199±27 | 220±22 | 247±34 |
| HopperGym | 1731±132 | 1830±97 | 1688±135 | **1920±114** | **1912±159** | 1856±98 | 1894±98 |
| WalkerGym | 1867±297 | 2179±217 | 1995±173 | 2087±293 | **2253±278** | 2055±319 | 2191±286 |
| Norm. Mean | 0.48 | 0.49 | 0.34 | 0.49 | 0.5 | 0.49 | **0.53** |

Table 11: AER scores and standard deviations (cf. Section 5.1) for the additional contextualized environments. Results for the **Eval-out** context set $\mathcal{C}_{\text{eval,out}}$. We compare our approaches DMA* and DMA*-SH to the baselines (cf. Section 5.2). Best AER scores are highlighted bold. In case multiple approaches are highlighted for an environment, they are within 99% of the maximal achieved AER score. Environment-specific normalization factors are used for the row *Norm. Mean*.

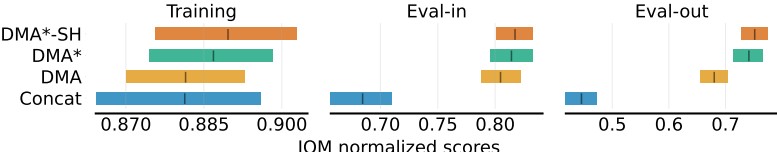

Figure 27: Interquartile mean (IQM) aggregated over six ODE variants, ODE-1, ODE-2, ... , ODE-6. In the default version ODE(-2) is governed by an ordinary differential equation parameterized by two context variables $c_0$ and $c_1$. To test for **scalability in the explicit context dimension** the number of parameters/contexts is increased.

