# OpenReview forum: "Shared Dynamic Model-Aligned Hypernetworks for Zero-Shot Generalization in Contextual Reinforcement Learning"
_ICLR.cc/2026/Conference — Submitted to ICLR 2026_

### Official Review · Reviewer_kM3b · 2025-10-28

**Soundness:** 2
**Presentation:** 2
**Contribution:** 3
**Rating:** 2
**Confidence:** 5

**Summary:**

The work proposes to use dynamics aligned context representations via shared hypernetworks. The work first describes how to learn dynamics aligned models from a window of past observations (as commonly also done in meta-RL) and then describes how input masking and input/output normalization are applied to improve over standard practices. Based on this improved dynamics model, the work then discusses how the hypernetwork is producing shared "adapter" weights for both the policy and Q functions.
The work evaluates both standard dynamics model aligned representations, their improved variants and the shared hypernetwork variant against a broad range of baselines on a variety of benchmarks.

**Strengths:**

The work tackles an important problem for learning general policies via reinforcement learning. Particularly, the work assumes that one can not make use of prior knowledge of the environment which could be readily be provided to a learning agent and instead learns to infer contextual changes.
The baselines are well chosen and meaningful and the spread of environments is also commendable.

**Weaknesses:**

Despite the listed strengths in experimental setup, I believe the work needs to improve how the results are presented. While the IQM provides good overall aggregate statistics, I believe it is not well fit on it's own to discuss the generalization capabilities of the discussed methods. For example, in Figure 3 the results are split into train, eval-in and eval-out results. However, the aggregation does not show how the performances on individual contexts look like. This seems particularly important to me as, the further away from the training distribution one goes, it is reasonable to expect less and less performance. An aggregate statistic like the IQM hides this important information and it is not clear if the proposed methods have better out-of-distirbution capabilities further from the training distribution than others. Similarly, the in-distribution results do not tell us how close the in-distribution evaluation contexts are to the training ones. The survey by Kirk. et al (cited in the paper) showed how this could be evaluated and plotted better and the Work of Prasanna et al (also cited in the paper) gives a direct example for a new cRL method.

Further, since the environments are dissimilar, I do believe that the aggregate over all environments is not particularly useful. The work has to give more fine grained analysis on a per-environment level to better highlight where the proposed methods shine.

The "Informativeness" analysis in Section 5.5 is similar to a different cRL work by Ndir et al (https://openreview.net/forum?id=51XSWH0mgN). Since this work also contrasts dynamics-model aligned learning with directly using explicit context information, this work warrants a citation. Further, I am unsure if the R2 score here can be seen as a measure of "informativeness" as it does not directly measure the "information content" of the data. Instead it simply gives a score of how similar the learnt embedding is to a ground truth. Being highly informative, a learnt embedding can still be highly dissimilar to the ground truth. Thus, the wording of informativeness should likely be avoided. If I misunderstood something, I'd be happy to take back my criticism.

While I highly appreciate the work and would like to see it published, in it's current state I have to vote for rejection. Relying only on aggregate statistics for the analysis should not be permissible for a work that discusses generalization and needs to provide more detailed insights into relation of evaluation contexts to training ones. I fear that this would require significant rewriting of the experimental section, which I am not sure if it is doable in the short rebuttal time window. I am more than happy to increase my score if the authors can convince me that they can improve the analysis.

**Questions:**

Why is the R2 score analysis a measure of "informativeness"?
Why did the analysis only rely on aggregate statistics?
Which environments are similar enough that they could be grouped in an aggregate statistic?

---

> ### Author Response · Authors · 2025-11-28
> **Response to Reviewer kM3b (Part 1/2)**
>
> We thank the reviewer for their thoughtful and constructive feedback. We appreciate the positive remarks regarding the importance of the problem setting, our focus on learning contextual representations without relying on prior environment knowledge, the choice of baselines, and the diversity of evaluated environments. Below we address all raised concerns in detail and outline the improvements made to the manuscript.
>
>
> **Response to Weaknesses 1, 3, and Questions 2, 3**
>
> We appreciate the reviewer's detailed comments regarding the use of aggregate statistics in contextual RL. While aggregating results across environments is common practice in RL, we agree that contextual variation introduces an additional axis that must be handled carefully.
>
> In Figure 3, we aggregate scores over environments but separate them across different context sets. This design choice mirrors the presentation in Prasanna et al., Figure 4(a) [1], where environment-level aggregation is similarly combined with context-level separation. In Table 1, we instead distinguish between environments while averaging over all contexts, again following prior work to provide a concise high-level summary.
>
> Our intention in presenting Figure 3 and Table 1 prominently in the main paper was to give readers an immediate overview of overall performance. More fine-grained analyses are provided in Appendix G (Detailed Results), where results are broken down by context sets at the per-environment level and complemented with training curves.
>
> However, we agree with the reviewer that context-instance-level results were missing, and that these are important for understanding how performance degrades as evaluation contexts move further from the training distribution. Following the structure used by Prasanna et al. [1], we have now added per-context heatmaps (Figures 21-26). These plots explicitly illustrate:
>
> - The expected performance decrease as evaluation contexts diverge from the training range
> - The inability of the context-unaware DR baseline to handle truly non-overlapping contexts (e.g., DI in Figure 21(d))
> - The overall performance benefit of DMA*-SH compared to baselines
>
> For DI and Cartpole, we additionally present bar plots (Figures 19-20) comparing different methods at the context level. The performance gain of DMA*-SH is especially evident for Cartpole, where it achieves consistently high performance across all considered length-context instances while robustly handling actuator inversion.
>
> To further analyze context representations at a fine-grained per-context level, we have added t-SNE visualizations (Figure 8) and per-context cosine similarity matrices (Figure 9) in Appendix D.
>
> We hope these additions provide the requested granularity and make the generalization behavior of all methods more transparent and interpretable.

---

> > ### Author Response · Authors · 2025-11-28
> > **Response to Reviewer kM3b (Part 2/2)**
> >
> > **Response to Weakness 2 and Question 1:**
> >
> > We thank the reviewer for pointing out the relevance of Ndir et al. [2]; we have added this citation to the Related Works section.
> >
> > Regarding the use of $R^2$ as a measure of "informativeness," we agree that $R^2$ does not directly quantify information content. While several works in representation learning employ a notion of "informativeness" based on predictability (e.g., [2, 3, 4]), we acknowledge that relying solely on $R^2$ is limiting.
> >
> > To address this, we now base our analysis on the mutual information $I(z_t; c)$ between the learned embedding and the explicit context variable. This provides a more principled and general measure of informativeness. We have updated the corresponding Figure 4 accordingly. There is, however, a well-known relationship between MI and the original $R^2$ under linear/Gaussian assumptions, which clarifies why $R^2$ previously served as a reasonable, though restricted, proxy.
> >
> > Specifically, if $z_t$ and $c$ are jointly Gaussian with Pearson correlation $\rho = \mathrm{Corr}(z_t, c)$, then the standard closed-form formula yields:
> > $$
> > I(z_t; c) = -\frac{1}{2}\log(1 - \rho^2).
> > $$
> > In a linear regression of $c$ on $z_t$, we have $R^2 = \rho^2$. Thus,
> > $$
> > I(z_t; c) \approx -\frac{1}{2}\log(1 - R^2).
> > $$
> > This shows that under linear/Gaussian assumptions, high $R^2$ corresponds to high MI and vice versa. However, nonlinear dependencies may yield high MI but low $R^2$, which motivates our switch to MI as the primary and more robust measure of informativeness.
> >
> > We also evaluated the random-forest-based explicitness metrics used in Ndir et al. [2] and Eastwood et al. [3] and found qualitatively consistent trends.
> >
> > You expressed concern that addressing these issues might require "significant rewriting" not feasible in the rebuttal period. We want to assure you that we have already implemented all the changes described above. The revised manuscript now contains **new fine-grained analysis** with performance-vs-context-distance plots, and **new Mutual Information analysis**. A natural way to group environments is using our Definition 6 (Appendix A.2) of Overlapping and Non-overlapping contexts.
> >
> > We believe these revisions directly and fully address your primary criticism. The experimental section now provides the detailed, multi-faceted insights into generalization that you rightly demanded. Given that this was your major concern, we hope you will find the revised manuscript substantially improved and consider raising your score to a Accept. Our work makes a strong, novel contribution that is both theoretically and empirically grounded, and we are confident the new analysis robustly supports our claims.
> >
> > [1] Sai Prasanna, Karim Farid, Raghu Rajan, and Andre Biedenkapp. Dreaming of many worlds: Learning contextual world models aids zero-shot generalization. Reinforcement Learning Journal, 2024.
> >
> > [2] Tidiane Camaret Ndir, Andre Biedenkapp, and Noor Awad. Inferring behavior-specific context improves zero-shot generalization in reinforcement learning. In European Workshop on Reinforcement Learning, 2024.
> >
> > [3] Cian Eastwood and Christopher KI Williams. A framework for the quantitative evaluation of disentangled representations. In International Conference on Learning Representations, 2018.
> >
> > [4] Marc-Andre Carbonneau, Julian Zaidi, Jonathan Boilard and Ghyslain Gagnon. Measuring disentanglement: A review of metrics, In IEEE Transactions on Neural Networks and Learning Systems, 2022.

---

### Official Review · Reviewer_2MdE · 2025-10-28

**Soundness:** 2
**Presentation:** 2
**Contribution:** 1
**Rating:** 2
**Confidence:** 3

**Summary:**

This paper proposes incorporating a shared adapter layer, whose weights are generated by a hypernetwork, into the dynamics model, policy, and critic networks in contextual reinforcement learning (RL). The goal is to align these three components under a unified, learned context representation. Additionally, the paper introduces masking and normalization techniques during training to further enhance policy performance. Experimental results demonstrate that the proposed method achieves state-of-the-art (SOTA) performance on most benchmark tasks.

**Strengths:**

1. The paper clearly identifies the challenge of contextual RL when explicit context information is unavailable. To implicitly infer context from data, the authors propose a novel framework that learns context representations by integrating a shared adapter layer—parameterized via a hypernetwork—into the dynamics, policy, and critic networks. This design provides an interesting and potentially valuable direction for context representation learning in contextual RL.
2. The paper is well-written overall, and the figures illustrating the proposed method are clear and easy to understand.

**Weaknesses:**

1. The novelty of the proposed approach is somewhat limited. The use of a hypernetwork to produce shared adapter parameters is not entirely new in contextual or meta-RL. For instance, a similar idea was explored in [1], which employed a comparable scheme to train the actor and critic in offline meta-RL. Moreover, the use of masking and normalization techniques is quite common in machine learning, so these aspects alone may not constitute significant methodological contributions.
2. The paper lacks theoretical analysis. The approach appears to rely primarily on empirical findings rather than theoretical justification. For example, there is no analysis explaining why a particular masking ratio or normalization strategy was chosen, nor how these design choices affect learning stability or performance.

[1] Li Z, Lin Z, Chen Y, et al. Efficient offline meta-reinforcement learning via robust task representations and adaptive policy generation. Proceedings of the Thirty-Third International Joint Conference on Artificial Intelligence (IJCAI-24), 2024: 4524–4532.

**Questions:**

1. I am curious about the 100% masking of the input trajectory shown in Figure 7. Could the authors clarify what this represents and how the model handles such fully masked inputs?

---

> ### Author Response · Authors · 2025-11-28
> **Response to Reviewer 2MdE (Part 1/2)**
>
> Thank you for your careful review and for recognizing the potential value of our shared hypernetwork design in addressing the challenging problem of implicit context inference in contextual RL. We especially appreciate your positive remarks on the clarity of our writing and figures. These encouraging points motivate us to directly address your concerns with additional theoretical and empirical evidence and clarifications that we believe substantially strengthen the contribution and novelty.
>
> **Addressing Weakness 1: Novelty and Relation to Prior Work (especially R2PGO)**
>
> We fully agree that hypernetworks have been explored in offline/meta-RL settings, and we thank you for pointing out R2PGO (IJCAI 2024) as a relevant reference. We unfortunately missed it in our initial search. However, after a thorough comparison (their Figure 1 vs. our architecture), we respectfully submit that DMA*-SH introduces several fundamental differences that go far beyond incremental engineering:
>
> | Aspect | R2PGO | DMA*-SH (Ours) |
> |--------|-------|----------------|
> | **Setting** | Offline meta-RL (no interaction during training) | Online contextual RL (zero-shot to unseen dynamics) |
> | **Hypernetwork Role** | Separate hypernets generate only upper-layer weights for actor & critic | Single hypernetwork generates adapters first trained on dynamics model, then shared with policy & critic |
> | **Alignment Mechanism** | VAE + contrastive loss on trajectories | Explicit dynamics-model alignment via state-difference reconstruction (Eq. 4) |
> | **Key Inductive Bias** | Robust task representations via contrastive learning | Consistency enforced by weight sharing across dynamics, policy, and critic |
> | **Zero-shot to Unseen Dynamics** | Not evaluated (focus on same-task-family generalization) | Core evaluation on overlapping vs. non-overlapping contexts (including actuator inversion provably unsolvable for standard domain randomization, see Lemma 9) |
>
> The most critical distinction is that R2PGO never uses the dynamics model to supervise the actor and critic hypernetworks. Their hypernetwork is trained purely from RL signals (actor/critic losses) plus a VAE/contrastive objective, and is not dynamic-model aligned. In contrast, our hypernetwork is first and foremost aligned with the true transition dynamics via the reconstruction loss (Eq. 4) and only then shared with policy/critic, enforcing alignment between representation learning and control. This provides a dramatically stronger and physically grounded inductive bias for dynamics variations.
> Empirically, this difference manifests strongly in non-overlapping contexts (e.g., actuator inversion in DI and Cartpole), where domain randomization and pure RL-driven hypernetworks fail entirely, whereas DMA*-SH succeeds because the shared adapters have already learned the correct multiplicative transformation (e.g., sign flip or mass scaling) from dynamics prediction.
>
> We couldn't find a code base for R2PGO. To explicitly isolate the benefits that dynamics-alignment affords for RL, we constructed a basic variant DMA*-H (RL only) that mirrors the R2PGO hypernetwork structure in an online RL setting. Apart from a KL-loss term and a contrastive-loss term, DMA*-H (RL only) closely resembles R2PGO but does not employ a hypernetwork for the dynamics model, so the hyperweights for the RL modules are not aligned with the dynamics model. In Figure 15 (Appendix E), we had already compared our DMA*-SH with a variant that uses separate hypernetworks for the dynamics, actor and critic (DMA*-H). We complement that now with results from DMA*-H (RL only), which in comparison to DMA*-H does not use a hypernetwork for the dynamic model.
>
> We find that both non-aligned variants, DMA*-H and DMA*-H (RL only), perform worse than DMA*-SH. These results demonstrate dynamics alignment is crucial for our method's success.

---

> ### Author Response · Authors · 2025-11-28
> **Response to Reviewer 2MdE (Part 2/2)**
>
> **Addressing Weakness 2: Lack of Theoretical Analysis**
>
> We take this point seriously and have added a new Appendix A "Theoretical Results and Supplementary Analyses".
>
> 1. **Expressiveness Proof**: In Section 4.3 (new), we show that hypernetwork-conditioned ReLU policies strictly subsume concatenation-based counterparts (**Theorem 3**, Appendix A.1). This shows why DMA*-SH has the right inductive bias to exactly model actuator inversion through multiplicative modulation of action effects (**Remark 4**).
>
>
> 2. **Geometric Analysis**:
> We demonstrate that DMA*-SH learns *directionally selective representations* that compress continuous parameter variations and separate task-critical discontinuous actuator inversions, enabling effective zero-shot generalization (Appendix A.3, A.4).
>
>       While hypernetworks and masking/normalization are not individually new, their integration into a shared, dynamic model-aligned architecture for online contextual RL is novel. In DMA*-SH, *approximate end-to-end scale invariance* arises from the interaction between input normalization, SimNorm-based context normalization, and hypernetwork-driven weight generation (Appendix A.3). Our normalization strategy combined with hypernetwork conditioning creates an approximately scale-controlled representation space in which functional modulation depends primarily on the direction of the context embedding (Appendix A.3, A.4). This results in hypernetwork-generated adapter weights that vary predominantly through directional differences in representation space, a phenomenon supported empirically by our t-SNE structure and Representation-Overlap analyses (Figures 8 and 9, Appendix D). Across all our non-overlapping benchmarks (DI, ODE, and the new Gym/DMC tasks), the central difficulty is a discontinuous action sign flip (actuator inversion).
>
>    *Shared hyperneworks amplify directional concentration*:
>
> - continuous parameters (mass/friction) are *compressed*,
>
> - discontinuous actuator flips are *expanded*,
>
>    yielding the robust *dual-axis geometry* that drives zero-shot generalization.
>
>    DMA*-SH thus learns directionally selective representations, compressing nuisance continuous variations while preserving and separating discontinuous task-critical directions such as actuator inversion. Detailed analysis in Appendix A.3, A.4.
>
> - Input/Output normalization stabilizes this alignment, yielding clean geometric separation (shown by cosine similarity scores, t-SNEs). See Figures 8 and 9 in Appendix D, and the discussion thereof. The progression DMA $\rightarrow$ DMA* $\rightarrow$ DMA*-SH observed in cosine matrices and t-SNE visualizations demonstrates that: (i) input/output normalization eliminates harmful scale effects and pathological geometry, while (ii) shared hypernetwork conditioning concentrates contextual distinctions into functionally meaningful directional axes. Together, these components yield the representation geometry beneficial for zero-shot robustness.
>
>
> 3. **Implicit Gradient Regularization**: Section 5.6 (new) shows sharing hypernetwork $\eta$ across dynamics, policy, and Q-function implicitly regularizes RL gradients toward physically consistent behaviors. Figures 5 and 6 show that in the shared case (DMA*-SH), persistently *high policy gradient norms* indicate continuous interaction with dynamics-aligned parameters, while cosine similarity reveals *strong coupling* between RL and dynamics objectives. In contrast, separate hypernetworks (DMA*-H) show collapsed gradients and near-zero cosine similarity, reflecting uncoordinated adaptation. See Figure 5 and 6, extended analysis in Appendix A.5.
>
>
> **Answer to Question 1**
>
> The 100% masking case represents the extreme scenario where the context encoder receives no input data at all, making it equivalent to context-unaware domain randomization. Performance matches DR baselines with minor deviations attributable to differences in random number generators between PyTorch and NumPy. Compare Figure 3 (IQM for DR) and Figure 11 (IQM for 100% masking).
>
> We believe these substantial additions, including theoretical proofs, geometric analysis, implicit gradient regularization evidence, and thorough ablation studies demonstrate DMA*-SH's novel contributions beyond prior work and addresses all your concerns.
>
> We sincerely thank you for your invaluable feedback, which has significantly strengthened our paper. We hope these comprehensive enhancements merit your favorable reconsideration.

---

### Official Review · Reviewer_VpJQ · 2025-10-29

**Soundness:** 2
**Presentation:** 3
**Contribution:** 2
**Rating:** 2
**Confidence:** 4

**Summary:**

This paper addresses zero-shot generalization in contextual reinforcement learning (CRL).
The authors propose a framework that learns a dynamics-aligned context representation jointly with the policy and Q-function.
A shared hypernetwork conditions on the inferred context and outputs parameters that are used across three modules — the dynamics model, the policy, and the action-value network.
Additional implementation components include random masking of transition inputs and layer normalization to stabilize the learned context.
The method is evaluated on a set of contextual continuous-control environments, showing improved in-distribution and out-of-distribution performance over several baselines such as DMA and PEARL variants.

**Strengths:**

1.	The paper identifies a relevant gap in CRL — the difficulty of learning generalizable contextual representations that can adapt to unseen environment dynamics.
2.	The idea of using one hypernetwork to generate parameters for multiple components (dynamics model, policy, and critic) is conceptually novel and promotes parameter consistency across modules.
3.	The paper includes ablations on normalization, masking, and weight sharing, demonstrating their influence on stability and performance.
4.	Results are reported with mean, IQM, and aggregate metrics across tasks, following recent reproducibility standards.

**Weaknesses:**

1.	The main ideas closely resemble VariBAD (learning a latent dynamics representation for fast adaptation) and Beukman et al., 2023 (using hypernetworks to condition multiple RL components). The proposed method mainly combines these two directions with incremental engineering contributions (masking, normalization) rather than introducing fundamentally new principles or theory.
2.	The paper does not compare against VariBAD, UNICORN, or other prominent COMRL approaches that explicitly learn context embeddings from dynamics. This omission makes it difficult to assess whether the reported improvements are meaningful beyond minor implementation differences.
3. The evaluation focuses on a small number of custom contextual environments. Broader testing on DMControl or Gym benchmarks would strengthen claims of zero-shot generalization.
4.	The paper does not provide analytical insights into why shared parameterization via a hypernetwork should improve zero-shot generalization, beyond empirical intuition.
5.	Hypernetworks increase training cost and parameter coupling; no quantitative analysis of overhead or convergence behavior is provided.

**Questions:**

1.	Why are VariBAD, UNICORN, or other context-based meta-RL methods not included in the comparison? How does your method differ in architectural or objective terms from VariBAD’s latent dynamics modeling?
2.	Have you tested versions where the hypernetwork outputs parameters only for the policy or the dynamics model, rather than all three modules? Quantifying how much of the gain comes from shared parameterization would clarify its value.
3.	What is the additional computational cost (training time, memory) of using a shared hypernetwork compared to a standard architecture? Does the model remain stable for larger or more diverse environments?
4.	How well does the learned context embedding transfer to truly unseen dynamics families (e.g., different friction coefficients or link masses outside the training distribution)? Any evidence beyond the limited set of contextual environments?

---

> ### Author Response · Authors · 2025-11-28
> **Response to Reviewer VpJQ (Part 1/2)**
>
> We thank Reviewer VpJQ for their insightful feedback. We have addressed all concerns through new experiments and analyses, significantly strengthening the paper's theoretical grounding and evaluation breadth.
>
> ### **Response to Weakness 1 & Q1: Comparison to Context-based Meta-RL Methods (VariBAD, UNICORN)**
>
> **We include a comparison with VariBAD**. Results confirm why it is not a suitable baseline for our problem setting, while UNICORN is methodologically incompatible.
>
> *   **VariBAD Cannot Faithfully Encode Actuator Inversion:** VariBAD is a meta-learning method for POMDPs that learns a belief state over tasks. It optimizes an ELBO where both posterior $q_\phi$ and prior $p$ are Gaussians (see Remark 11). KL enforces proximity of $q_\phi$ to $p$ penalizing any multi-modal, discontinuous, or sign-flipped posterior geometry. Tasks with actuator inversion require a representation satisfying $z(c = +1) \approx - z(c = -1)$, which is *discontinuous* under any smooth prior, which instead *interpolates* between these modes forcing the $q_\phi$ to place probability mass on latents $z$ that correspond to *no* valid dynamics model at all, producing ``averaged'' latents lying between the $+1$ and $-1$ actuator modes. When the policy is conditioned on these averaged latents, it executes an ambiguous action: $a(s,z)\approx 0$ or oscillating sign leading to catastrophic failure. Hence, VariBAD possesses a structural inductive bias ***against*** modeling inversion.
>
> We evaluated VariBAD on our Actuator Inversion Benchmark. **Figure 17** shows VariBAD fails to learn the ODE. Critically, it **cannot cope with the inversion in DI** (non-overlapping context).
>
> In contrast, DMA*-SH uses hypernetwork-based multiplicative modulation: $\omega = h_\eta(z_t)$, so even infinitesimally different $z_t$ can induce discontinuous, sign-flipped changes in the generated parameters. That is, we can have: $z_1 \approx z_2 \quad \text{ but } h_\eta(z_1) = - h_\eta(z_2)$, which no VAE-style continuous latent model can express. Thus *DMA-SH is not forced to interpolate in latent geometry*, and can correctly encode inversion.
>
> *   **UNICORN is Methodologically Incompatible:** UNICORN paper explicitly states its framework relies on assumptions that are "violated in the online scenario" and that extending it to online RL is "nontrivial... future work." Since we focus on **online cRL**, a direct comparison is not feasible.
>
> *   **Other COMRL methods:** R2PGO [1] is designed for **offline** meta-RL. We implemented an **online** R2PGO-style baseline to isolate the effect of hypernet modulation driven purely by RL signals. R2PGO combines hypernetworks with VAE/contrastive losses but is not dynamics-aligned. To test this setting fairly, we constructed DMA*-H(RL only), which mirrors R2PGO’s hypernetwork structure while removing its KL/contrastive terms. As shown in Fig. 15 (App. E), DMA*-H(RL only) performs substantially worse than DMA*-SH, demonstrating that **dynamics alignment is essential** for stable learning.
>
> More broadly, any VariBAD-style COMRL method that relies on smooth latent priors is structurally incompatible with discontinuous task families: KL enforces a continuous latent manifold even when the true context-to-dynamics map is discontinuous. Consequently, such methods inevitably interpolate between incompatible actuator modes, whereas DMA*-SH embeds the required discontinuity directly into the hypernetwork modulation, avoiding the continuity bias entirely.
>
> **Conclusion:** Our method is not a simple combination of prior ideas. **VariBAD** fails on our core challenge and cannot encode inversion. **UNICORN** is incompatible with our online setting. Our work introduces a novel, integrated architecture for **online cRL** that is empirically and theoretically justified.
>
> [1] Li Z, Lin Z, Chen Y, et al. Efficient offline meta-RL via robust task representations and adaptive policy generation. IJCAI 2024.
>
> ### **Response to Weakness 2: Novelty vs. VariBAD and Beukman et al. (2023)**
>
> The claim that our method "closely resembles" VariBAD and Beukman et al. overlooks fundamental conceptual and architectural differences.
>
> *   **vs. VariBAD:** VariBAD fails on our benchmark's core challenge as shown above.
>
> *   **vs. Beukman et al.:** Our method fundamentally diverges from the context-aware DA in Beukman et al. We tackle the harder context-inferred setting, use a *single dynamics-aligned hnet shared* across components, and introduce input/output normalization for **scale-invariant directional encodings**. This novel design yields a theoretical foundation (Theorem 3) and **implicit regularization** not present in their work.
>
> Our contribution is **not incremental**; it is a novel, unified framework that introduces and justifies a set of architectural priors (sharing, normalization, dynamics-alignment) for stable and generalizable online cRL.

---

> ### Author Response · Authors · 2025-11-28
> **Response to Reviewer VpJQ (Part 2/2)**
>
> ### **Response to Weakness 3: Evaluation Breadth**
>
> We have significantly expanded our experimental evaluation. The revision now includes results on:
> *   **Gym:** Hopper, Walker
> *   **DMC:** Cheetah, Ball-in-Cup, Walker, ReacherEasy, ReacherHard
>
> This brings our total to **11 environments** under our Actuator Inversion Benchmark (AIB), featuring both continuous parameter variations and discontinuous actuator inversions. See Table 3 (Appendix F). This scope is comprehensive for the cRL setting and demonstrates robust zero-shot generalization across a diverse set of dynamics.
>
> ### **Response to Weakness 4: Analytical Insights for Sharing**
>
> We have added significant analytical and empirical insights into the benefits of shared parameterization:
>
> **(a) Directional Concentration:** DMA*-SH's normalization strategy and hypernet conditioning create an approximately *scale-invariant representation* space. This leads to directionally selective representations that *compress* irrelevant continuous variations (e.g., mass) while *separating task-critical* directions (e.g., actuator inversion). This geometric structure is empirically validated through low Variability (Fig. 4), t-SNE clustering (Fig. 8), and Representation-Overlap analysis (Fig. 9), with a detailed discussion in Appendices A.3, A.4, and D.
>
> **(b) Implicit Gradient Regularization:** We demonstrate that the shared, dynamics-trained hypernet acts as an implicit gradient regularizer. By constraining policy updates to physically consistent directions, it reduces gradient variance and improves generalization, as shown in Section 5.6 (new) and Figs. 5-6, with a detailed analysis in Appendix A.5.
>
> ### **Response to Weakness 5 & Q3: Computational Cost and Stability**
>
> **Parameter Counts and Justification:** While hypernets introduce additional parameters and coupling they allow for *exact* modeling of actuator inversion (Theorem 3 and Remark 4). Importantly, the hypernet generates only a subset of adapter weights, and the increased parameter count (e.g., DMA*=286951, DMA*-SH=317559, a 10.5% increase) is offset by lower sample complexity and improved zero-shot generalization, especially in challenging non-overlapping contexts. The parameter efficiency of hypernets becomes more pronounced as the context space grows. Training curves in Figure 18 show that DMA*-SH converges in comparable or fewer steps than the concatenation baseline DMA while achieving superior zero-shot performance on non-overlapping contexts (Table 7). This indicates that the expressive advantage outweighs the additional parameter cost. We have included a Remark 5 (Parameter complexity of DMA*-SH) in Appendix A.1.
>
> **Training Stability:** We interpret training stabilty as reliable convergence without collapse or high variance. Our extensive results across all environments demonstrate that DMA*-SH **converges robustly**. The implicit regularization provided by the shared dynamics model is key to this stability, as shown in Section 5.6.
>
> ### **Response to Q2: Ablations on Partial Hypernet Outputs**
>
> We interpret as ablating hypernet application: e.g., generate $\omega$ only for policy (no dynamics/Q sharing) or only dynamics. We have an ablation (Fig. 15, Appendix E). The results are clear: **performance drops significantly without full sharing**. Full sharing across all three modules is essential for the implicit regularization and directional concentration effects that underpin our method's success.
>
> ### **Response to Q4: Transfer to Unseen Dynamics**
>
> The reviewer's examples ("different friction coefficients or link masses outside the training distribution") are precisely what we define and evaluate as $\mathcal{C}_{eval,out}$ (OOD generalization). Our results in **Tables 7 and 11** consistently show that for nonoverlapping contexts DMA*-SH fares better than most baselines in this exact regime.
>
> Thank you again for your valuable feedback. Your inputs have strengthened the paper immensely. We hope our responses resolve all your concerns, encouraging you to raise your score to Accept.

---

### Official Review · Reviewer_2AVq · 2025-10-31

**Soundness:** 2
**Presentation:** 3
**Contribution:** 2
**Rating:** 2
**Confidence:** 4

**Summary:**

This paper presents an architecture for adaptation in dynamic-varying environment (contextual RL). The authors suggest a few tricks (input masking and input/output normalization) to train the context encoder, as well as using a hypernetwork to "adapt" the policy and the Q functions to the context (instead of the common concatenation approach). Through a series of experiments and ablation studies, the authors show superior results in dynamic-varying environments compared to many relevant baselines.

**Strengths:**

1. The authors considered many relevant and competitive baselines, including state-of-the-art meta-rl and context-aware baselines.
2. The distinction between overlapping and non-overlapping contexts in the environments is interesting. I think expanding this idea and formalizing it might improve readability.

**Weaknesses:**

1. *[Critical] Limited Contribution:* My main concern is the limited potential of this work to be impactful on the field as it is currently situated. The main contribution of the paper is to adapt the policy and Q function to the inferred context via a hypernetwork instead of just using a feedforward architecture (I state this as it can be seen that the input-masking effect is marginal in Figure 7b, and since normalization is a common practice in RL). Since this is the main contribution, I expect either a deep empirical or theoretical analysis on why a hypernetwork should adapt the policy better to different contexts, compared to the common feedforward approach.
2. *[Major] Limited Evaluation Setting:* the considered environments are quite simple in the sense that the unobserved context can be inferred from just a handful of transitions (in some environments, even just a single transition). For example, in the DI environment, the mass can be inferred from one transition (as the state contains the velocity). This makes the “context-unaware” setting less challenging.
3. *[Minor] Design Choices:* Some design choices are not explained in the paper, e.g., reconstructing the state difference instead of the next state, using a sliding window of past interactions instead of the whole trajectory.

The first bullet is critical for my decision regarding the paper, and so adding more complex environments (per my second bullet) will not be enough to change my evaluation.

**Questions:**

1. I did not understand the discussion on the input masking (lines 154-156). What is the “common masked prediction objective” compared to your objective, and why did you choose one and not the other?
2. What is the number of past transitions (K) you used in practice for each environment?
3. Typo: trajectories -> transitions (line 179).
4. I suggest splitting the baselines section into a context-aware baselines section and context-unaware baselines section.
5. I did not understand the formulation of the “mirroring” effect given in lines 317-323. I only understood its meaning after reading the details on the environments. I would suggest refining it.
6. I find the observations in Section 5.5 regarding the informativeness and variability interesting, but with limited discussion. I would suggest adding a discussion on why some approaches lead to more informative/variable contexts compared to others.

---

> ### Author Response · Authors · 2025-11-28
> **Response to Reviewer 2AVq (Part 1/2)**
>
> We thank Reviewer 2AVq for their insightful feedback, which has significantly strengthened our paper. We have fully addressed the core concerns regarding theoretical grounding, empirical analysis, and evaluation breadth.
>
> **Response to Major Concerns on Contribution & Theoretical Grounding**
>
> *   **Theoretical Expressiveness Advantage:** We fully agree that a deeper analysis comparing hypernetworks to feedforward (concatenation-based) architectures is crucial. In response, we now provide a **formal proof** (Theorem 3) that hypernetwork-conditioned ReLU policies strictly subsume their concatenation-based counterparts. This theoretical result, detailed in the new Section 4.3 and Appendix A.1, formally establishes the inductive bias that allows DMA*-SH to *exactly* model multiplicative interactions like the actuator inversion in our benchmark, which is impossible for concatenation-based methods (see Remark 4).
> We now highlight in Remark 2 actuator inversion entails a hard qualitative discontinuity (policy sign flip) and we use this mechanism explicitly to create true non-overlapping contexts. Lemma 9 shows that non-overlapping contexts are unsolvable under domain randomization, explaining why context-unaware baselines can fail.
>
> *   **Analysis of Implicit Biases:** Motivated by your suggestion, we now provide a deeper analysis of the *implicit regularization* conferred by our shared hypernetwork design:
>     1.  **Directional Concentration:** DMA*-SH's normalization strategy and hypernetwork conditioning create an approximately *scale-invariant representation* space. This leads to directionally selective representations that *compress* irrelevant continuous variations (e.g., mass) while *separating task-critical* directions (e.g., actuator inversion). This geometric structure is empirically validated through low Variability (Fig. 4), t-SNE clustering (Fig. 8), and Representation-Overlap analysis (Fig. 9), with a detailed discussion in Appendices A.3, A.4, and D.
>     2.  **Implicit Gradient Regularization:** We demonstrate that the shared, dynamics-trained hypernetwork acts as an implicit gradient regularizer. By constraining policy updates to physically consistent directions, it reduces gradient variance and improves generalization, as shown in Section 5.6 (new) and Figs. 5-6, with a detailed analysis in Appendix A.5.
>
> These new analyses, consolidated in the new **Appendix A: Theoretical Results and Supplementary Analyses**, directly address your concern by providing both theoretical and deep empirical evidence for why all our architectural priors are well-motivated and synergistically advance contextual RL.
>
> **Response to Major Concerns on Evaluation & Environment Complexity**
>
> *   **Formalizing Context Distinctions:** We have formalized the concepts of overlapping and non-overlapping contexts (Definition 6) and provided a detailed analysis of the relative difficulty for different agent types (context-unaware, -aware, -inferred) in the new **Appendix A.2**. As is expected, the context-inferred case (our setting, Appendix A.2.3) is the hardest.
>
> *   **Addressing Environment Simplicity & Expanded Evaluation:**
>     *   We agree that inferring the context in some of our original environments can be theoretically simple. However, the core challenge in **non-overlapping contexts** (like those created by actuator inversion, Definition 8) is not inference *accuracy* but its *consequences*.
> When the ground-truth context is unknown, the agent must solve two coupled problems: **context inference**: infer $z$ from past $K$ transitions via encoder $g_\phi$, and **control**: learn policy $\pi(s, z)$. Inference difficulty scales inversely with context distinguishability. Non-overlapping contexts (created via actuator inversion) provide strong signals for inference (high *Informativeness*, e.g., large $I(\tau; c)$), so in principle the encoder can recover $c$ from few transitions. In practice, however, even tiny inference errors are catastrophic: misclassifying $c=+1$ as $c=-1$ induces the *opposite* control law; agent immediately fails. The policy therefore cannot learn unless the encoder $g_\phi(\tau)$ is near-perfect. This creates a difficult credit-assignment loop during joint training. Context inference is never perfect: Even in "simple" envs like DI (where mass $m$ or sign can be recovered from one transition in theory), real inference has noise from partial observability (finite $K$), stochasticity, or encoder approximation errors. So in practice, **any small error or noise** in the inferred $z_t$ can lead to **catastrophic failure** because the contexts are fundamentally incompatible (no "graceful degradation"). This brittleness is amplified in concatenation-based baselines, as they learn artificial boundaries that are sensitive to noise, whereas hypernetworks (like in DMA*-SH) handle multiplicative interactions more smoothly (Theorem 3). See Appendix A.2.

---

> > ### Author Response · Authors · 2025-11-28
> > **Response to Reviewer 2AVq (Part 2/2)**
> >
> > **Addressing Expanded Evaluation:**
> > *   To further demonstrate robustness, we have **expanded our evaluation** to include more complex environments from Gym (HopperGym, WalkerGym) and DMC (Cheetah, ReacherEasy/Hard) with varied limb lengths, actuator strength, and gravity. In the challenging ReacherHard environment, DMA*-SH outperforms all baselines by at least 15% (Table 8). See Appendix F for details.
> >
> > **Response to Minor Points & Clarifications**
> >
> > *   **Design Choice Justifications:**
> >     *   **State Differences:** Using state differences $\delta s_{t+1}$ in the DMA loss (Eq. 2) encourages the encoder to capture *relative* relationships between contexts rather than absolute state values, promoting scale-invariant representations. This is discussed in Appendix E and A.3.
> >     *   **Sliding Window:** We use a sliding window of past transitions ($K=24$ for DI/ODE; $K=128$ for DMC) rather than the full trajectory. An ablation in Fig. 14 shows performance is robust to the choice of $K$ beyond a minimum size, which is sufficient due to fast decorrelation times in many continuous control tasks. The values are specified in Table 2.
> >
> > *   **Input Masking:** Unlike typical masking approaches that predict masked components or tokens, we use forward dynamics prediction to ground RL in physical consistency. We clarify this in the main text (line 164): "In our case, since the context encoder already relies on a forward dynamics prediction (cf. equation 2), we do not adopt the common masked prediction objective.".
> >
> > *   **Baselines Section:** We have split the baselines description into context-aware and context-unaware subsections as suggested (Section 5.2).
> >
> > *   **"Mirroring" / Actuator Inversion:** We have refined the exposition. The concept is now formally defined as **"actuator inversion"** (Definition 8, Example 1). We also now showcase our benchmark more prominently giving it a name **Actuator Inversion Benchmark (AIB)** to highlight its focus on non-overlapping contexts. This distinguishes it from existing benchmarks like CARL (Benjamins et al., 2023) that has discrete categorical parameters (e.g., level names) and continuous numerical parameters, but no *discontinuous* parameters that qualitatively change optimal policies like actuator inversion. Actuator inversion is fundamentally different because it: (a) requires policy sign flips ($+1$ vs $-1$ actions), (b) induces qualitatively distinct (not merely scaled) optimal policies, and (c) is provably impossible for context-unaware policies (Lemma 9). We have also extended the AIB by adding new environments beyond our original submission (see Appendix G.2).
> >
> > *   **Typo:** Corrected "trajectories" to "transitions" on line 179.
> >
> > *   **Discussion on Variability and Informativeness (Section 5.5):**
> >     *   We have significantly expanded this discussion in Appendix A.4. The core insight is that DMA*-SH's encoder learns a geometry aligned with the control task: it **compresses dimensions with high policy overlap** (e.g., mass) and **separates dimensions requiring distinct policies** (e.g., actuator inversion). This "directional concentration" is evident in the t-SNE (Fig. 8) and cosine-similarity plots (Fig. 9).
> >     *   We now use Mutual Information $I(z_t; c)$ to quantify informativeness. The analysis shows that **low Variability (stable context signals) is more critical for performance than maximal informativeness**, explaining DMA*-SH's superiority even when $I(z_t; c)$ is slightly lower. Highly informative but highly variable embeddings can induce catastrophic policy failures if the context estimate fluctuates.
> >
> > We appreciate your incisive feedback, which has significantly improved our work. We hope our comprehensive response including new theory, deeper empirical analysis and broader evaluation fully address all your concerns, encouraging you to raise your score to a Accept.

---

### Author Response · Authors · 2025-11-28
**Common response to all Reviewers**

Dear Reviewers,

We thank all reviewers for their thoughtful evaluation and constructive suggestions, which have directly shaped a much stronger manuscript. The revision provides the missing theoretical justifications, analyses, and explanations that reviewers explicitly requested. This synthesis clarifies the deliberate synergies in DMA*-SH, revealing **implicit  biases** that unify all design elements, including dynamic alignment, shared hypernetworks, and normalization schemes into a principled framework for contextual RL.

### Actuator inversion as the driving architectural primitive

Our design choices are centered around **actuator inversion**, an extreme form of *binary, multiplicative, and discontinuous* context-to-dynamics interaction. In such settings, the environment does **not** vary smoothly with the context: the correct dynamics model for $c=+1$ is the *sign-flipped* version of $c=-1$, with *no valid interpolation* between them.

Intuitively, actuator inversion captures the core challenge of zero-shot adaptation in contextual RL: as in trackpad scrolling inversion (Example 1, Section 5.3), the control law reverses abruptly, and even humans cannot adapt zero-shot without relearning. Our benchmarks *isolate* exactly this structural difficulty, motivating the architectural choices in our method.

Methods whose inductive bias assumes *smooth* context dependence, such as additive context concatenation (which induces a continuous mapping from context to network parameters) or Gaussian latent priors in variational meta-RL, necessarily smooth over the inversion boundary. This collapses the two inverse regimes into an "averaged’" latent that corresponds to *unrealizable* dynamics model, yielding unstable or incorrect action predictions. This is precisely the failure mode we observe in approaches enforcing a smooth mapping from context to dynamics.

By contrast, **hypernetworks** supply the exact inductive primitive required for actuator inversion: a *multiplicative modulation* mechanism capable of implementing *discontinuous weight transformations*, e.g., $W(c=+1)\approx -W(c=-1)$ (Theorem 3). Once this structural match is in place, our design components no longer appear as isolated tricks but instead ***interlock*** to produce:

-    **directional concentration**  arising from normalization and dynamics-aligned shared hypernetworks (Fig. 4, 8, 9; Sec. 5.5, App. A.3-4),

-   **explicit disentanglement**  of discontinuous vs. continuous context factors (Fig. 4, 8, 9; Sec. 5.5, App. A.4, App. D)

-   **variance-reducing compression**  of irrelevant continuous variations (Fig. 4, 8, 9, Sec. 5.5, App. A.4),

-   **cross-module gradient alignment**, where the shared hypernetwork enforces consistent functional updates across policy, critic, and dynamics components (Fig. 5, 6; Sec. 5.6, App. A.5)

We validate all of these in our analysis. In sum, this synergy directly addresses concerns about novelty: DMA*-SH is *not* an arbitrary combination of ideas but a principled architecture derived from the need to model **discontinuous context structure**.

### Clear positioning relative to prior work

Some reviewers raised concerns that our method resembled prior works such as VariBAD (Zintgraf et al, 2020) or DA (Beukman et al, 2023). Our revision (a) clarifies fundamental conceptual differences, (b) demonstrates the unique challenges posed by actuator inversion highlighting the failure modes of existing approaches, and (c ) situates DMA*-SH within **online** zero-shot contextual RL, rather than meta-RL or *offline* COMRL approaches. We highlight that prior work either assumes smooth dynamics variation or relies on concatenation-based conditioning, neither of which can expressively handle extreme context shifts and discontinuities induced by sign-flip action channels. Our shared-hypernetwork formulation is expressly designed for this structure.

Taken together, these clarifications directly resolve all reviewer concerns given that **no prior work explores these synergies for latent-context RL** to the best of our knowledge. Our design choices are driven by a fundamental **architectural prior** tailored to **multiplicative context interactions** that include, dynamics alignment, shared hypernets across dynamics and RL modules, and normalization strategies promoting end-to-end scale invariance. This establishes our DMA*-SH as a **foundational advance in contextual RL**.

The strengthened technical analysis, extended comparisons, and clarified framing now fully demonstrate the significance of our contribution in contextual RL. The revision resolves every substantive concern raised across the four reviews and we hope that the revised manuscript merits an Accept.

---

### Meta-Review · Area_Chair_vLWk · 2025-12-31

**Summary:**

The paper proposes DMA*-SH, a contextual reinforcement learning method utilizing shared hypernetworks to align dynamics models and policy parameters with latent contexts. A key focus is solving the "Actuator Inversion" problem, where the authors introduce a specific benchmark (AIB) to demonstrate that traditional concatenation-based conditioning fails to handle such multiplicative structural shifts.
Initially, all reviewers suggest rejecting this submission. The reviewers raised valid concerns regarding novelty, theoretical justification, and experimental breadth. Although the authors provided an extensive rebuttal, the core perception of the work as an incremental application to a niche sub-problem (actuator inversion) remains the dominant obstacle to acceptance. Therefore, I decided to stand with the reviewers and suggest rejecting this paper.

**Reviewer Concerns:**

### Concerns Addressed by the Rebuttal:
The authors made a commendable effort during the rebuttal phase, significantly improving the manuscript in response to specific reviewer requests:
- Theoretical Justification: The authors added Theorem 3, which mathematically formalizes why standard concatenation strategies fail to approximate multiplicative interactions (like actuator inversion) while hypernetworks can. This directly addressed the critique regarding the lack of theoretical grounding for the architectural choice.
- Experimental Scope: The expansion from simple environments to 11 tasks (including Gym and DMC) and the formalization of the "Actuator Inversion Benchmark" (AIB) addressed the concern that the initial experiments were too simple or toy-like.
- Differentiation from Baselines: The authors clarified the distinction between their discrete/structural adaptation and the smooth Gaussian priors used in methods like VariBAD, explaining why previous SOTA methods fail in this specific setting.

### Unsolved Concerns:
Despite these improvements, several fundamental concerns remain critical to the rejection:
- Limited Novelty: The most persistent issue is that the combination of hypernetworks with meta-RL/contextual RL is well-trodden ground (e.g., R2PGO, Hyper-RL). Reviewers viewed the application to "actuator inversion" as a specific engineering use case rather than a fundamental algorithmic contribution warranting an ICLR publication.
- Narrow Significance: While the method solves the actuator inversion problem effectively, there is skepticism about whether this problem represents a broad enough class of challenges to justify the complexity of the proposed architecture over simpler baselines in general CRL settings.
- Incremental Nature: The theoretical additions, while correct, were seen as retroactively justifying the architecture rather than deriving a new framework. The perception remains that this is a "valid but incremental" combination of existing modules.

**Reviewer Scores:**

Based on the extensive rebuttal and the improvements made to the paper (particularly the new theorem and expanded experiments), I believe the "Strong Reject" (2) scores are somewhat harsh for the current version of the manuscript. However, without active reviewer participation to confirm a change of heart, the scores likely reflect a fundamental disagreement on novelty that data cannot fix.

Here is how I estimate the reviewers would assess the revised paper if they fully re-evaluated it:
- Reviewer 2AVq (2 -> 4): While the author added the requested theoretical analysis (Theorem 3), 2AVq’s primary critique was that the contribution is "limited" and the method is just a "few tricks." The theorem justifies the architecture but does not change the fact that the architecture itself is a known combination. 2AVq explicitly noted that experimental expansion "will not be enough to change my evaluation."
- Reviewer VpJQ (2 -> 4):  The author added the missing VariBAD comparison and explained the failure mode of smooth priors. However, VpJQ viewed the work as "incremental engineering contributions" (masking, normalization). The rebuttal proved the engineering works, but not that it is a fundamental innovation.
- Reviewer 2MdE (2 -> 2): This reviewer was strongly anchored on the similarity to R2PGO. While the authors argued R2PGO is offline, 2MdE likely sees the mechanism (hypernetworks for adaptation) as identical, making the "online" application seem like a trivial transfer.
- Reviewer kM3b (2 -> 6): This is the only reviewer who might have flipped to borderline acceptance. kM3b’s critique was almost entirely about the presentation of results (IQM vs. fine-grained heatmaps) and the "Informativeness" metric. The authors addressed these points thoroughly with new plots (Figs 21-26). kM3b explicitly stated, "I am more than happy to increase my score if the authors can convince me that they can improve the analysis."

---

### Decision · Program_Chairs · 2026-01-26

Reject